# Taxonomic Revision of *Tipula* (*Vestiplex* Bezzi) Crane Flies (Diptera, Tipulidae) in Taiwan with Descriptions of Six New Species [note 1]

**DOI:** 10.3390/insects14070616

**Published:** 2023-07-08

**Authors:** Pavel Starkevich, Chen W. Young

**Affiliations:** 1Nature Research Centre, Akademijos 2, LT-08412 Vilnius, Lithuania; 2Section of Invertebrate Zoology, Carnegie Museum of Natural History, Pittsburgh, PA 15213-4080, USA; dolichopeza@gmail.com

**Keywords:** distribution, hypopygium, new species, new synonymy, ovipositor taxonomy, Tipulinae

## Abstract

**Simple Summary:**

The crane fly subgenus *T.* (*Vestiplex*) includes 189 species level taxa which are distributed throughout the Nearctic, Palaearctic, and Oriental Regions. In this work, the *T.* (*Vestiplex*) fauna of Taiwan is revised, entailing the recognition of 18 species, including six new species. DNA barcode sequences are also provided.

**Abstract:**

The crane fly subgenus *Tipula* (*Vestiplex*) Bezzi, 1924 (Diptera, Tipulidae) of Taiwan is taxonomically reviewed, entailing the recognition of 18 species, with six of these species newly described, including *T.* (*V.*) *diamondi* **sp. nov.**, *T.* (*V.*) *formosae* **sp. nov.**, *T.* (*V.*) *graciliana* **sp. nov.**, *T.* (*V.*) *pseudobiserra* **sp. nov.**,  *T.* (*V.*) *survilai* **sp. nov.**, and *T.* (*V.*) *taiwanica* **sp. nov**. Three species, *T. biaciculifera* Alexander, 1937, *T. niitakensis* Alexander, 1938, and *T. pseudobiaciculifera* Men, Xue and Wang, 2016, listed previously as members of the subgenus *T.* (*Pterelachisus*) Rondani, 1842, are now formally placed as member taxa within *T.* (*Vestiplex*). *Tipula subnata* Alexander, 1949 and *T.* (*V.*) *longarmata* Yang and Yang, 1999 are designated as junior synonyms of *T.* (*V.*) *coxitalis* Alexander, 1935; *T.* (*V.*) *nokonis* Alexander, 1935 is designated as a junior synonym of *T.* (*V.*) *terebrata* Edwards, 1921; and *T.* (*V.*) *takahashiana* Alexander, 1938 is designated as a junior synonym of *T.* (*V.*) *bicornigera* Alexander, 1938. Included in this taxonomic revision are a key to species, species diagnoses for all species, complete descriptions for newly described species, and illustrations of the male genitalia for all species, and for female terminalia when available. DNA barcode sequences for all Taiwanese species of *T.* (*Vestiple*x) are provided. Males are associated with conspecific females based on CO1 results and maximum likelihood trees resulting from the analyses are presented. The Taiwan fauna of *Tipula* (*Vestiplex*) is highly endemic with 16 of the 18 species restricted to the island. At the species group level, no groups are endemic to Taiwan and the groups show closest biogeographic relationships to the Qinghai–Tibetan Plateau.

## 1. Introduction

*Vestiplex* was first proposed by Bezzi [1] as a subgenus of the genus *Tipula* by original designation for the type species *Tipula cisalpina* Riedel, 1913, a West Palaearctic species (Italy, Switzerland). The subgenus *Vestiplex* was originally characterized by a reduction of wings and halteres in females. Edwards [2] later indicated that both wing and halter reductions occur in a number of unrelated lineages within the genus *Tipula* and thus could not be treated as a reliable subgeneric character. Edwards also observed several characters that are shared by the two species originally included in the subgenus *Vestiplex*, *Tipula cisalpina* Riedel, 1913 and *Tipula hemapterandra* Bezzi, 1924, including Rs long, fully one and half times the length of m-cu; a dark spot at the origin of Rs; M4 arising near the middle of the discal cell; squama bare; branches of M with short macrotrichia; tibial spurs 1-2-2; claws of male simple; pleurae bare; nasus absent; in the ovipositor with cerci very strong and thick, horizontally widened, with external edge serrate; and hypovalvae rudimentary or absent. He also pointed out that all these characters, especially those of the ovipositor, are also found in the species of *Tipula arctica* Curtis, 1835 group, and concurred with the usage of *Vestiplex* as the subgeneric name for this group.

Additional characters of *T.* (*Vestiplex*) were provided by Alexander [3] as more species were included in this subgenus. The combined diagnostic characters he compiled that separate the subgenus *T.* (*Vestiplex*) from other subgenera included antenna short to elongate, with flagellum strongly incised in males of some species; verticils of moderate length; frontal prolongation of head elongate; nasus present or absent; thoracic pleura mostly glabrous; tibial spurs 1-2-2; claws simple or with a basal tooth in males; legs stout in females; wings marmorate or submarmorate; squama naked; branches of M with sparse to abundant trichia; R1 + 2 entire; Rs long, fully one and half times or longer than that of m-cu; m-cu at or close to fork of M3 + 4, approximately opposite one third to one half length of cell 1st M2; and ovipositor with reduced hypovalvae and distinct serration on outer margin of elongated and heavily sclerotized cerci.

Both Edwards [2] and Alexander [3] recognized that the external structures of the male hypopygium of the subgenus *T.* (*Vestiplex*) differ significantly among species. The tergite 9 of the male hypopygium varies from the heavily sclerotized, shallowly depressed area forming a “tergal saucer” to a tergite completely divided longitudinally by a pale membrane. Due to these complexities of the male hypopygium, the character used to characterize the subgenus *T.* (*Vestiplex*) has been mostly based on the female ovipositor, which is particularly horizontally widened, heavily sclerotized cerci with a serrated outer margin. Alexander [4] later also noted that the hypovalvae of many species are very small and rudimentary, not extending beyond the bases of the cerci, and the outer marginal serration on the cerci were absent in several Asiatic species of *T.* (*Vestiplex*).

The catalogue of the craneflies of the world [5] recognized 189 valid *T.* (*Vestiplex*) species and eight subspecies. This subgenus is a surprisingly large group distributed throughout the Holarctic and Oriental regions with the highest species diversity in the East Palaearctic (91 species) and Oriental (83 species) regions [5]. The species of subgenus *T.* (*Vestiplex*) have been separated by various authors into 17 species groups based on the characters of the male hypopygium. The *coquillettiana*, *erectiloba*, *excisa*, *leucoprocta*, *nubeculosa*, *scripta*, and *virgatula* groups are characterized by tergite 9 forming a shallowly concave saucer ([6,7,8,9,10,11]). Another 10 species groups are characterized by a longitudinally divided tergite 9, along with features such as the shape of the ventral portion of tergite 9, the shape of the appendages of sternite 9, and other features discussed below: *himalayensis* species group (Alexander [12,13] and Edwards [14], as a subgroup of *avicularia*); *avicularia*, *divisotergata,* and *subtincta* [10]; *arisanensis* and *coxitalis* [4] (as members of *T.* (*Oreomyza*)); *bicornigera*, *deserrata,* and *eurydice* [15] and *immota* [16].

There were 10 species of *Tipula* (*Vestiplex*) from Taiwan described by F. W. Edwards and C. P. Alexander between 1920 and 1938. Since then, our knowledge of Taiwanese crane flies, including *Vestiplex*, has been expanded due to an extensive collection effort by C. W. Young. This recent entomological field work in Taiwan has focused on a comprehensive survey of Taiwan’s natural areas, especially in ecologically restricted habitats of montane systems at higher elevation, in order to taxonomically revise the island’s Tipuloidea fauna. Results from these new samplings from 2010 to 2014 field research have revealed a remarkable diversity of previously unknown Tipuloidea species in many lineages, including numerous undescribed species, as well as expanded our knowledge of the geographic and ecological distributions of many previously described species [17].

This is the fourth contribution of a faunistic research project dealing with the Tipuloidea of the island of Taiwan. This paper represents an updated taxonomic treatment of Taiwanese *T.* (*Vestiplex*) species, based on this recent field sampling. It updates the original 10 species of *T.* (*Vestiplex*) to 18 species with 15 of these endemic to the island, six new species described here, and four species treated as junior synonyms. Along with the new species described below, we provide redescriptions and figures of the previously known species because the original descriptions were overly simplified and imprecise. We also provide an identification key, illustrations, and additional distribution records for all the Taiwanese *T.* (*Vestiplex*) species.

Species delimitation in this study was achieved based on both morphological and molecular data. A CO1 barcode library for all the *T.* (*Vestiplex*) species of Taiwan, including species from nearby geographic regions for comparative purposes, has been established. This CO1 barcode library provides the ability to identify all life stages and the ability to associate individuals of different sexes represents an important extension of existing identification capabilities. The assembly of this CO1 library for the crane flies of Taiwan will not only provide insights into the origins and extent of crane fly diversity, but it will create a new tool for both routine and challenging identifications. Species previously described from a single sex (usually males) now have confirmed associations with the opposite sex, based on DNA sequence association.

## 2. Materials and Methods

Abbreviations for institutional collections used herein:

BMNH Natural History Museum, London, United Kingdom;

CMNH Carnegie Museum of Natural History, Pittsburgh, Pennsylvania, USA;

NRC Nature Research Centre, Vilnius, Lithuania;

USNM United States National Museum, Washington, D.C., USA;

ZFMK Museum Alexander Koenig, Bonn, Germany.

This study is based mainly on field-collected specimens with additional specimens borrowed from several institutions. Field specimens were collected by hand net or light sheet and preserved dry and mounted on paper points following Byers [18]. Genital preparations were made by soaking the three posterior abdominal segments in cold 10% KOH overnight. They were rinsed with acetic acid and water after removal from the KOH, and then stored in glycerin-filled microvials pinned below the corresponding specimens. Young et al. [19] provided detailed methods of collection and preservation.

DNA barcoding is a technique that uses a short DNA sequence from a standardized and agreed-upon position in the genome as a molecular diagnostic for species-level identification. DNA sequence analysis based on the standard gene for animal DNA barcoding, a very short fragment relative to the entire genome of a mitochondrial gene cytochrome c oxidase subunit I (COI), has been employed at the Biodiversity Institute of Ontario (Guelph, ON, Canada) on all the recently collected specimens. The result has been performed to identify genetic clusters and then compared with species identified using morphological methods. Only females that were collected in copula with males or their association with the male as confirmed from CO1 analyses are described in this paper.

CO1 barcode sequence data of 89 specimens belonging to 18 species of *T.* (*Vestiplex*) were obtained, and analyses were performed (Table 1). The resulting sequence data were stored in two projects, SATIP (Tipuloidea of the World) and TIPTW (Tipuloidea of Taiwan) at BOLD (Barcode of Life Data systems). Table 1 lists all the voucher specimens used in the DNA barcode CO1 analyses with associated BOLD sample numbers and CMNH (Carnegie Museum Natural History) specimen numbers. The molecular analyses were conducted using the MEGA11 program [20] to produce a maximum likelihood tree (1000 bootstrapping replicates).

Photographs of whole adult specimens were taken with a Canon Eos 6D camera using MJKZZ automated focus stacking rail set and layers were stacked using the program ZereneStacker. Pictures of dissected terminalia were taken with an Infinity1 camera mounted on a Nikon Si-L stereomicroscope or a Canon EOS 80D camera mounted on an Olympus SZX10 dissecting microscope. The distribution maps were generated using SASplanet software (Release 211230).

Descriptive terminology of the adults generally follows that of Cumming and Wood [21] and de Jong [22] for wing venation with some additions for particular features of *T*. (*Vestiplex*). The term appendage of sternite 9 (A9s) is adopted from Mannheims [23]; the terms ventral lobe and dorsal lobe of appendage of sternite 9 are adopted from Gelhaus [24]; and the term gonocoxal fragment (equal to sclerites *sp1* and *sp2* [25], equal to the genital bridge [26]) is adopted from Brodo [27].

The overall world distribution of species is given according to Oosterbroek [5].

## 3. Results

### 3.1. Systematic Entomology

Order Diptera Linnaeus, 1758

Family Tipulidae Latreille, 1802

Subfamily Tipulinae Latreille, 1802

Genus *Tipula* Linnaeus, 1758

Subgenus *Vestiplex*, Bezzi 1924

*Nigromarginatae* Riedel, [28]: 37

*Tipula*, subg. *Vestiplex* Bezzi, [1]: 230. Type species: *Tipula cisalpina* Riedel, 1913 (original designation). Additional references: Edwards, [2]: 79; Alexander, [29]: 396; [4]: 117; 1965: 355 [30]; Mannheims, [7]: 116; Savchenko, [10]: 132.

Taiwanese *T.* (*Vestiplex*) are small- to large-sized, with the body length of males being 9.0–18.2 mm and females 12.1–32.6 mm. Body coloration varies from yellow to brownish yellow. Antennal (males) length ranges from short, just reaching the pronotum if bent backward, to elongate, reaching almost to the end of the abdomen.

All local *T.* (*Vestiplex*) species are most easily recognized by the striking marbled wing pattern. The wings are pale brown to yellowish brown in ground color and are marmorated with variegated brown areas. Dark brown areas are typically located in the wing bases of cells R and M, at the origin of Rs, at the stigma, and from the anterior cord to cell 1st M_2_. The lighter brown areas are located on the broad wing tips; in cells R, M, and most of Cu_1_; and the ends of Cu, 1st A and 2nd A. The remaining areas of the wing have scattered, various hyaline spots, with some appearing along the wing margins. Wing patterns show intraspecific variation and therefore are not a reliable measure for separating species. Four local alpine *T.* (*Vestiplex*) species have evolved brachypterous females.

The genitalic structure of the male is rather polymorphic and provides means for the specific identification and separation of closed related species. Tergite 9 of the male hypopygium is usually completely divided horizontally by a pale membrane, except in the *arisanensis* species group where dorsally this is a broad pale area. The dorsal portion of tergite 9 has a posterolateral part bearing an elongated arm or short lobe, or is just simple without any modifications. The posterior margin of tergite 9 is simple or with additional lobes. The size of the dorsal portion of tergite 9 varies from relatively large, occupying most of the area, to narrow and medium length. The ventral portion of tergite 9 has a pair of yellow plates or additional blackened armatures; some species have a hypertrophied ventral portion forking into two flattened brown plates.

Sternite 9 has appendages that vary in size from massive and swollen distally to reduced into a small sclerite. The gonocoxal fragment is generally reduced except in several species.

Females usually come with cercus with strong serration on both dorsal and ventral margins; several species have a smooth cercus, without visible serration. Hypovalvae are short and plate-shaped or longer and blade-shaped.

Taiwanese species can be separated into seven species groups: *arisanensis*, *bicornigera*, *coxitalis*, *deserrata*, *divisotergata*, *immota*, and *subtincta*, based on the morphological structures of the hypopygium and ovipositor. Details of these species’ groups are discussed in the corresponding sections.

### 3.2. List of Taiwanese Tipula (Vestiplex)

A total of 18 species of *T.* (*Vestiplex*) are currently recognized in Taiwan, belonging to the following seven species groups:


***Tipula* (*V.*) *arisanensis* species group**


*Tipula* (*Vestiplex*) *arisanensis* Edwards, 1921

*Tipula* (*Vestiplex*) *diamondi* Starkevich and Young **sp. nov.**

*Tipula* (*Vestiplex*) *foliacea* Alexander, 1924


***Tipula* (*V.*) *bicornigera* species group**


*Tipula* (*Vestiplex*) *bicornigera* Alexander, 1938

*Tipula* (*Vestiplex*) *graciliana* Starkevich and Young **sp. nov.**

*Tipula* (*Vestiplex*) *survilai* Starkevich and Young **sp. nov.**


***Tipula* (*V.*) *coxitalis* species group**


*Tipula* (*Vestiplex*) *coxitalis* Alexander 1935

*Tipula* (*Vestiplex*) *sternotuberculata* Alexander, 1935


***Tipula* (*V.*) *deserrata* species group**


*Tipula* (*Vestiplex*) *biserra* Edwards, 1921

*Tipula* (*Vestiplex*) *formosae* Starkevich and Young **sp. nov.**

*Tipula* (*Vestiplex*) *pseudobiserra* Starkevich and Young **sp. nov.**


***Tipula* (*V.*) *divisotergata* species group**


*Tipula* (*Vestiplex*) *nestor* Alexander, 1934

*Tipula* (*Vestiplex*) *niitakensis* Alexander, 1938

*Tipula* (*Vestiplex*) *subapterogyne* Alexander, 1920

*Tipula* (*Vestiplex*) *taiwanica* Starkevich and Young **sp. nov.**

*Tipula* (*Vestiplex*) *terebrata* Edwards, 1921


***Tipula* (*V.*) *immota* species group**


*Tipula* (*Vestiplex*) *bicornuta* Alexander, 1920


***Tipula* (*V.*) *subtincta* species group**


*Tipula* (*Vestiplex*) *parvapiculata* Alexander, 1934

### 3.3. Key to Taiwanese Tipula (Vestiplex) Crane Flies


**Males**


1.Tergite 9 with posterolateral angles produced into flattened, blade-shaped arms (Figure 2A,B) … 2

-Tergite 9 without posterolateral arms … 4

2.Femora brown (Figure 1A). Tergal arms decurved in lateral view (Figure 2A) … *Tipula* (*Vestiplex*) *arisanensis*

-Femora brownish yellow (Figures 5A and 7A). Tergal arms nearly straight or slightly curved (Figures 6A and 8A) … 3

3.Tergal arms elongated (Figure 8A) … *Tipula* (*Vestiplex*) *foliacea*

-Tergal arms short (Figure 6A) … *Tipula* (*Vestiplex*) *diamondi*
**sp. nov.**

4.Gonocoxite massive, elongated, broad at base, apically produced into long arm with tip tapering or expanded (Figures 23A,D,G and 27A,B,E). Sternite 9 near caudal end of either side of midline with a pencil of long reddish setae or basal tubercle tufted with long setae (Figures 23A and 27A) … 5

-Gonocoxite not massive, bearing spine, horn, point or unarmed (Figures 11A, 15A, 19A, 31A, 34A, 39A, 43A, 47A, 51A, 55A, 59A, 63A and 66A) … 6

5.Gonocoxite apically nearly straight (Figure 23A). Sternite 9 with a pencil of long reddish setae (Figure 23A,G) … *Tipula* (*Vestiplex*) *coxitalis*

-Gonocoxite apically produced into gently curved arm with flattened tip (Figure 27A,E). Sternite 9 with basal tubercle tufted with long setae (Figure 27A) … *Tipula* (*Vestiplex*) *sternotuberculata*

6.Gonocoxite apically armed with black spine, horn or small acute point (Figures 11A, 15A, 19A, 31A, 34A, 39A, 63A and 66A) … 7

-Gonocoxite unarmed (Figures 43A, 47A, 51A, 55A and 59A) … 14

7.Gonocoxite apically with small point, ventromesal portion in the shape of blackened arm (Figure 66A,D) … *Tipula* (*Vestiplex*) *parvapiculata*

-Gonocoxite apically armed with black spine or horn (Figures 11A, 15A, 19A, 31A, 34A, 39A and 63A), ventromesal portion simple or with blackened point (Figures 31E, 35C and 39E) … 8

8.Antenna elongated, reaching base or end of abdomen if bent backward (Figures 10A, 14A, 18A and 62A). Gonocoxite with ventromesal portion not modified (Figures 11A, 15E, 19F and 63F) … 9

-Antenna reaching base of wing or only pronotum if bent backward (Figures 30A, 34A and 38A). Gonocoxite with ventromesal portion modified into blackened point (Figures 31E, 35C and 39E). … 12

9.Tergite 9 with ventral portion hypertrophied, forked into flattened plates (Figures 11C, 15C and 19C). Gonocoxite in the shape of straight or apically curved spine (Figures 11F, 15E and 19F) … 10

-Tergite 9 with ventral portion narrow, terminating into a pair of short, blackened processes (Figure 63B,D,E). Gonocoxite horn-shaped (Figure 63A,F) … *Tipula* (*Vestiplex*) *bicornuta*

10.Tergite 9 with ventral portion terminating into a pair of blackened points (Figure 11B–E). Gonocoxite with apical outgrowth dilated at base, tip curved, spine-shaped (Figure 11A,F) … *Tipula* (*Vestiplex*) *bicornigera*

-Tergite 9 ventral portion terminating into a pair of blackened, rounded, or triangular processes (Figures 15C and 19C). Gonocoxite apically with straight spine (Figures 15E and 19F) … 11

11.Flagellar segments elongated, about 6–7 times as long as their middle width (Figure 14B). Femur without yellow preapical ring (Figure 14A) … *Tipula* (*Vestiplex*) *graciliana* **sp. nov.**

-Flagellar segments not elongated, about four times as long as their middle width (Figure 18B). Femur with yellow preapical ring (Figure 18A) … *Tipula* (*Vestiplex*) *survilai* **sp. nov.**

12.Antenna 13-segmented, reaching base of wing if bent backward (Figure 34A). Abdominal segments 1–4 yellow, remaining black … *Tipula* (*Vestiplex*) *formosae* **sp. nov.**

-Antenna 12-segmented, reaching pronotum if bent backward (Figures 30A and 38A). Abdominal segments 1–5 yellow, remaining black … 13

13.Inner gonostylus with dorsal margin without emargination, dorsolateral crest rounded, medial surface without blackened ridge (Figure 31G) … *Tipula* (*Vestiplex*) *biserra*

-Inner gonostylus dorsally with shallow emargination, dorsolateral crest nearly rectangular, medial surface with blackened ridge (Figure 39G) … *Tipula* (*Vestiplex*) *pseudobiserra* **sp. nov.**

14.Sternite 8 laterally provided with group of yellow setae (Figure 59A,E). Sternite 9 ventrally produced into prominent tubercle (Figure 59A,D)... *Tipula* (*Vestiplex*) *terebrata*

-Sternite 8 without group of setae. Sternite 9 without ventral tubercle … 15

15.Tergite 9 with posterolateral angle rounded (Figures 51D and 55B) … 16

-Tergite 9 with posterolatreral angle extended into slender lobe (Figures 43C and 47C) … 17

16.Tergite 9 with deep median U-shaped notch on posterior margin (Figure 51B,D). Ventral lobes without preapical tooth … *Tipula* (*Vestiplex*) *subapterogyne*

-Tergite 9 with shallow median notch on posterior margin (Figure 55B). Ventral lobes with preapical tooth (Figure 55B) … *Tipula* (*Vestiplex*) *taiwanica* **sp. nov.**

17.Tergite 9 blackish, ventral lobes slender, with preapical incision (Figure 43C) … *Tipula* (*Vestiplex*) *nestor*

-Tergite 9 brown, ventral lobes flattened, crescent-shaped (Figure 47C) … *Tipula* (*Vestiplex*) *niitakensis*


**Females**


(Excluding *Tipula* (*Vestiplex*) *diamondi* **sp. nov.** and *T.* (*V.*) *bicornuta* as females of both species are still unknown)

1.Wing reduced, not reaching middle of abdomen (Figures 14D,18D, 46C and 50C)... 2

-Wing well developed, extends beyond middle of abdomen... 5

2.Cercus stocky, dark brown (Figure 53A) … *Tipula* (*Vestiplex*) *subapterogyne*

-Cercus slender, yellow or brownish yellow (Figures 49A, 17A and 21A) … 3

3.Hypovalva extended, blade-shaped (Figure 49A–C) … *Tipula* (*Vestiplex*) *niitakensis*

-Hypovalva short, plate-shaped (Figures 17B and 21A,B) … 4

4.Femur with yellow preapical ring (Figure 18D) … *Tipula* (*Vestiplex*) *survilai* **sp. nov.**

-Femur without yellow preapical ring (Figure 14D) … *Tipula* (*Vestiplex*) *graciliana* **sp. nov.**

5.Cercus extended, blade-shaped (Figures 4A–C, 9A,B, 25A–C, 29B,C, 45A–C, 57A–C and 61A,B) … 6

-Cercus short, plate-shaped (Figures 33A–C, 37B,C, 41A–C and 68A,B) … 12

6.Large species with body length exceeding 24 mm (25.4–25.6 mm). Hypovalva extending before base of cercus (Figure 61A)... *Tipula* (*Vestiplex*) *terebrata*

-Smaller species with body length less than 17.5 mm (10.8–17.5 mm). Hypovalva extending to the base of cercus... 7

7.Cercus shorter that tergite 10 (one and half or two times shorter than tergite 10 (Figures 9A and 25A) …8

-Cercus subequal or the same length as tergite 10 … 9

8.Cercus two times shorter than the length of tergite 10 (Figure 25A) … *Tipula* (*Vestiplex*) *coxitalis*

-Cercus one and half times shorter than tergite 10 (Figure 9A) … *Tipula* (*Vestiplex*) *foliacea*

9.Thorax with longitudinal stripes not bordered (Figures 1B and 26B). Femur without yellow preapical ring (Figures 1A and 26A) … 10

-Thorax with longitudinal stripes bordered (Figures 42B and 54B). Femur with yellow preapical ring (Figures 42A and 54A) … 11

10.Flagellum yellowish brown. Sternite 8 laterally with distinct obtuse lobes (Figure 4B,C) … *Tipula* (*Vestiplex*) *arisanensis*

-Flagellum bicolored. Sternite 8 with lateral lobes small, indistinct (Figure 29B,C) … *Tipula* (*Vestiplex*) *sternotuberculata*

11.Thorax grayish brown (Figure 42B). Sternite 8 with lateral lobe not distinct (Figure 45B,C) … *Tipula* (*Vestiplex*) *nestor*

-Thorax brownish ochraceous (Figure 54B). Sternite 8 with small lateral lobe (Figure 57B,C) … *Tipula* (*Vestiplex*) *taiwanica* **sp. nov.**

12.Cercus narrow (the ratio between length and middle width is about 11 times) (Figure 68A) … *Tipula* (*Vestiplex*) *parvapiculata*–. Cercus not narrow (the ratio between length and middle width is about six times) … 13

13.Femur with obscure yellow subterminal ring … *Tipula* (*Vestiplex*) *bicornigera*

-Femur without yellow subterminal ring … 14

14.Antenna 13-segmented, flagellum dark brown … *Tipula* (*Vestiplex*) *formosae* **sp. nov.**

-Antenna 12-segmented, flagellum bicolored or yellowish brown … 15

15.Hypovalva wavy in lateral view (Figure 41A) … *Tipula* (*Vestiplex*) *pseudobiserra* **sp. nov.**

-Hypovalva not wavy in lateral view, comparatively shorter (Figure 41B) … *Tipula* (*Vestiplex*) *biserra*

### 3.4. Taxonomy


***Tipula arisanensis* species group**


The *arisanensis* group was proposed by Alexander [4] for *T.* (*V.*) *arisanensis* and *T.* (*V.*) *foliacea* and we included *T.* (*V.*) *diamondi*
**sp. nov.** The following species: *T. biaciculifera* Alexander, 1937 and *T. pseudobiaciculifera* Men, Xue and Wang, 2016, recently listed as members of subgenus *T.* (*Pterelachisus*) Rondani, 1842 [31], are placed here according to the features discussed below.

Males of the *arisanensis* species group can be easily recognized by the posterolateral angle of tergite 9 produced into flattened, blade-shaped arms or needle-like spines and the massive dorsal lobe of A9s. They can be also characterized by tergite 9 dorsally undivided, still with a broad pale area present which is more visible on dry specimens; ventral lobes flattened, darkened at margin; gonocoxite without outgrowth; outer gonostylus small, filiform; and inner gonostylus elongated.

Females are known for *T.* (*V.*) *arisanensis*, *T.* (*V.*) *foliacea*, and *T.* (*V.*) *biaciculifera*, with the last species not being illustrated, and a cercus with a smooth margin being mentioned [32]. The same smooth cercus, without visible serration has been found in *T.* (*V.*) *arisanensis* and *T.* (*V.*) *foliacea*. They can be also characterized by sternite 8 flattened from the middle to the lateral angle; hypovalva developed blade-shaped; and median incision between hypovalvae basally with slightly serrated edge and provided with setae.


***Tipula* (*Vestiplex*) *arisanensis* Edwards**


Figure 1, Figure 2, Figure 3 and Figure 4; Figure 1A

*Tipula arisanensis* Edwards, [33]: 109;

*Tipula* (*Oreomyza*) *arisanensis*: Alexander, [29]: 348; [4]: 122;

*Tipula* (*Unplaced*) *arisanensis*: Alexander and Alexander, [34]: 66;

*Tipula* (*Vestiplex*) *arisanensis*: Oosterbroek, [5].


**Diagnosis**


*Tipula* (*V.*) *arisanensis* can be recognized by the distinct flattened, blade-shaped, strongly decurved arm of tergite 9. The wing is clear pale brown, tinged with brown. The female has the cercus straight, without visible serration. Sternite 8 is at lateral angle produced into an obtuse lobe. The hypovalva is blade-shaped.

**Type of material examined. TAIWAN** • Holotype ♀; Formosa, Arisan; 24.v.1917; T. Shiraki leg.; BMNH(#)246104; BMNH • Paratype ♀; same data as for preceding; BMNH(#)246105; BMNH.

**Additional material examined. TAIWAN**—**Yilan Co.** • 1 ♂; Cilan Forest Conservation Station; N24.5879, E121.4223; alt. 1637 m; 22 April 2010; Chen Young leg.; CMNH406953; NRC • 1 ♂, 1 ♀; same data as for preceding; CMNH450662; dissected; CMNH448774; dissected; NRC • 6 ♂♂, 4 ♀♀; same data as for preceding; CMNH401544; CO1 Barcode TIPTW052-10; CMNH543670; CO1 Barcode TIPTW059-10; CMNH543677; CMNH449609; CMNH414126; CMNH543676; CMNH416117; CMNH411234; CMNH449051; CMNH543659; CMNH • 2 ♂♂, 2 ♀♀; same data as for preceding; 26 September 2010; CO1 Barcode TIPTW531-10; CMNH544149; CMNH544146 CMNH449227; CMNH452359; CMNH • 1 ♂; Cilan woods around Yuan, Yang Hu; N24.5771, E121.4061; alt. 1606 m; 22 April 2010; Chen Young leg.; CMNH450501; CMNH • 2 ♂, 1 ♀; Ming-Chih Recreation Park; N24.5771, E121.4068; alt. 1119 m; 6 April 2010; Chen Young leg.; CO1 Barcode TIPTW1013-12; CMNH576481; CMNH576538; CMNH576482; CMNH • 1 ♂, 4 ♀; same data as for preceding; 23 April 2010; CMNH404821; CMNH454617; CMNH569612; CMNH450022; CMNH454948; CMNH • 1 ♂, 2 ♀; same data as for preceding; 4 May 2010; CMNH543776; CO1 Barcode TIPTW160-10; CMNH543778; CMNH445399; CMNH • 1 ♂; same data as for preceding; 30 May 2010; CMNH544562; CMNH—**Kaohsiung Co.** • 1 ♀; Taoyuan Township, TengZhih Station; N23.0848, E120.7877; alt. 1806 m; 4 May 2011; Yu-Chuan Chiu leg.; 2007-000062; CMNH • 2 ♀; same data as for preceding; 5 May 2011; 2007-000067; 2007-000009; CMNH—**Nantou Co.** • 1 ♀; Lugu Twp., Sitou Forest Park; N24.106, E121.198; alt. 2292; 17 September 2011; Jin-Chou Chang leg.; CO1 Barcode TIPTW976-12; CMNH576444; CMNH • 1 ♂; Ren-Ai Huei-Sun Forest Station; N24.0879, E121.0340; alt. 769 m; 24 May 2011; Chen Young leg.; CMNH544520; CMNH • 1 ♀; Ren-Ai Township, Hsing-Pai-Young; N24.1980, E121.4326; 1486 m; 18 October 2011; Yu-Da Chuan leg.; 0808-000444; CMNH—**Taipei Co.** • 1 ♂; Shihlin Yangmingshan Da Tun Falls; N25.1270, E121.5885; alt. 235 m; 17 April 2010; Chen Young leg.; CMNH543640; CMNH—**TaiTung Co.** • 1 ♂; Jhihben Li-Jia Forest Road 14K; N22.7986, E121.0334; alt. 1060 m; 2 May 2010; Chen Young leg.; CMNH405392; CMNH—**Taoyuan Co.** • 1 ♂; Fushing Township, Lalashan Cottage; N24.7094, E.121.4332; alt. 1581 m; 6 April 2012; Wen-Chen Chu leg.; CMNH447852; CMNH • 1 ♂; same data as for preceding; Han-Tzu Hsu leg.; CMNH452613; CMNH • 1 ♀; same data as for preceding; 24 May 2011; Chen Young leg.; 2025-000062; CMNH.


**Redescription**


**Male** (Figure 1A). Body length is 9.0–12.0 mm, wing length is 12.1–14.6 mm, length of antenna is 4.2–4.7 mm. General body coloration is brownish ochreous.

Head. Ochreous, with median longitudinal black line on vertex and occiput. Rostrum short, stout, with nasus. Palpus brown. Antenna 13-segmented, if bent backward reaching base of the wing. Scape and pedicel light yellow, first flagellar segment yellowish brown, remaining segments brown. Each flagellomere, except first, with small enlargement. Apical flagellomere small. Long verticils shorter than corresponding segments.

Thorax (Figure 1B). Overall brownish ochreous. Pronotum with brown median line. Prescutum and presutural scutum with four distinct brown stripes, without dark borders. Central stripes confluent anteriorly. Interspace between central and lateral stripes brownish ochreous. Postsutural scutum with median line; two brown spots on each lobe, anterior spot small. Scutellum and mediotergite with dark median line. Pleura yellowish. Coxa and trochanter yellow; femur yellow with brown distal end; tibia brown; tarsal segments dark brown; claw with tooth. Wing clear pale brown, tinged with brown (Figure 1A). Wing cells marbled with darker and clear spots. Brown postarcular darkening in cells br and bm, at origin of Rs and stigma area. A faint brown cloud in the middle of br and bm and the distal area of bm. Vein m-cu joining dm before fork of M_3+4_, M_1+2_ about 0.2 × as long as the length of cell m_1_. Halter ochreous, base of knob darker.

Abdomen. Abdominal segments 1–5 brownish yellow, remaining segments brown. Tergites with darker median and lateral longitudinal stripes.

Hypopygium (Figure 2 and Figure 3). Male hypopygium with tergite 9 large, fused with sternite 9 on basal half (Figure 2A). Tergite 9 with dorsal portion undivided, still medially with broad pale area more visible on dry specimen; posterolateral angle produced into flattened, blade-shaped, strongly decurved arm; tip of arm narrowed outwardly forming acute beak-shaped point (Figure 2A,B). Tip of tergal arm reaches about the same level of tip of inner gonostylus. Posterior margin of dorsal portion with two triangular points. Ventral portion developed into two flattened, nearly oval, light brown plates; inner margins of plates blackened at base (Figure 2C,D,E). Gonocoxite fused with sternite 9, apically rounded, without projection (Figure 3A,B). Outer gonostylus small, narrow, filiform (Figure 3C). Inner gonostylus in the shape of elongated plate; upper beak short and black, dorsal crest with black projection, dorsal margin with yellow lobe; lower beak black, ill-developed (Figure 3D). Appendage of sternite 9 with ventral lobe, long and slender narrow, covered long setae; dorsal lobe long and massive, in the shape of flattened blade (Figure 3A,B). Adminiculum relatively narrow, nearly triangular in ventral view, medially protruded at basal half (Figure 3A,B). A preapical round incision, apex funnel-shaped (Figure 3E). Sperm pump with central vesicle swollen (Figure 3F,G). Compressor apodeme flattened, with median rod (Figure 3G). Posterior immovable apodeme narrow, anterior immovable apodeme short. Aedeagus about 2× as long as sperm pump, basally brown, becoming yellow towards apex (Figure 3F).

**Female** (Figure 1C). Body length is 13.3–15.0 mm, wing length is 12.7–14.7 mm, length of antenna is 2.2–2.5 mm. Body color is the same as that of male except as follows. Antenna shorter, if bent backward reaching before the base of the wing.
Figure 1Habitus of *Tipula* (*Vestiplex*) *arisanensis*. (**A**). Male, lateral view. (**B**). Male thorax, dorsal view. (**C**). Female, lateral view. (**D**). Male right wing, dorsal view. Scale bars: a (**A**,**C**) = 5 mm, b (**B**) = 1 mm, c (**D**) = 5 mm. Abbreviations: A_1_—first branch of anal vein; bm—basal medial cell; br—basal radial cell; c—costal cell; CuA—anterior branch of cubital vein; CuP—posterior branch of cubital vein; dm—discal cell; M_1_—first branch of media; M_2_—second branch of media; M_3_—third branch of media; M_4_—fourth branch of media; m-cu—medial–cubital crossvein; R_1_—anterior branch of radius; R_3_—lower branch of second branch of radius; R_4_—upper branch of third branch of radius, R_5_—lower branch of third branch of radius.
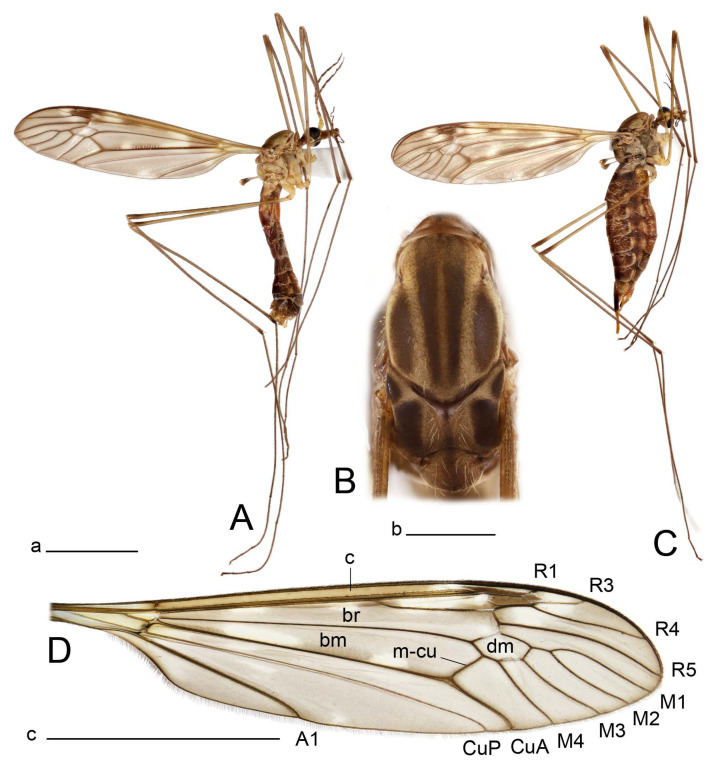



*Ovipositor* (Figure 4). Tergite 10 shiny, dark brown. Cercus brown, shorter than tergite 10, straight, with tip rounded; ventral margin smooth, without visible serration (Figure 4A). Hypovalva not reaching base of cercus. Sternite 8 with hypovalva developed, distally darkened, blade-shaped; median incision between hypovalvae basally with indistinctly serrated edge and covered with setae (Figure 4B–D). Sternite 8 flattened from the middle, at lateral angle produced into obtuse lobe. Sternite 9 nearly shovel-shaped, extended at posterior margin; the surface covered with short filaments (Figure 4E). Furca posteriorly flattened, anteriorly membranous (Figure 4E). Spermatheca nearly spherical (Figure 4F).

**Distribution.** Taiwan (Figure 1A).

**Elevation range in Taiwan.** Adults were collected at altitudes ranging from 200 m to 3000 m.

**Period of activity.** Adults are active from April through to October.

**Remarks**. *Tipula* (*V.*) *arisanensis* was described by Edwards [33] based on two female specimens collected by T. Shiraki in 1917 from Arisan (嘉義阿里山), Taiwan. Alexander [29] designated a male allotype of *T.* (*V.*) *arisanensis* collected by J. L. Gressitt in 1932 from Pianna-ambu (宜蘭南山村), Taiwan. Alexander expressed his association of the two sexes of this species based mainly on the coloration of the external morphology, without giving details of their specified similarity. The male and female associations of this species were confirmed by a DNA gene sequence study.
Figure 2Male terminalia of *Tipula* (*Vestiplex*) *arisanensis*. (**A**). Hypopygium, lateral view. (**B**). Hypopygium, dorsal view. (**C**) Tergite 9, dorsal view. (**D**) Tergite 9, ventral view. (**E**) Tergite 9, caudal view. Abbreviations: dl—dorsal lobe of A9s dp—dorsal portion of tergite 9; ig—inner gonostylus; s9—sternite 9; t9—tergite 9; ta—tergal arm; vp—ventral portion of tergite 9. Scale bars: a (**A**), b (**B**), c (**C**–**E**) = 0.5 mm.
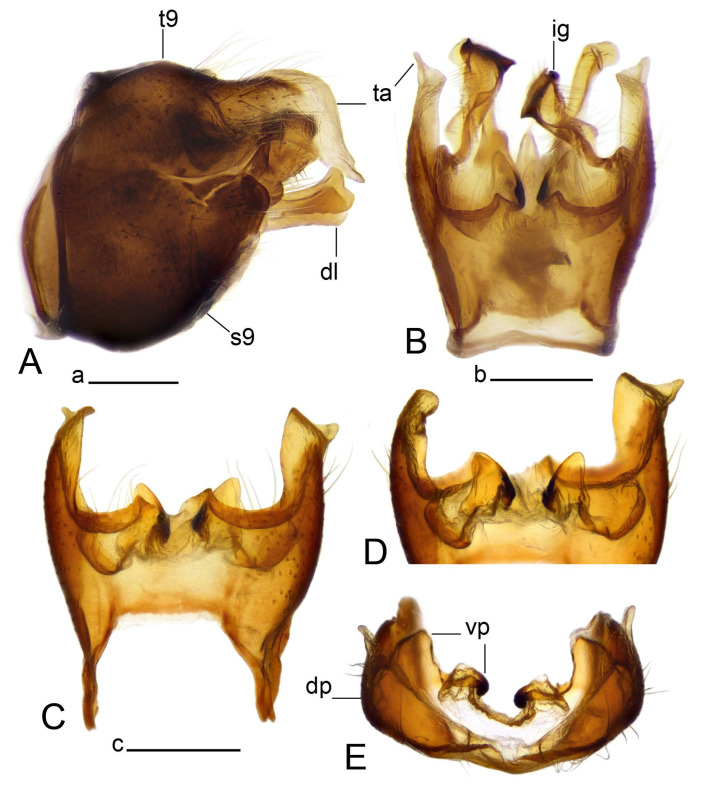

Figure 3Male terminalia of *Tipula* (*Vestiplex*) *arisanensis*. (**A**). Sternite 9, ventral view, gonostyli removed. (**B**). Sternite 9, ventral view, the same sclerite pressed by cover glass. (**C**). Outer gonostylus. (**D**). Left inner gonostylus, lateral view. (**E**). Adminiculum, lateral view. (**F**). Semen pump and intromittent organ, lateral view. (**G**). Semen pump, dorsal view. Abbreviations: adm—adminiculum; aia—anterior immovable apodeme; ca—compressor apodeme; dl—dorsal lobe of A9s; gcx—gonocoxite; pia—posterior immovable apodeme; vl—ventral lobe of A9s. Scale bar: a (**A**–**G**) = 0.5 mm.
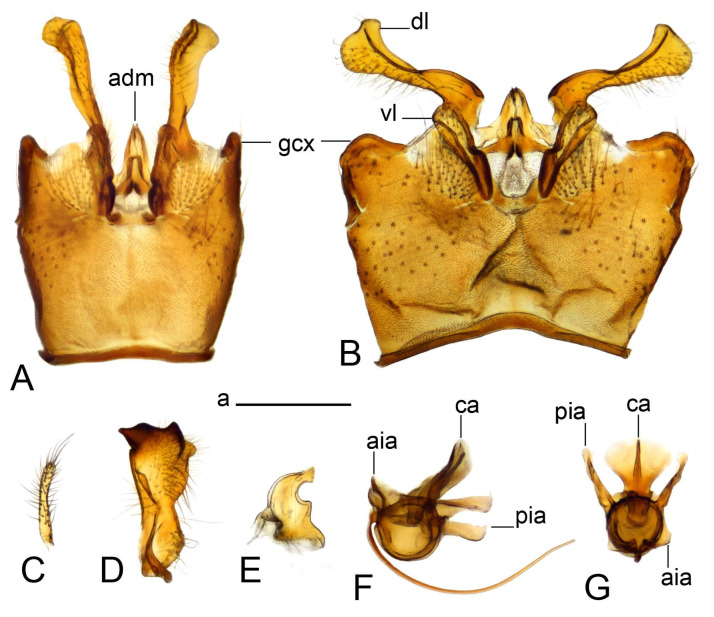

Figure 4Female terminalia of *Tipula* (*Vestiplex*) *arisanensis*. (**A**). Ovipositor, left lateral view. (**B**). Sternite 8 with hypovalvae, ventral view. (**C**). Sternite 8 with hypovalvae, the same sclerite gently pressed by cover glass. (**D**). Basal part of hypovalvae, ventral view. (**E**). Sternite 9 and furca, dorsal view. (**F**). Spermatheca. Abbreviations: c—cerci; f—furca; h—hypovalvae; pll—posterolateral lobe of sternite 8; s8—sternite 8; s9—sternite 9; t10—tergite 10. Scale bar: a (**A**–**F**) = 0.5 mm.
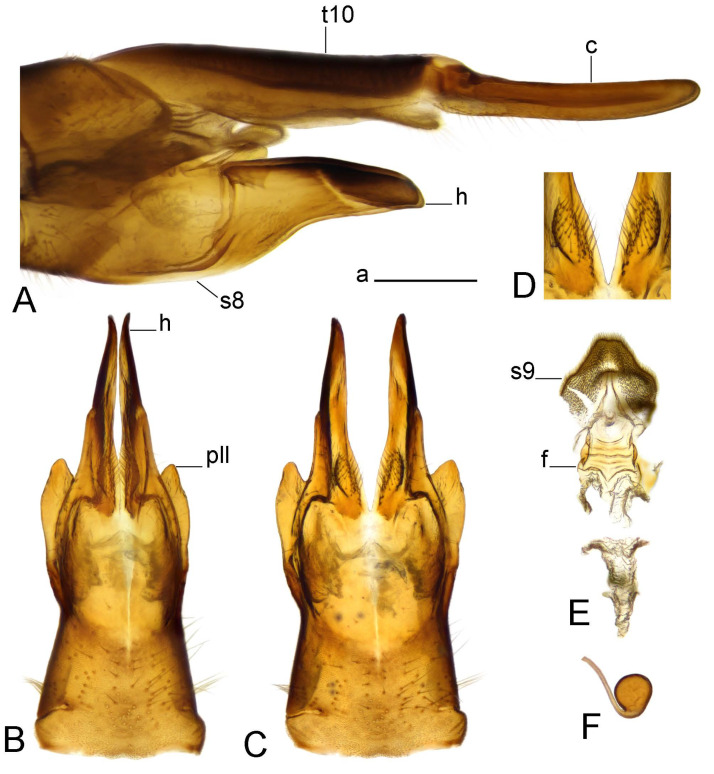



***Tipula* (*Vestiplex*) *diamondi* Starkevich and Young sp. nov.**


Figure 5 and Figure 6; Figure 1B

Differential diagnosis

*Tipula* (*V.*) *diamondi* **sp. nov.** can be easily recognized by the flattened blade-shaped lateral tergal arms. These arms are short and near-parallel-sided, narrowed outwardly to form beak-shaped tips. The female of this species is unknown. *Tipula* (*V.*) *diamondi* **sp. nov.** is closest to *T.* (*V.*) *biaciculifera* (China in Anhui, Zhejiang, and Jiangxi) based on the shape of the tergal arms ([35]: Figure 21). Both species are characterized by their near-parallel-sided tergal arms; still, those are narrow in *T.* (*V.*) *biaciculifera* while *T.* (*V.*) *diamondi*
**sp. nov.** has flattened arms. *Tipula* (*V.*) *diamondi*
**sp. nov.** can be also separated by the distal lobe of A9s which is strongly curved at a right angle, while the lobe of *T.* (*V.*) *pseudobiaciculifera* is distinctly swollen in the distal half ([35]: Figure 26).


**Type of material**



**Holotype**


**TAIWAN** • ♂; Yilan Co.; MingChih Recreation Park; N24.5771, E121.4068; alt. 1119 m; 6 April 2010; Han-Tzu Hsu leg.; CO1 Barcode TIPTW1028-12; CMNH576496; CMNH.


**Paratypes**


**TAIWAN** • 1 ♂; Taichung Co.; Pilushi; [N24.24750, E121.30720]; alt. 2200 m; 22 May 1988; R. Davidson, J. Rawlins, C. Young leg.; CMNH544048; dissected; CMNH • 1 ♂; Heping Twp, Snow Mountains; N24.2453, E121.5368; 2240 m; 30 May 2011; Yu-Chuan Chiu leg.; CO1 Barcode TIPTW903-11; CMNH544821; CMNH.


**Description.**


**Male** (Figure 5A). Body length is 13.2–14.7 mm, wing length is 15.8–17.7 mm, length of antenna is 4.4–5.0 mm. General body coloration is yellow.
Figure 5Habitus of *Tipula* (*Vestiplex*) *diamondi*
**sp. nov.** (**A**). Male, holotype, lateral view. (**B**). Male thorax, paratype, dorsal view. Scale bars: a (**A**) = 5 mm, **b** (**B**) = 1 mm.
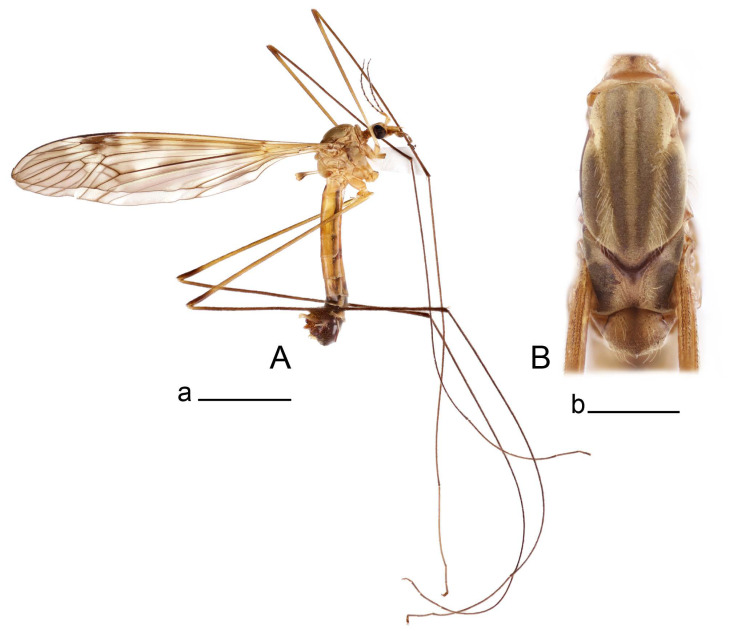



Head. Yellow, vertex and occiput thinly dusted with gray, with median line. Rostrum yellow, nasus distinct. Palpus brown. Antenna 13-segmented, if bent backward reaching base of the wing. Scape and pedicel and first flagellar segment light yellow, flagellar segments 2–4 with bicolor appearance, remaining brown. Each flagellomere except the first, with darkened and slightly enlarged base. Long verticils equal to length of corresponding flagellomeres.

Thorax (Figure 5B). Overall brownish ochreous. Pronotum brownish yellow with darker median spot. Prescutum and presutural scutum with four distinct ochreous brown stripes. Central stripes separated. Lateral stripes indistinctly bordered by lighter brown. Interspaces yellowish gray. Postsutural scutum with median line; two brown spots on each lobe. Scutellum and mediotergite yellowish brown with darker median line. Pleura yellowish brown. Coxae, trochanters, and femora yellow; distal end of femora darkened; tibiae and tarsal segments brown; claw with tooth. Wing clear pale brown, tinged with brown (Figure 5A). Wing cells marbled with darker and clear spots. Three brown spots placed at postarcular area, at origin of Rs and at stigma area. A faint brown cloud in the middle of br and bm and distal area of bm. Vein m-cu joining dm close to fork of M_3+4_, M_1+2_ about 0.15× as long as length of cell m_1_. Halter yellow, knob basally brown.

*Abdomen*. Abdominal segments 1–5 yellow, tergites with distinct lateral stripes, median stripe indistinct. Remaining segments brownish black.

*Hypopygium* (Figure 6). Male hypopygium with tergite 9 large, fused with sternite 9 on basal half (Figure 6A). Tergite 9 with dorsal portion undivided, still medially with broad pale area (Figure 6B); posterolateral angle produced caudad and slightly ventrad into flattened, short, near-parallel-sided, blade-shaped tergal arms. Tip of tergal arm oblique, reaching to about the same level of tip of inner gonostylus. Posterior margin shallow, with a small V-shaped median notch subtended on either side by a large triangular projection. Ventral portion developed into two elongated, distally flattened plates, with outer margin extended and inner one blackened. Gonocoxite fused with sternite 9, nearly rectangular, without projection (Figure 6C). Outer gonostylus small, narrow, filiform (Figure 6D). Inner gonostylus in the shape of elongated plate (Figure 6E); upper beak black, short, dorsal crest round with triangular darkened point. Appendage of sternite 9 with ventral lobe long and slender, covered long setae (Figure 6F); dorsal lobe long and massive, strongly curved at right angle (Figure 6G). Adminiculum relatively narrow, nearly triangular in ventral view, medially protruded with preapical round incision (Figure 6F,H). Sperm pump with central vesicle swollen (Figure 6I,J). Compressor apodeme flattened, with median rod (Figure 6I); posterior immovable apodeme narrow, anterior immovable apodeme round. Aedeagus about 2.3× as long as sperm pump, basally brown, becoming yellow towards apex (Figure 6J).

**Female**. Unknown.

**Distribution.** Taiwan (Figure 1B).

**Elevation range in Taiwan.** Adults were collected at altitudes ranging from 1000 m to 2200 m.

**Period of activity.** Adults are active from April through to May.

**Etymology.** The new species is named after Jared M. Diamond, an American geographer and biologist, best known for his popular science books such as *The Third Chimpanzee* (1991) and *Guns, Germs, and Steel* (1997).
Figure 6Male terminalia of *Tipula* (*Vestiplex*) *diamondi*
**sp. nov.** (**A**). Hypopygium, lateral view. (**B**). Tergite 9, dorsal view. (**C**). Gonocoxite. (**D**). Outer gonostylus. (**E**). Left inner gonostylus, lateral view. (**F**). Ventral lobes of A9s and adminiculum, ventral view. (**G**). Dorsal lobe of A9s. (**H**). Adminiculum, lateral view. (**I**). Semen pump, dorsal view. (**J**). Semen pump and intromittent organ, lateral view. Abbreviations: adm—adminiculum; gcx—gonocoxite; vl—ventral lobe of A9s. Scale bars: a (**A**), b (**B**–**J**) = 0.5 mm.
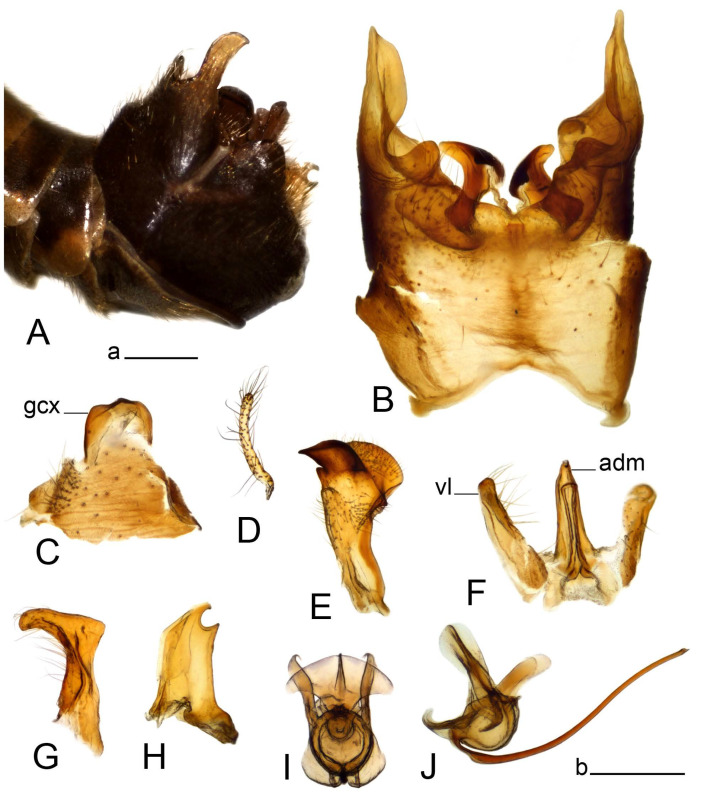

insects-14-00616-sch001_Scheme 1Scheme 1Distribution maps of *Tipula* (*Vestiplex*) crane flies in Taiwan. (**A**). *T.* (*V.*) *arisanensis*. (**B**). *T.* (*V.*) *diamondi* **sp. nov.** Purple circle represents holotype, yellow circles represent paratypes.
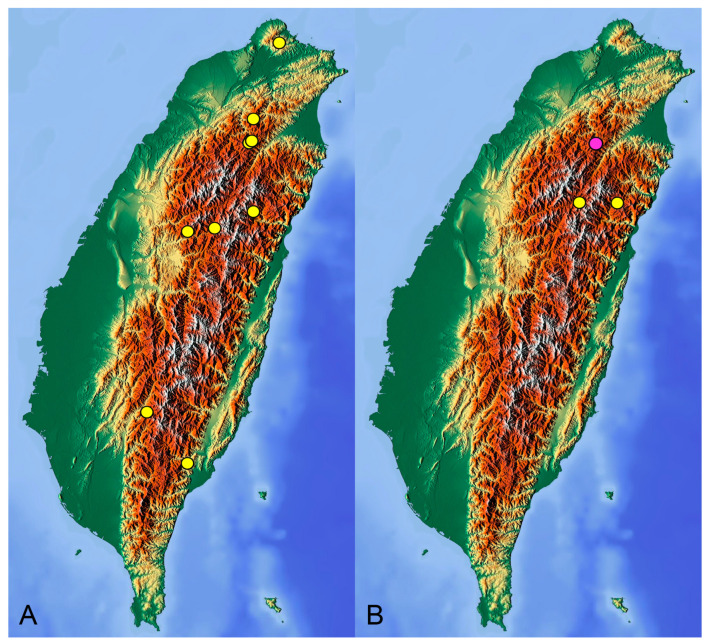




***Tipula* (*Vestiplex*) *foliacea* Alexander**


Figure 7, Figure 8 and Figure 9; Figure 2A

*Tipula foliacea* Alexander, [36]: 73;

*Tipula *(*Oreomyza*) *foliacea*: Alexander, [4]: 122;

*Tipula* (*Vestiplex*) *foliacea*: Alexander and Alexander, [34]: 63.


**Diagnosis**


*Tipula* (*V.*) *foliacea* can be recognized by the visibly elongated, blade-shaped arm of tergite 9 which have outwardly forming oblique tips with a single, minute spur near the tip. The female has the cercus straight, without visible serration and Sternite 8 with a flattened lobe from the middle to the lateral angles. The hypovalva is blade-shaped.
Figure 7Habitus of *Tipula* (*Vestiplex*) *foliacea*. (**A**). Male, lateral view. (**B**). Male thorax, dorsal view. (**C**). Female, lateral view. Scale bars: a (**A**, **C**) = 5 mm, b (**B**) = 1 mm.
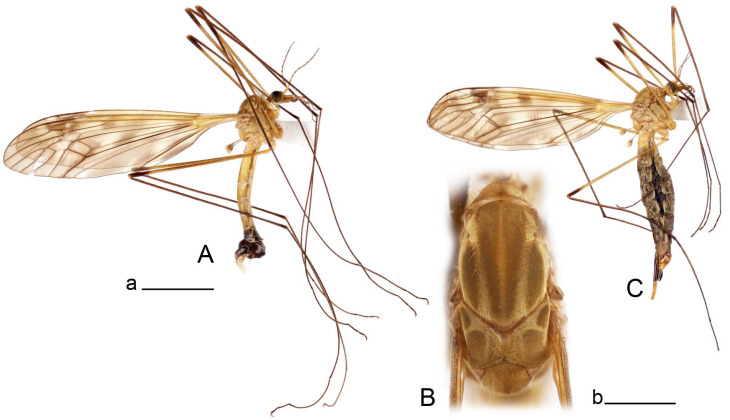


**Type of material examined. TAIWAN** • Holotype ♂; Arisan; alt. 7500 ft.; 9 May 1922; K. Takeuchi leg.; USNM.

**Additional material examined. TAIWAN**—**Chiayi Co.** • 1 ♂; Alishan; [23°21′ N, 120°48′ E]; 12 April 1965; Yoshimoto; CMNH455307; dissected; NRC • 1 ♂; same data as for preceding; CMNH451437; CMNH • 1 ♀; same data as for preceding; CMNH—**Hualien Co.** • 1 ♂; KuanYuan; [N24.1850667, E121.3421278]; alt. 2734 m; 15 May 2007; Ling-Chu Lin leg.; CMNH543834; CMNH • 1 ♂; SiouLin, Hsio-Fon-Ko parking area; N24.1618, E121.2882; alt. 2999 m; 25 May 2011; HuiSheng Fang leg.; CMNH413785; CMNH • 1 ♂, 1 ♀; same data as for preceding; 20 July 2011; CO1 Barcode TIPTW1113-12; CMNH551699; CO1 Barcode TIPTW1114-12; CMNH551700; CMNH—**Yilan Co.** • 1 ♀; Cilan Forest Conservation Station; N24.5879, E121.4223; alt. 1637 m; 26 September 2010; Chen Young leg.; CMNH544155; CMNH—**Nantou Co.** • 1 ♂; Hehuan Mountain, North Peak; [N24.1629140, E121.2878330]; alt. 2995m; 7 July 2008; CMNH403325; CMNH • 1 ♂; Ren-Ai Endemic Species Res. Centr.; N24.1612, E121.2869; alt. 2986 m; 28 June 2011; Chen Young leg.; CMNH404170; NRC • 1 ♀; same data as for preceding; 22 March 2011; Han-Tzu Hsu leg.; CMNH576494; CMNH • 1 ♂; same data as for preceding; Chen Young leg.; CMNH576495; CMNH • 1 ♀; same data as for preceding; N24.1063, E121.1981; alt. 2294 m; 2 May 2011; Yu-Da Chuan leg.; CMNH544824; CMNH • 1 ♂; Ren-Ai, Yuan-Feng; N24.1181, E121.2374; alt. 2761 m; 19 July 2011; HuiSheng Fang leg.; CMNH • 1 ♂; same data as for preceding; CMNH454334; CMNH • 1 ♀; same data as for preceding; CMNH416575; NRC • 1 ♀; Ren-Ai, Jui-Yen; N24.1612, E121.2869; alt. 2986 m; 22 March 2011; Han-Tzu Hsu 001-000005; CMNH • 1 ♂; Ren-Ai, HeHuan-Shan Main Peak; N24.1425, E121.2726; alt. 3398 m; 6 June 2011; Wen-Chen Chu leg.; CMNH450599; CMNH • 1 ♂; same locality; N24.1815, E121.2816; alt. 3424 m; 29 June 2011; Wen-Chen Chu leg.; CMNH451385; CMNH.


**Redescription**


**Male** (Figure 7A). Body length is 12.8–14.2 mm, wing length is 15.7–17.5 mm, length of antenna is 3.5–4.1 mm. General body coloration is yellow.

Head. Yellow with narrow median black line on vertex and occiput. Rostrum ventrally brown, nasus conspicuous. Palpus brown. Antenna 13-segmented, if bent backward reaching before base of the wing. Scape and pedicel and first flagellar segment light yellow, flagellar segments 2–6 with bicolor appearance, remaining brown. Each flagellomere except the first darkened, without distinct enlargement. Apical flagellomere small. Long verticils equal to length of corresponding flagellomeres.

Thorax (Figure 7B). Overall brownish ochreous. Pronotum brownish ochreous anteriorly, posterior half ochreous yellow. Prescutum and presutural scutum yellowish ochreous, with four distinct ochreous brown stripes. Central stripes separated by yellowish, interspaces yellow. Postsutural scutum without median line; two ochreous brown spots on each lobe. Scutellum and mediotergite yellowish ochreous with dark median line. Pleura yellow. Coxae and trochanters yellow; femora yellow at base, becoming brown towards darkened end; tibiae and tarsal segments dark brown; claw with tooth. Wing clear yellowish brown, variegated with light brown clouds (Figure 7A). Costal and subcostal area yellow, stigma dark brown. Vein m-cu joining dm close to fork of M_3+4_, M_1+2_ about 0.2× as long as length of cell m_1_. Halter yellow, knob basally brown.

*Abdomen*. Abdominal tergites 1–5 yellow, with narrow median and lateral stripes, remaining tergites brownish black. Sternites 1–4 yellow, remaining brownish black.

*Hypopygium* (Figure 8). Male hypopygium with tergite 9 large, fused with sternite 9 on basal half (Figure 8A). Tergite 9 with dorsal portion undivided, medially with broad pale area more visible on dry specimen (Figure 8B,C); posterolateral angle produced caudad and slightly ventrad into flattened, elongated, blade-shaped arm, narrowed, outwardly forming oblique tips with single minute spur near tip. Tip of arm reaches far beyond tips of inner gonostylus. Posterior margin of dorsal portion squarely truncated, with a small V-shaped median notch subtended on either side by a round projection. Ventral portion developed into two flattened plates, which are widened posteriorly with black inner margin. Gonocoxite fused with sternite 9, nearly rectangular in outline, without projection (Figure 8A). Outer gonostylus small, narrow, filiform (Figure 8E). Inner gonostylus in the shape of elongated plate, narrowed in the middle (Figure 8F); upper beak short and darkened dorsal crest round with obtuse point; lower beak indistinct. Appendage of sternite 9 with ventral lobe long, blackened at base, covered long setae (Figure 8D); dorsal lobe long and massive, narrow at base, distal part swollen with round projection in the middle of inner side (Figure 8A,D). Adminiculum relatively narrow in ventral view, medially protruded with preapical round incision (Figure 8D,G). Sperm pump with central vesicle swollen (Figure 8H,I). Compressor apodeme flattened, with median rod (Figure 8H). Posterior immovable apodeme narrow, anterior immovable apodeme round. Aedeagus about 2× as long as sperm pump, basally brown, becoming light brown towards apex (Figure 8I).

**Female** (Figure 7C). Body length is 16.4–17.2 mm, wing length is 14.4–18.8 mm, length of antenna is 2.5–2.7 mm. Generally similar to male in body coloration.

*Ovipositor* (Figure 9). Tergite 10 shiny dark brown. Cercus brown, short, about 1.5X as short as tergite 10, straight, with tip rounded; ventral margin smooth, without visible serration (Figure 9A). Hypovalva extending base of cercus (Figure 9A). Sternite 8 with hypovalva developed, blade-shaped (Figure 9B); median incision between hypovalvae basally with indistinctly serrated edge and covered with setae. Sternite 8 with flattened lobe from the middle to the lateral angles. Sternite 9 shovel-shaped, round at posterior margin; the surface covered with short filaments (Figure 9C). Furca membranous, posteriorly flattened (Figure 9C). Spermatheca nearly spherical, broadened at base (Figure 9D).
Figure 8Male terminalia of *Tipula* (*Vestiplex*) *foliacea*. (**A**). Hypopygium, lateral view. (**B**). Hypopygium, dorsal view. (**C**) Tergite 9, dorsal view. (**D**) Sternite 9, tergite 9 and gonostyli removed, ventral view. (**E**). Outer gonostylus. (**F**). Left inner gonostylus, lateral view. (**G**). Adminiculum, lateral view. (**H**). Semen pump, dorsal view. (**I**). Semen pump and intromittent organ, lateral view. Abbreviation: dl—dorsal lobe of A9s. Scale bars: a (**A**,**C**–**I**), b (**B**) = 0.5 mm.
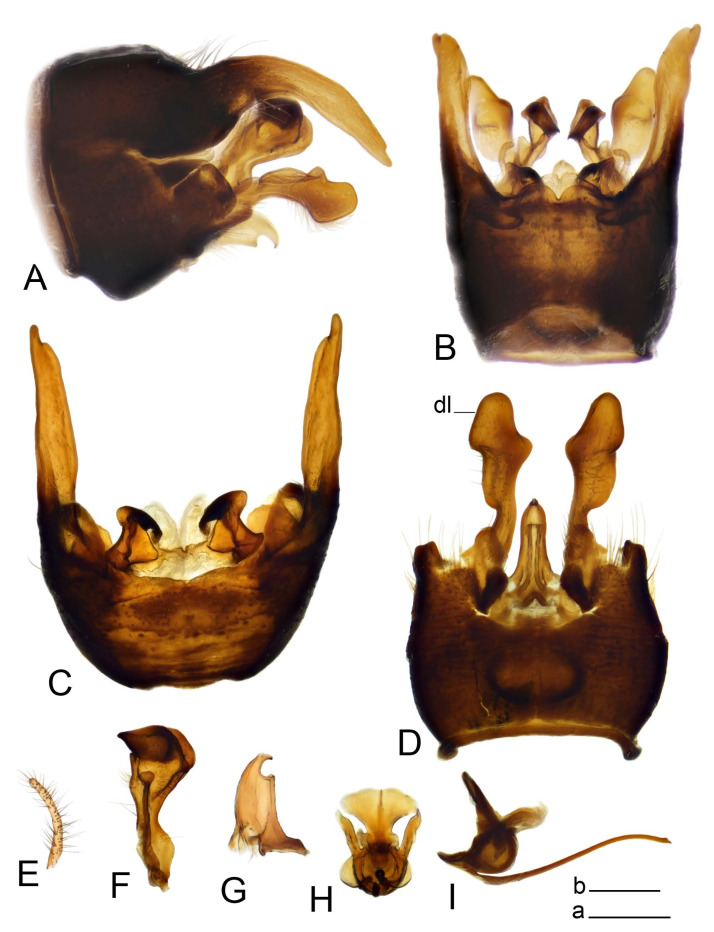

Figure 9Female terminalia of *Tipula* (*Vestiplex*) *foliacea*. (**A**). Ovipositor, left lateral view. (**B**). Sternite 8 with hypovalvae, ventral view. (**C**). Sternite 9 and furca, dorsal view. (**D**). Spermatheca. Abbreviations: f—furca; s9—sternite 9; t10—tergite 10. Scale bars: a (**A**), b (**B**,**C**) = 0.5 mm; c (**D**) = 0.25 mm.
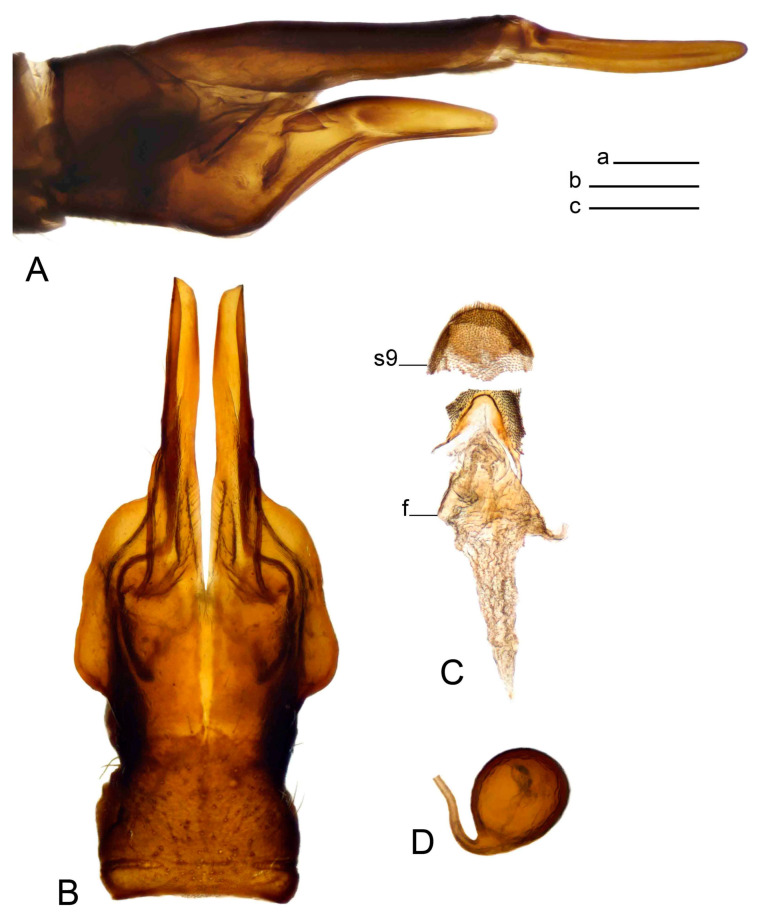



**Distribution.** Taiwan (Figure 2A).

**Elevation range in Taiwan.** Adults were collected at altitudes ranging from 1600 m to 3400 m.

**Period of activity.** Adults are active from April through to September.

**Remarks.***Tipula* (*V.*) *foliacea* was described by Alexander [36] based on a single male specimen collected by K. Takeuchi in 1922 from Arisan (嘉義阿里山 2286 m). The female asssociation of this species is based on a current DNA gene sequence study. The nearest ally of *T.* (*V.*) *foliacea* is undoubtedly *T.* (*V.*) *arisanensis* based on its morphological features. The DNA gene sequence data also support this. The latter species is a trifle smaller and with a very differently constructed male hypopygium; the tergal arms of segment 9 are much smaller and do not extend beyond the tips of the inner gonostylus.


***Tipula* (*Vestiplex*) *bicornigera* species group**


The *bicornigera* group was proposed by Starkevich et al. [37] with two species added later [38]. Three species are recognized in Taiwan, including *T.* (*V.*) *bicornigera*, *T.* (*V.*) *graciliana* **sp. nov.**, and *T.* (*V.*) *survilai* **sp. nov.**

Males of the *bicornigera* group can be recognized by tergite 9 which is completely divided into two parts at the midline by a pale membrane, with ventral portion hypertrophied and occupying from half to nearly the entire tergite area, forking into two flattened brown plates; inner and posterior margin of each plate blackened, and microscopically roughened; dorsal portion of tergite 9 narrow in midlength or reduced in size, posteriorly with one of two pair of lobes directed caudad and provided with setae. They can be also characterized by gonocoxite dorsally produced into a black spine or with small acute point; sternite 9 with dorsal lobe of A9s usually reduced into a small triangular-, rod-, wedge-shaped or narrow and distinct sclerite.

Females are illustrated for *T.* (*V.*) *butvilai* Starkevich, Saldaitis and Men, 2019 ([37]: Figures 9–13) and the Taiwanese species described herein.

Cercus with slightly visible serration on ventral and dorsal margins; hypovalvae short, blackened, plate-shaped, posteriorly with additional incisions.

***Tipula* (*Vestiplex*) *bicornigera*** Alexander

Figure 10, Figure 11, Figure 12 and Figure 13; Figure 2B

*Tipula* (*Vestiplex*) *bicornigera* Alexander, [39]: 102;

*Tipula* (*Vestiplex*) *bicornigera*: Alexander and Alexander, [34]: 62;

*Tipula* (*Vestiplex*) *takahashiana* Alexander, [34]: 101 **syn. nov.**;

*Tipula* (*Vestiplex*) *takahashiana*: Alexander and Alexander, [34]: 65.


**Diagnosis**


*Tipula* (*V.*) *bicornigera* can be recognized by its relatively small body size, brownish ochreous, golden yellow pollinose thorax and femora with broad yellow preapical ring. Male can be distinguished by the gonocoxite having apical outgrowth, which is distinctly dilated at base and terminating into a black spine with a slightly curved acute tip and elongated antenna, reaching the base of abdomen if bent backward. Females can be recognized by straight cercus with ventral and dorsal margins slightly serrated and short, toothed hypovalva.

**Type of material examined.** *Tipula* (*Vestiplex*) *bicornigera* Alexander: **TAIWAN** • Holotype ♂; Formosa, Oiwake, Noko-gun; alt. 7570 ft.; 12 August 1936; Takahashi leg.; antenna, leg, wing, and genitalia mounted on slide; USNM.

*Tipula* (*Vestiplex*) *takahashiana* Alexander: **TAIWAN** • Holotype ♀; Formosa, Hattsukan; alt. 9000 ft.; 23 August 1936; Takahashi leg.; antenna, leg and wing on slide mounted; USNM.

**Additional material examined. TAIWAN**—**Hualien Co.** • 1 ♂; GuanYu, Youth Activities Center; N24.1879, E121.3377; alt. 2355 m; 3 May 2010; Chen Young leg.; CMNH543754; CMNH • 1 ♀; KuanYuan; [N24.1850667, E121.3421278]; alt. 2734 m; 18 May 2007; Chen Young leg.; CMNH544040; CMNH • 1 ♂; SiouLin, E. end of JinMa Tunnel; N24.1773, E121.3727; alt. 2508 m; 20 September 2011; Chen Young leg.; 0808-0000851; CMNH • 1 ♂; same data as for preceding; N24.177, E121.373; 20 September 2011; Wen-Chen Chu leg.; CO1 Barcode TIPTW1066-12; CMNH576534; CMNH • 1 ♀; same data as for preceding; N24.1794, E121.3647; alt. 2456 m; 5 October 2010; Chen Young leg.; CMNH449454; NRC • 5 ♂♂, 8 ♀♀; same data as for preceding; CMNH544142; CMNH454389; CMNH455973; CMNH569045; CMNH544140; CMNH544137; CMNH544135; CMNH443261; CMNH432254; CMNH568930; CMNH569095; CMNH401349; CMNH544134; dissected; CMNH—**Yilan Co.** • 1 ♂; Cilan Forest Conservation Station; N24.5879, E121.4113; alt. 1637 m; Chen Young leg.; CMNH450820; NRC • 2 ♂♂; same data as for preceding; CMNH543675; CMNH454183; dissected; CMNH • 2 ♂♂, 1 ♀; same data as for preceding; 26 September 2010; CMNH544145; CMNH544150; CMNH544147; CMNH—**Kaohsiung Co.** • 1 ♂; Taoyuan Township, Tenchin; 23-04-03N, 120-45-13E; alt. 1550 m; 30 March 2009; Chen Young leg.; CMNH543733; CMNH—**Nantou Co.** • 3 ♂♂; Mei Fang, NE Wushe; N24.1, E121.16667; 15 March 1988; J. Rawlins, C. Young leg.; CMNH544038; CMNH • 1 ♀; Ren-Ai, Jui-Yen; N24.1063, E121.1981; alt. 2294 m; 2 August 2011; Yu-Da Chuan leg.; 1420-000080; CMNH—**Taichung Co.** • 8 ♂♂, 4 ♀♀; Bi-LuSi, Taiwan Forest Res. Inst.; N24.2226, E121.3050; alt. 2149 m; 12 May 2010; Chen Young leg.; CMNH543576; CMNH543581; CMNH452845; CMNH543583; CMNH411100; dissected; CO1 Barcode SATIP1096-10; CMNH543574; CMNH543584; CMNH543580; CMNH543577; CMNH543564; CO1 Barcode SATIP1107-10; CMNH543585; CMNH543582; CMNH • 3♂♂, 2 ♀♀; same data as for preceding; 4 October 2010; CMNH451363; CMNH409494; dissected; CMNH443381; dissected; CMNH455143; dissected; CMNH452108; NRC • 5 ♂♂, 14 ♀♀; same data as for preceding; CMNH411265; CMNH544185; CMNH544160; CMNH569527; CMNH544187; CMNH455236; CMNH449723; CMNH568931; CMNH451295; CMNH455960; CMNH401214; CO1 Barcode TIPTW568-10; CMNH544186; CMNH444898; CMNH450249; CO1 Barcode TIPTW543-10; CMNH544161; CMNH409684; CMNH544184; CO1 Barcode TIPTW547-10; CMNH544165; CMNH544170; CMNH • 1 ♂, 1 ♀; same data as for preceding; 22 March 2011; TIPTW817-11; CMNH544570; CMNH544569; CMNH • 1 ♂; LiShan Rest area at light; N24.2551, E121.2507; alt. 1929 m; 23 March 2011; Chen Young leg.; CMNH544557; CMNH • 4 ♀♀; Pilushi; [N24.24750, E121.30720]; alt. 2200 m; 1988.05.22-23; R. Davidson, J. Rawlins, C. Young leg.; CMNH544031; CMNH544030; CMNH—**Taipei Co.** • 1 ♂; Sungsham; 10 June 1984; 25.059788°N, 121.55727°E; Chen Young; CMNH544037; CMNH—**TaiTung Co.** • 1 ♂, 2 ♀♀; Tientzi; N23.2776625, E120.9149205; 28 August 1984; W. J. Wu leg.; CMNH569209; CMNH446779; CMNH447403; CMNH—**Taoyuan Co.** • 1 ♂; Fushing Township, Lalashan Cottage; N23.0848; E.120.7877; alt. 1806 m; 28 March 2012; Yu-Da Chuan leg.; 2025-000136; CMNH • 1 ♂; Fushing Township, Lalashan Ecology Room; N24.7094; E121.4347; alt. 1634 m; 5 April 2012; Han-Tzu Hsu leg.; CMNH449364; CMNH • 2 ♂♂; Fushing Township, Lalashan Cottage; N24.7094; E.121.4332; alt. 1581 m; 6 April 2012; Han-Tzu Hsu leg.; 2025-000073; 2025-000082; CMNH • 4 ♂♂; same data as for preceding; Wen-Chen Chu leg.; CMNH412451; 2025-000075; 2025-000068; 2025-000077; CMNH.


**Redescription**


**Male** (Figure 10A). Body length is 10.5–12.5 mm, wing length is 12.7–14.0 mm, length of antenna is 4.8–5.8 mm. General body coloration is yellow.

Head. Yellow, vertex and occiput with dark vitta. Rostrum dorsally yellow, ventrally yellowish brown, nasus short. Palpus brown. Antenna 13-segmented, if bent backward reaching base of abdomen. Scape and pedicel yellow, first flagellar segment basally yellowish brown, distally brown, remaining segments dark brown. Each flagellomere, except first, with light basal enlargement. Apical flagellomere small, distinctly shorter than preceding flagellomere. Long verticils shorter than corresponding flagellomeres.
Figure 10Habitus of *Tipula* (*Vestiplex*) *bicornigera*. (**A**). Male, lateral view. (**B**). Male thorax, dorsal view. (**C**). Female, lateral view. Scale bars: a (**A**,**C**) = 5 mm, b (**B**) = 1 mm.
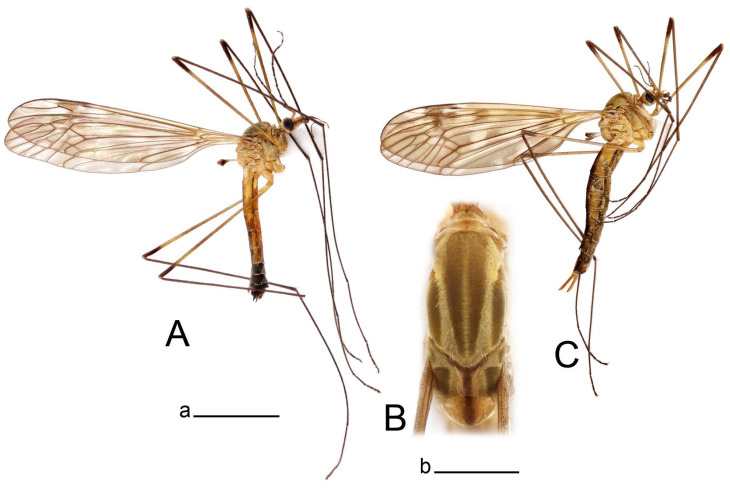



Thorax (Figure 10B). Pronotum yellowish brown, with dark median vitta. Prescutum and presutural with four brownish ochreous stripes. Central stripes confluent at prescutum and narrowly divided on posterior two-thirds by a narrow pale median line; basal two-thirds darker, anteriorly narrowly bordered by brown. Interspaces with golden yellow pollinose. Postsutural scutum yellowish, with brown median line. Each scutal lobe with two ochreous spots. Scutellum yellow, with brown median line. Mediotergite yellow with pale median line. Pleura yellow. Legs with coxae and trochanters yellow. Femora brownish yellow, tip brownish black, preceeded by broad yellow subterminal ring. Rest of segments brownish black. Claw with tooth. Wing clear pale brown, tinged with brown (Figure 10A). Stigma area dark brown. Vein m-cu joining dm at fork of M_3+4_, M_1+2_ about 0.3× as long as length of cell m_1_. Halter yellow, knob brown.

*Abdomen*. Abdominal tergites 1–5 yellow, with broad brown median and narrow lateral line. Sternites 1–5 yellow. Remaining abdominal segments black.

*Hypopygium* (Figure 11 and Figure 12). Male hypopygium with tergite 9 fused with sternite 9 basally (Figure 11A). Tergite 9 completely divided at midline by pale membrane, posteriorly with V-shaped notch (Figure 11B–E). Dorsal portion reduced, narrow in midlength, posterior margin with two pair of lobes directed caudad. Posterolateral lobes narrowed, posteromedian lobes flattened twice longer Ventral portion hypertrophied and occupies nearly the entire tergite area, forking into flattened brown plates. Each plate serrulated on inner margin along midline, with surrounding surface densely covered with black dots, posterior margin terminating into black, nearly triangular process. Gonocoxite separated from sternite 9 by suture, with apical outgrowth basally distinctly dilated, produced into a black spine with slightly curved and acute tip (Figure 11A,F). Outer gonostylus distally oval (Figure 11G). Inner gonostylus terminating into obtuse beak, lower beak nearly oval (Figure 11H,I). Dorsal margin with medial portion bent inwardly, outer basal lobe slightly flattened, with apical margin darkened, obtuse, slightly bent inwards. Sternite 9 with ventral lobe of A9s nearly triangular covered with setae (Figure 11K,J). Dorsal lobe of A9s fused with ventral lobe, grip-shaped. Adminiculum triangular in ventral view, basally dilated with lateral parts raised (Figure 11J–L). Medial portion protruded in basal half, apex slightly curved, curve preeceeded by broad incision. Gonocoxal fragment reduced. Sperm pump with central vesicle swollen (Figure 12A). Compressor apodeme with broad median incision (Figure 12B). Posterior immovable apodeme narrow, anterior immovable apodeme rounded. Aedeagus about 3.7× as long as sperm pump (Figure 12A).
Figure 11Male terminalia of *Tipula* (*Vestiplex*) *bicornigera*. (**A**). Hypopygium, lateral view. (**B**). Hypopygium, dorsal view. (**C**) Tergite 9, dorsal view. (**D**). Tergite 9, same sclerite pressed by cover glass, pml directed caudad. (**E**) As previous, pml directed cephalad. (**F**). Gonocoxite. (**G**). Outer gonostylus. (**H**). Left inner gonostylus, lateral view. (**I**). Right inner gonostylus, mesal view. (**J**). Sternite 9 with adminiculum and ventral lobes of A9s, ventral view. (**K**). Adminiculum and ventral lobes of A9s. (**L**). Adminiculum, lateral view. Abbreviations: dl—dorsal lobe of A9s; dp—dorsal portion of tergite 9; pll—posterolateral lobe of tergite 9; pml—posteromedian lobe of tergite 9; vl—ventral lobe of A9s; vp—ventral portion of tergite 9. Scale bars: a (**A**,**B**), b (**C**–**J**) = 0.5 mm; c (**K**,**L**) = 0.25 mm.
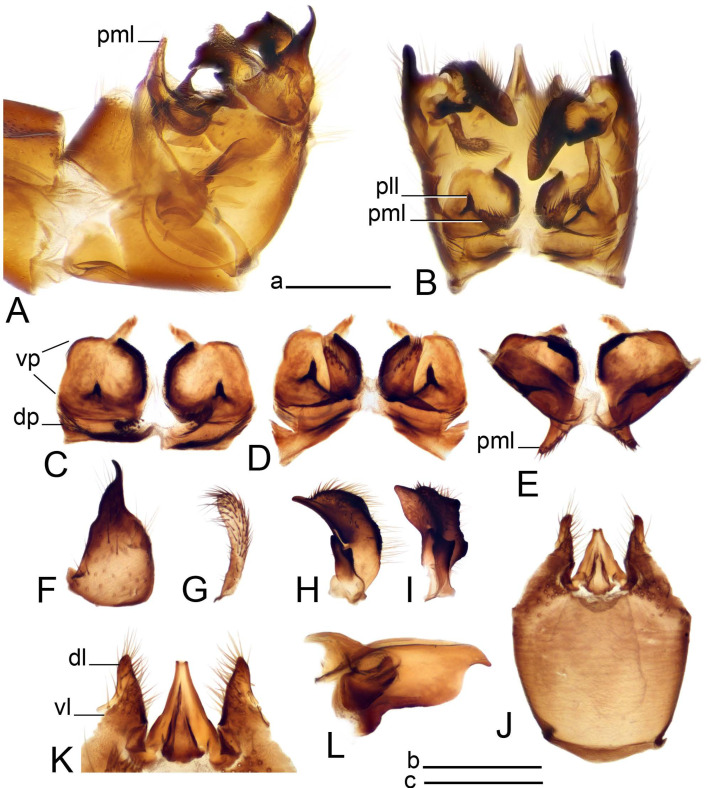

Figure 12Male terminalia of *Tipula* (*Vestiplex*) *bicornigera*. (**A**). Semen pump and intromittent organ, lateral view. (**B**). Semen pump, dorsal view. Scale bar: a (**A**,**B**) = 0.5 mm.
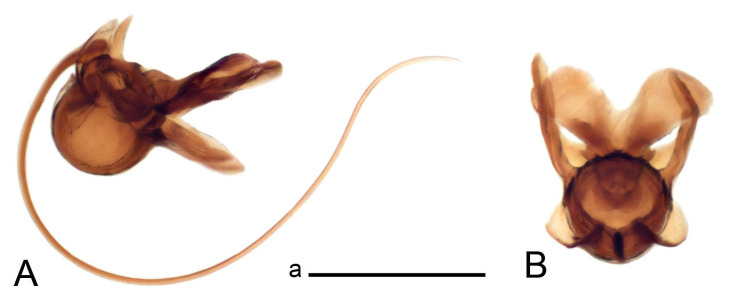



**Female** (Figure 10C). Body length is 14.0–16.0 mm, wing length is 13.0–15.5 mm, length of antenna is 2.0–2.3 mm. Generally similar to male in body coloration.

*Ovipositor* (Figure 13). Tergite 10 shiny brown. Cercus straight, brown, same length as tergite 10, tip narrowed (Figure 13A). Ventral and dorsal margins with slightly visible serration. Sternite 8 with hypovalva heavily blackened, short, plate-shaped (Figure 13B,C). Posterior margin of hypovalva medially with deep incision forming inner acute and outer obtuse teeth. Sternite 9 with posterior half shovel-shaped, surface covered with short filaments, with broad median groove; anterior parts nearly straight (Figure 13D). Furca posteriorly flattened, membranouos, anterior part long and narrow (Figure 13E). Spermatheca spherical (Figure 13F).

**Distribution.** Taiwan (Figure 2B).

**Elevation range in Taiwan.** Adults were collected at altitudes ranging from 1900 m to 23,400 m.

**Period of activity.** Adults are active throughout the year except in November and December.

**Remarks.** *Tipula bicornigera* was described by Alexander [39] based on a single male specimen collected by Takahashi in 1936 from Oiwake, Nako-gun (翠峰, 南投能高 2307 m). The males were indentified via a comparison with a holotype and genitalia mounted on a slide by C. P. Alexander. The female asssociation of this species in this study is based on ca urrent DNA gene sequence study.

*Tipula takahashiana* was described in the same paper based on a single female specimen collected by Takahashii in 1936 from Hattsukan (Patungkuan). The holotype was compared with the females of *Tipula bicornigera* previously assigned to males using DNA. All specimens were characterized by the identical shape of cerci and hypovalvae and the synonymy *T.* (*V.*) *takahashina* = *T.* (*V.*) *bicornigera*
**syn. nov.** was established.
Figure 13Female terminalia of *Tipula* (*Vestiplex*) *bicornigera*. (**A**). Ovipositor, left lateral view. (**B**). Sternite 8 with hypovalvae, ventral view. (**C**). Hypovalvae, pressed by cover glass, ventral view. (**D**). Sternite 9, dorsal view (**E**). Furca, dorsal view. (**F**). Spermatheca. Scale bars: a (**A**), b (**B**–**E**) = 0.5 mm.
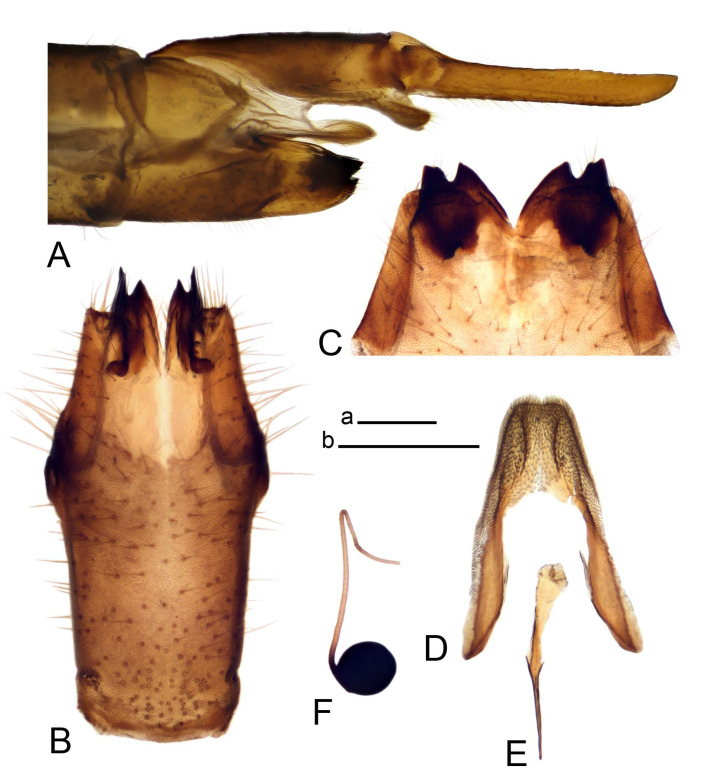

insects-14-00616-sch002_Scheme 2Scheme 2Distribution maps of *Tipula* (*Vestiplex*) crane flies in Taiwan. (**A**). *T.* (*V.*) *foliacea*. (**B**). *T.* (*V.*) *bicornigera*. Yellow circles represent paratypes.
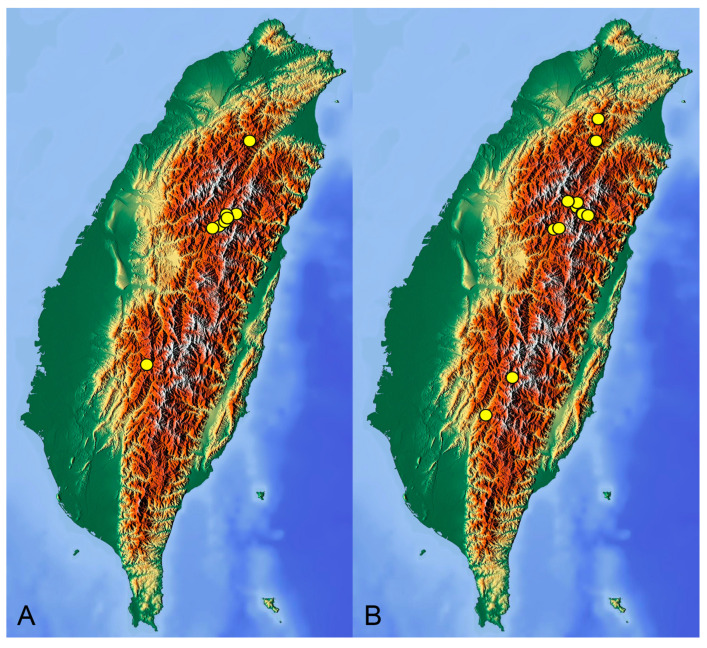




***Tipula* (*Vestiplex*) *graciliana* Starkevich and Young sp. nov.**


Figure 14, Figure 15, Figure 16 and Figure 17; Figure 3A


**Differential diagnosis**


*Tipula* (*V.*) *graciliana* **sp. nov.** can be recognized by long antenna almost reaching the end of the abdomen, slender flagellomeres, a brown thorax, and an elongated abdomen having a striped appearance. Males can be also recognized by their gonocoxite apically produced into a black spine and Tergite 9 with the ventral portion hypertrophied, forking into flattened, brown, serrulated plates. The female is brachypterous, with wings short and narrow, and a cercus with tip rounded and indisnctly flattened, as well as slightly up-turned, ventral and dorsal margins with visible serration. Hypovalva is heavily blackened, short, and plate-shaped.


**Type of material**



**Holotype**


**TAIWAN** • ♂; Nantou Co.; Ren-Ai HeHuanShan Yuanfeng; N24.1231, E121.2711; alt. 3088 m; 21 March 2011; Chen Young leg.; CO1 Barcode TIPTW782-11; CMNH 544535; CMNH.


**Paratypes**


**TAIWAN**—**Nantou Co.** • 1 ♀; Yuanfeng; [N24.12083, E121.24333]; 13 March 2007; C.S. Lin and W. T. Yang leg.; TIPTW826-11; CMNH 544579; dissected; CMNH • 1 ♂; Ren-Ai HeHuanShan Yuanfeng; N24.1181, E121.2374; alt. 2761 m; 15 February 2012; Han-Tzu Hsu leg.; CMNH410674; dissected; NRC • 1 ♂; same locality; N24.1231, E121.2711; alt. 3088 m; 21 March 2011; Chen Young leg.; CMNH452102; NRC • 4 ♂♂; Ren-Ai He-Huan-Shan LuoYing Inn; N24.1730, E121.3016; alt. 2830 m; 22 March 2011; Chen Young leg.; CMNH452834; dissected; CO1 Barcode TIPTW781-11; CMNH 544534; CO1 Barcode TIPTW780-11; CMNH 544533; CMNH • 1 ♀; Ren-Ai HeHuan-Shan Main Peak; N24.1425, E121.2726; alt. 3398 m; 3 June 2011; HuiSheng Fang leg.; 0808-000033; CMNH • 1 ♂; Ren-Ai Yuan-Feng; N24.1181, E121.2374; alt. 2761 m; 19 July 2011; Yu-Da Chuan leg.; 0808-000167; CMNH • 3 ♂♂; same data as for preceding; 18 January 2012; HuiSheng Fang leg.; 0808-001370; 0808-001366; 0808-001364; CMNH • 1 ♂; same data as for preceding; 19 January 2012; CMNH449405; CMNH • 1 ♂, 1 ♀; same data as for preceding; 15 February 2012; Wen-Chen Chu leg.; 0808-001336; CMNH569249; CMNH • 1 ♂; same data as for preceding; 6 June 2012; HuiSheng Fang leg.; CMNH395812; CMNH—**Taichung Co.** • 1 ♂; LiShan Rest area at light; N24.2551, E121.2507; alt. 1929 m; 23 March 2011; Chen Young leg.; CMNH.


**Description**


**Male** (Figure 14A). Body length is 14.7–17.4 mm, wing length is 17.6–21.9 mm, length of antenna is 10.4–12.5 mm. General body coloration is yellowish brown.
Figure 14Habitus of *Tipula* (*Vestiplex*) *graciliana*
**sp. nov.** (**A**). Male, holotype, lateral view. (**B**). Male antennae, paratype. (**C**). Male thorax, holotype, dorsal view. (**D**). Female, paratype, lateral view. Scale bars: a (**A**,**D**) = 5 mm, b (**B**) = 2 mm, c (**C**) = 1 mm.
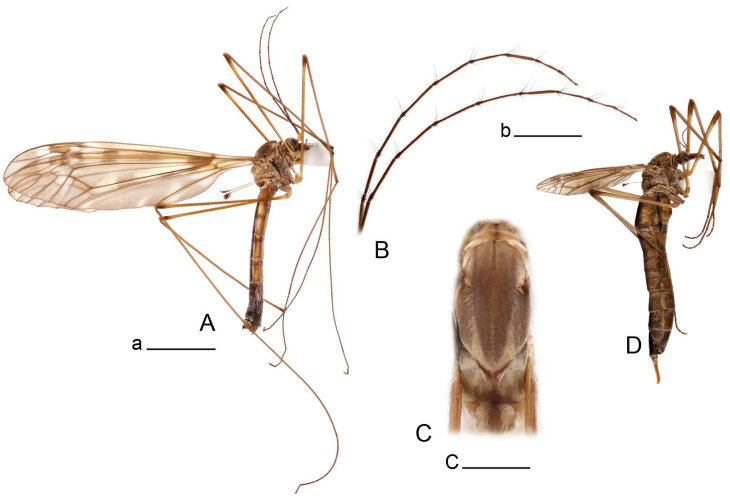



Head. Yellow, vertex and occiput with narrow, brown median line. Rostrum brown, dorsally yellow including distinct nasus. Palpus brown. Antenna 13-segmented (Figure 14B), if bent backward almost reaching the end of abdomen. Scape and pedicel yellow; the base first flagellar segment narrowly yellow, remaining segments brown. Each flagellomere, except first, with light basal enlargement. Flagellomeres visibly elongated, e.g., flagellar segments 2–4 about 5.7–6.8 times as long as their width in the middle (Figure 14B). Apical flagellomere small, distinctly shorter than preceding flagellomere. Long verticils about 2–2.5 as short as corresponding flagellomeres.

*Thorax* (Figure 14C). Overall brown. Pronotum brown. Prescutum and presutural scutum with four brown stripes, bordered by darker brown. Central stripes confluent at base and anteriorly. Interspaces light brown. Postsutural scutum light brown, with brown median line. Each scutal lobe with two brown spots bordered by darker brown. Scutellum brown, with median line. Mediotergite brown, thinly dusted with gray, with vestige of median vitta. Pleura brown. Legs with coxae and trochanters yellow. Femora yellowish brown, apically yellow, tip blackened. Remaining segments yellowish brown. Claw without tooth. Wing clear pale brown, tinged with brown (Figure 14A). Costal and subcostal area yellowish. Brown postarcular spot in cells br and bm and stigma area. Vein m-cu joining dm at fork of M_3+4_, M_1+2_ about 0.15× as long as length of cell m_1_. Halter yellow, knob brown.

*Abdomen*. Tergites 1–4 dorsally dark brown, laterally yellow with dark brown band in anterior portion, creating a striped appearance. Sternites 1–4 yellow, with brown line. Remaining segments brownish black.

*Hypopygium* (Figure 15 and Figure 16). Male hypopygium with tergite 9 fused with sternite 9 basally (Figure 15A). Tergite 9 completely divided at midline by pale membrane, posteriorly with V-shaped notch (Figure 15B–D). Dorsal portion reduced, narrow in midlength, posterior margin with two pair of lobes directed caudad. Posteromedian lobes flattened, nearly oval, covered with setae, located on either side of midline; posterolateral lobes short and narrow. Ventral portion hypertrophied and occupies nearly the entire tergite area, forking into flattened brown plates. Each plate blackened and serrulated on inner margin along midline, terminating into black rounded lobe. Gonocoxite separated from sternite 9 by suture, apically produced into black spine (Figure 15A,E). Outer gonostylus finger-shaped (Figure 15F). Inner gonostylus a blackened plate terminating into short beak, lower beak nearly oval (Figure 15G). Dorsal margin without lobes, outer basal lobe flattened, with apical margin terminating into acute, blackened tooth. Sternite 9 with ventral lobe of A9s nearly triangular, covered with setae (Figure 15I). Dorsal lobe of A9s small, fused with ventral lobe, nearly wedge-shaped. Adminiculum triangular in ventral view, basally dilated with lateral parts raised (Figure 15I). Medial portion protruded, apex slightly curved, curve preeceeded by round incision (Figure 15H). Gonocoxal fragment reduced. Sperm pump with central vesicle swollen (Figure 16A). Compressor apodeme with broad median incision (Figure 16B). Posterior immovable apodeme narrow, anterior immovable apodeme rounded. Aedeagus about 3.5× as long as sperm pump, basally brownish yellow, becoming yellow towards apex (Figure 16A).

**Female** (Figure 14D). Body length is 19.1–19.5 mm, wing length is 6.2–8.9 mm, length of antenna is 1.9–2.0 mm. General body coloration is dark brown. Female is brachypterous, with wings short and narrow with visible veins.

Ovipositor (Figure 13). Tergite 10 shiny brown. Cercus brown, longer than tergite 10, tip rounded and indisnctly flattened, slightly up-turned (Figure 17A). Ventral and dorsal margins with visible serration. Sternite 8 with hypovalva heavily blackened, short, plate-shaped (Figure 17B). Posterior margin of hypovalva medially with incision forming inner protrusion terminating into acute point and outer triangular teeth. Sternite 9 with posterior half flattened, nearly rectangular, surface covered with short filaments, with broad median groove; anterior parts bent at basal half (Figure 17C). Furca posteriorly flattened, membranous, anterior part narrowed (Figure 17D). Spermatheca oval (Figure 17E).

**Etymology.** The species epithet is derived from Latin “gracilia” refering to the elongated slender antennae and relatively slender abdomen.

**Distribution.** Taiwan (Figure 3A).

**Elevation range in Taiwan.** Adults were collected at altitudes ranging from 1900 m to 4000 m.

**Period of activity.** Adults are active from January through to July.
Figure 15Male terminalia of *Tipula* (*Vestiplex*) *graciliana*
**sp. nov.** (**A**). Hypopygium, lateral view. (**B**). Hypopygium, dorsal view. (**C**) Tergite 9, dorsal view. (**D**). Tergite 9, caudal view. (**E**) Gonocoxite (**F**). Outer gonostylus (**G**). Left inner gonostylus, lateral view. (**H**). Adminiculum, lateral view. (**I**). Sternite 9 with adminiculum and ventral lobes of A9s, ventral view. Abbreviations: pll—posterolateral lobe of tergite 9; pml—posteromedian lobe of tergite 9. Scale bars: a (**A**,**B**), b (**C**–**I**) = 0.5 mm.
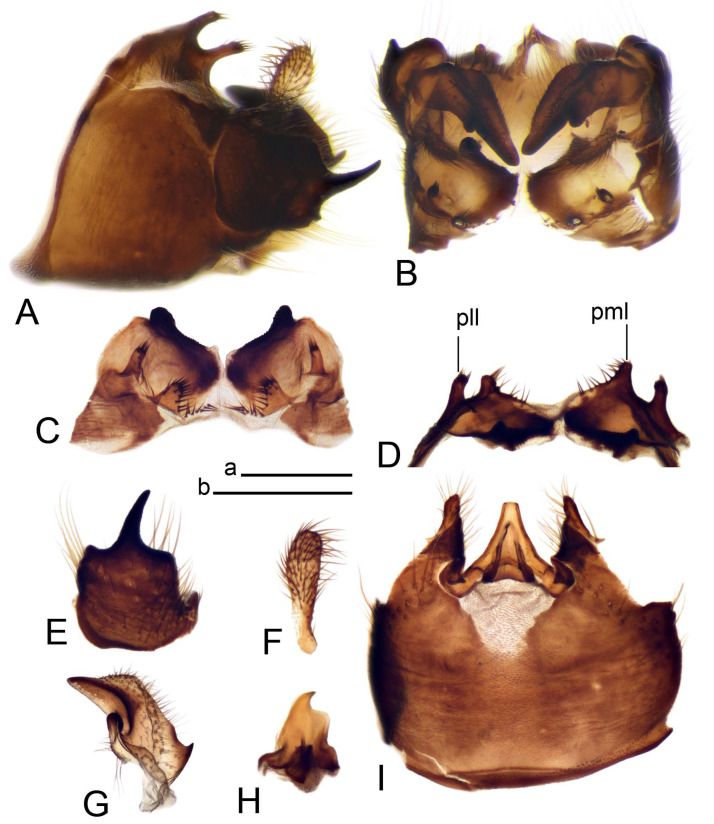

Figure 16Male terminalia of *Tipula* (*Vestiplex*) *graciliana*
**sp. nov.** (**A**). Semen pump and intromittent organ, lateral view. (**B**). Semen pump, dorsal view. Scale bar: a (**A**,**B**) = 0.5 mm.
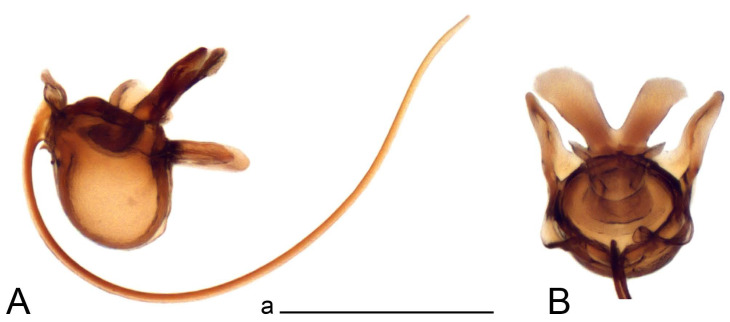

Figure 17Female terminalia of *Tipula* (*Vestiplex*) *graciliana*
**sp. nov.** (**A**). Ovipositor, sternites removed, left lateral view. (**B**). Posterior part of sternite 8 with hypovalvae, ventral view. (**C**). Sternite 9, dorsal view (**D**). Furca, dorsal view. (**E**). Spermatheca. Scale bars: a (**A**), b (**B**–**E**) = 0.5 mm.
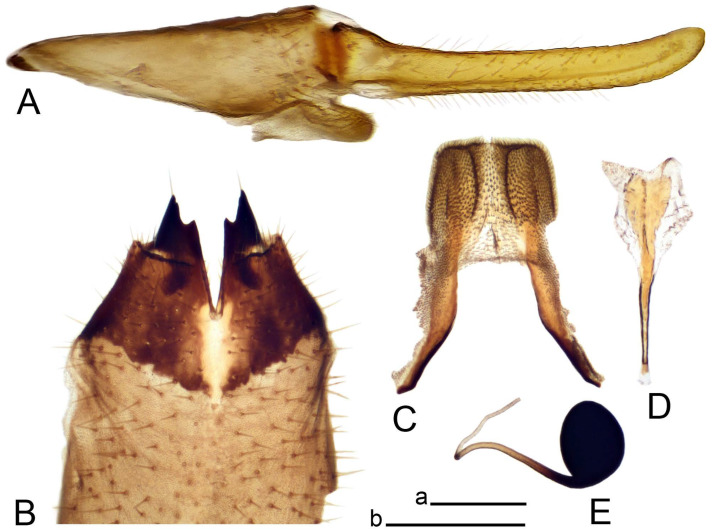




***Tipula* (*Vestiplex*) *survilai* Starkevich and Young sp. nov.**


Figure 18, Figure 19, Figure 20 and Figure 21; Figure 3B


**Differentional diagnosis**


*Tipula* (*V.*) *survilai* **sp. nov**. can be recognized by its elongated antenna reaching beyond the base of abdomen if bent backward, greenish yellow thorax, femora with yellow preapical ring, gonocoxite with black, slightly curved spine, and tergite 9 having a ventral portion occupying nearly the entire tergite 9, forking into flattened, brown, serrulated plates. The female is brachypterous, with wings short and narrow; cercus nearly straight, tip rounded and slightly up-turned, ventral and dorsal margins with visible serration; hypovalva heavily blackened and short, plate-shaped. Both *T.* (*V.*) *graciliana* **sp. nov**. and *T.* (*V.*) *survilai* **sp. nov**. are closely related to *T.* (*V.*) *bicornigera* in having two pairs of lobes on tergite 9, as well as a similar shape of outer gonostylus, adminiculum, and appendages of sternite 9. However, they can be easily separated by the gonocoxite bearing a nearly straight black spine, while *T.* (*V.*) *bicornigera* has a gonocoxite with an apical outgrowth dilated at the base and a curved tip.

In the meantime, *T.* (*V.*) *graciliana* **sp. nov**. can be separated by its antenna, almost reaching the end of the abdomen; elongated flagellomeres; a femur with a darkened tip, ventral plates of tergite 9 with a rounded tip; and an inner gonostylus with an acute basal lobe. *Tipula* (*V.*) *survilai* **sp. nov**. has a shorter antenna, reaching beyond the base of the abdomen; shorter flagellomeres; ventral plates of tergite 9 narrowing at the tip; and an inner gonostylus with an obtuse outer basal lobe.


**Type of material**



**Holotype**


**TAIWAN** • ♂; Nantou Co.; Ren-Ai Chi-Lai Mountain Entrance; N24.1388, E121.2874; alt. 3110 m; 10 August 2011; Wen-Chen Chu leg.; CMNH576565; CMNH.


**Paratypes**


**TAIWAN**—**Chiayi Co.** • 2 ♂♂; Alishan; [23°21′ N, 120°48′ E]; 12 April 1965; Yoshimoto; CMNH • 1 ♂; same data as for preceding; CMNH450077; dissected; NRC—**Hualien Co.** • 1 ♂; GuanYu Youth Activities Center; N24.1879, E121.3377; alt. 2355 m; 3 May 2010; Chen Young leg.; CMNH543768; dissected; CMNH • 2 ♂♂; SiouLin, E. end of JinMa Tunnel; N24.1794, E121.3647; alt. 2456 m; 13 May 2010; Chen Young leg.; CMNH543587; CMNH543588; CMNH—**Nantou Co.** • 2 ♂♂; Yuanfeng; [N24.12083, E121.24333]; IV/10---V/8/2007; Malaise trap (KCN); C. S. Lin & W. T. Yang leg.; CMNH544550; CO1 Barcode TIPTW802-11; CMNH 544555; CMNH • 2 ♂♂; Ren-Ai Chi-Lai Mountain Entrance; N24.1388, E121.2874; alt. 3110 m; 10 August 2011; Wen-Chen Chu leg.; CMNH576564; CMNH576545; CMNH • 1 ♂; Ren-Ai Endemic Species Res. Centr.; N24.1813, E121.2866; alt. 2939 m; 13 May 2010; Chen Young leg.; CMNH448962; dissected; CMNH • 1 ♂; same data as for preceding; 17 May 2011; 0808-000172; CMNH • 1 ♂; same data as for preceding; 10 August 2011; Yu-Da Chuan leg.; 0808-000320; CMNH • ♂; Ren-Ai HeHuan-Shan Main Peak; N24.1423, E121.2715; alt. 3416 m; 14 May 2010; Chen Young leg.; CO1 Barcode SATIP1043-10; CMNH543521; CMNH • ♂; same locality; N24.1425, E121.2726; alt. 3398 m; 3 June 2011; HuiSheng Fang leg.; 0808-000070; CMNH • ♂; same data as for preceding; 27 June 2011; Han-Tzu Hsu leg.; CMNH576572; CMNH576573; CMNH • ♂; same data as for preceding; CMNH451635; dissected; NRC • 2 ♂♂; Ren-Ai HeHuan-Shan North Peak; N24.1815, E121.2816; alt. 3424 m; 29 June 2011; Han-Tzu Hsu leg.; 0808-000277; CMNH—**Taichung Co.** • 1 ♂; Pilushi; [N24.24750, E121.30720]; alt. 2200 m; 1988.05.22-23; R. Davidson, J. Rawlins, C. Young leg.; CMNH544046; CMNH • 1 ♂; Highway 8 at 108 mark; N24.0035, E121.2916; alt. 2410 m; 13 May 2010; Chen Young leg.; CMNH543600; CMNH—**Taipei Co.** • 1 ♂; Keelung; [25°08′ N, 121°44′ E]; 6 October 1957; Yoshimoto leg.; CMNH416559; CMNH.


**Description**


**Male** (Figure 18A). Body length is 12.2–15.0 mm, wing length is 14.3–17.3 mm, length of antenna is 8.2–10.0 mm. General body coloration is brownish yellow.
Figure 18Habitus of *Tipula* (*Vestiplex*) *survilai*
**sp. nov**. (**A**). Male, holotype, lateral view. (**B**). Male antennae, holotype. (**C**). Male thorax, paratype, dorsal view. (**D**). Female, paratype, lateral view. Scale bars: a (**A**,**D**) = 5 mm, b (**B**) = 2 mm, (**C**) = 1 mm.
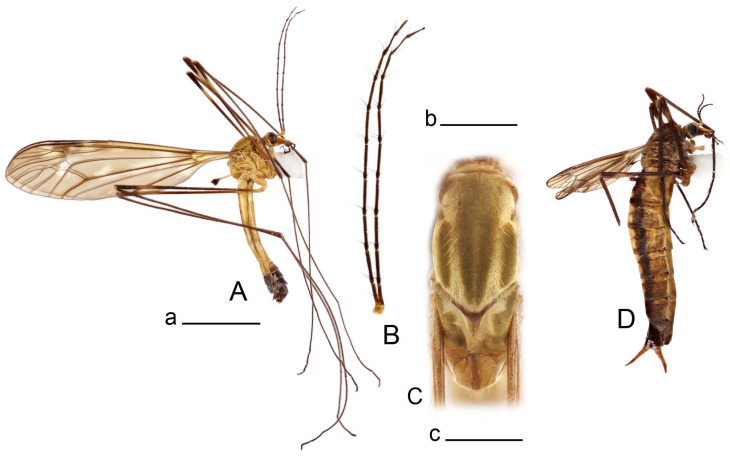



Head. Yellow, vertex and occiput with narrow, brown median line. Rostrum yellow, nasus distinct. Palpus brown. Antenna 13-segmented (Figure 18B), if bent backward reaching beyond the base of abdomen. Scape and pedicel yellow; the base first flagellar segment narrowly yellow, remaining segments brown. Each flagellomere, except first, with light basal enlargement. Apical flagellomere small, distinctly shorter than preceding flagellomere. Long verticils about twice as short as corresponding flagellomeres.

Thorax (Figure 18C). Pronotum yellow, with narrow line. Prescutum and presutural scutum with four olive brown stripes, bordered by greenish brown. Central stripes separated by light area. Interspaces yellow pollinose. Postsutural scutum greenish brown, with median line. Each scutal lobe with two olive brown spots. Scutellum yellow, with median line. Mediotergite yellow, with vestige of median vitta. Pleura greenish yellow. Legs with coxae and trochanters yellow. Femora with base narrowly yellow, rest brown with yellow preapical ring. Tibiae and tarsal segment brown. Claw without tooth. Wing clear pale brown, tinged with brown (Figure 18A). Costal and subcostal area yellowish. Brown postarcular darkening in cells br and bm, brown spot at origin of Rs and stigma area. A faint brown cloud in the middle of br and bm and distal area of bm. Vein m-cu joining dm at fork of M_3+4_, M_1+2_ about 0.15X as short as length of cell m_1_. Halter yellow, knob brown.

Abdomen. Abdominal tergites 1–5 yellow, with broad brown median and narrow lateral lines. Remaining tergites brown. Sternites 1–5 yellow. Remaining segments brown.

Hypopygium (Figure 19 and Figure 20). Male hypopygium with tergite 9 fused with sternite 9 basally (Figure 19A). Tergite 9 completely divided at midline by pale membrane, posteriorly with V-shaped notch (Figure 19B–E). Dorsal portion reduced, narrow in midline, posterior margin with two pair of lobes directed caudad. Posteromedian lobes flattened, broadened at base, covered with setae, located on either side of midline; posterolateral lobes oval with setae at tip. Ventral portion hypertrophied and occupying nearly the entire tergite area, forking into flattened brown plates. Each plate serrulated on inner margin along midline, with surrounding surface densely covered with black dots, posterior margin terminating into black, nearly triangular process. Gonocoxite separated from sternite 9 by suture, apically produced into black spine (Figure 19A,F). Outer gonostylus distally oval (Figure 19G). Inner gonostylus a blackened plate terminating into obtuse beak, lower beak nearly oval (Figure 19G,I). Dorsal margin black, apically produced into obtuse angle which is bent inwardly, outer basal lobe flattened and bent inwardly. Sternite 9 with ventral lobe of A9s nearly triangular, covered with setae (Figure 19K). Dorsal lobe of A9s fused with ventral lobe, wedge-shaped. Adminiculum triangular in ventral view, basally dilated with lateral parts raised. Medial portion protruded, apex slightly curved, preceding round incision (Figure 19J). Gonocoxal fragment reduced. Sperm pump with central vesicle swollen (Figure 20A). Compressor apodeme with broad median incision (Figure 20B). Posterior immovable apodeme narrow, anterior immovable apodeme rounded. Aedeagus about 3.3× as long as sperm pump (Figure 20A).
Figure 19Male terminalia of *Tipula* (*Vestiplex*) *survilai*
**sp. nov**. (**A**). Hypopygium, lateral view. (**B**). Hypopygium, dorsal view. (**C**) Tergite 9, dorsal view. (**D**). Tergite 9, same sclerite pressed by cover glass, pml and pll directed caudad. (**E**). Tergite 9, caudal view. (**F**). Gonocoxite. (**G**). Outer gonostylus. (**H**). Left inner gonostylus, lateral view. (**I**). As previous, dorsal view. (**J**). Adminiculum, lateral view. (**K**). Sternite 9 with adminiculum and appendages. Abbreviations: dl—dorsal lobe of A9s; pll—posterolateral lobe of tergite 9; pml—posteromedian lobe of tergite 9; vl—ventral lobe of A9s. Scale bar: a (**A**–**K**) = 0.5 mm.
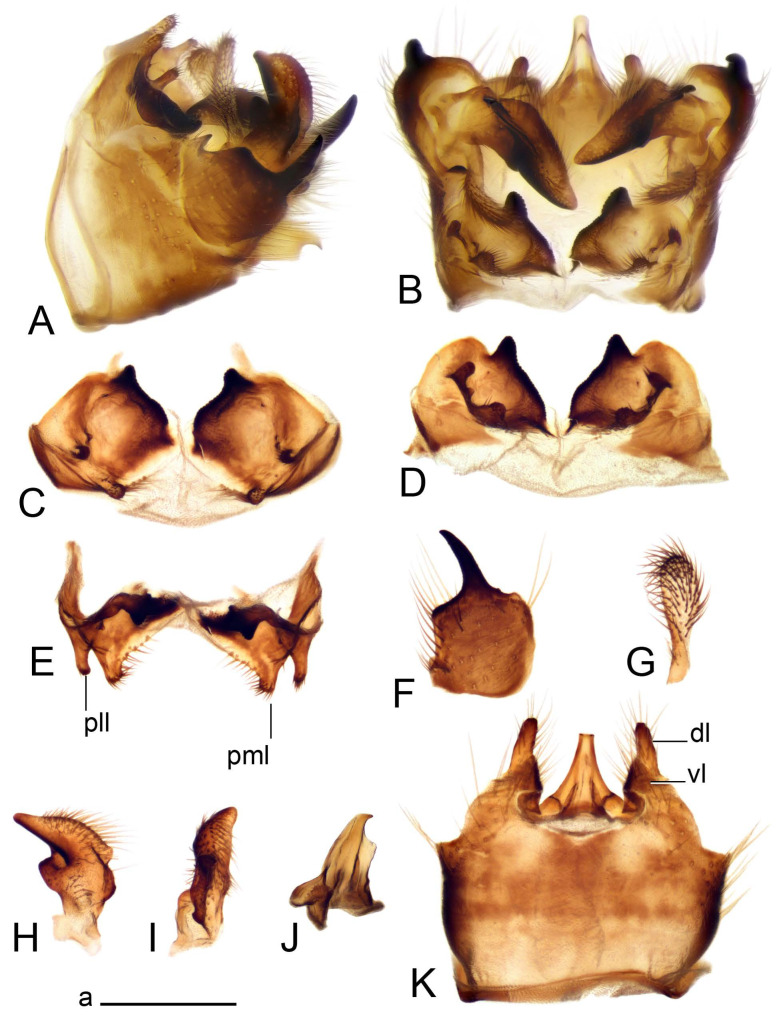

Figure 20Male terminalia of *Tipula* (*Vestiplex*) *survilai*
**sp. nov**. (**A**). Semen pump and intromittent organ, lateral view. (**B**). Semen pump, dorsal view. Scale bar: a (**A**,**B**) = 0.5 mm.
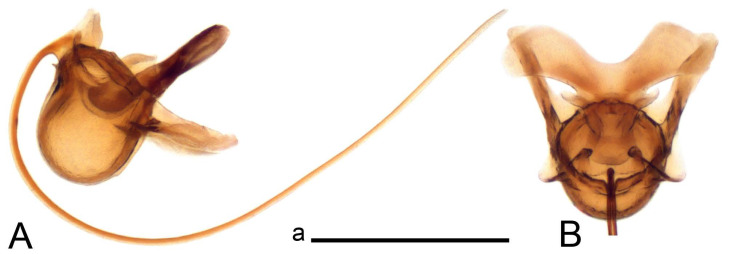



**Female** (Figure 18D). Body length is 17.7–18.1 mm, wing length is 5.5–7.6 mm, length of antenna is 1.5–2.1 mm. Generally similar to male in body coloration, except for brown thorax. Female is brachypterous, with wings short and narrow with visible veins.

*Ovipositor* (Figure 21). Tergite 10 shiny brown. Cercus brown, nearly straight, slightly longer than tergite 10, tip rounded and slightly up-turned (Figure 21A). Ventral and dorsal margins with visible serration. Sternite 8 with hypovalva heavily blackened and short (Figure 21B,C). Posterior margin of hypovalva with submedian incision forming inner triangular lobe with acute tip, and outer lobe possessing additional small emargination. Median incision between hypovalvae basally with setae. Sternite 9 with posterior half flattened, nearly rectangular, surface covered with short filaments, with broad median groove; anterior parts slightly bent at base (Figure 21D). Furca posteriorly flattened, anterior part narrowed (Figure 21E). Spermatheca pear-shaped (Figure 21F).

**Etymology.** The new species is named after Mindaugas Survila, a Lithuanian naturalist and filmmaker known for his films as *The Ancient Woods* (2017) and *The Field of Magic* (2001).

**Distribution.** Taiwan (Figure 3B).

**Elevation range in Taiwan.** Adults were collected at altitudes ranging from 2200 m to 3400 m.

**Period of activity.** Adults are active from October through to August.
Figure 21Female terminalia of *Tipula* (*Vestiplex*) *survilai*
**sp. nov**. (**A**). Ovipositor, left lateral view. (**B**). Sternite 8 with hypovalvae, ventral view. (**C**). Hypovalvae, pressed by cover glass, ventral view. (**D**). Sternite 9, dorsal view (**E**). Furca, dorsal view. (**F**). Spermatheca. Scale bars: a (**A**), b (**B–E**) = 0.5 mm.
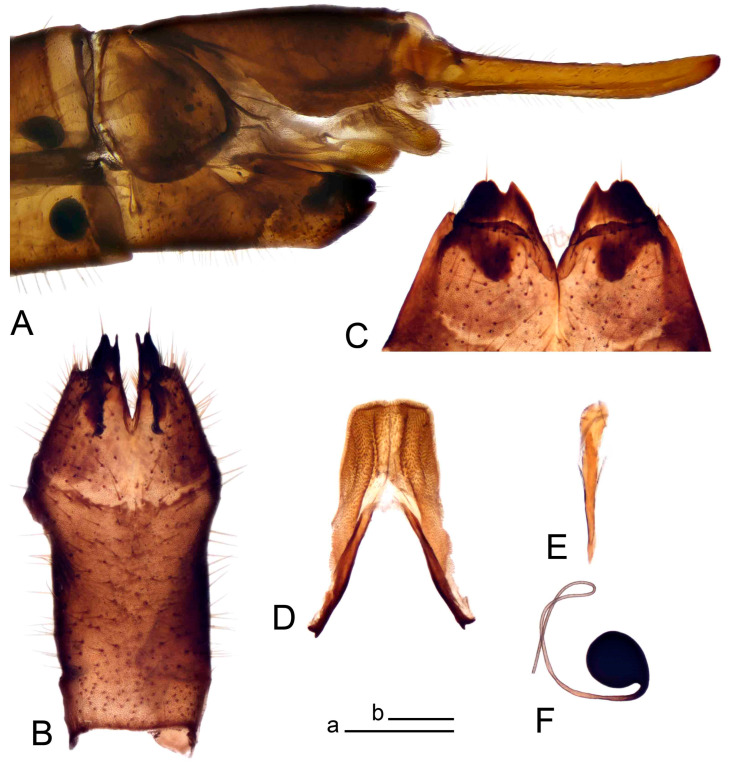

insects-14-00616-sch003_Scheme 3Scheme 3Distribution maps of *Tipula* (*Vestiplex*) crane flies in Taiwan. Purple circles represent holotype, yellow circles represent paratypes. (**A**). *T.* (*V.*) *graciliana* **sp. nov.** (**B**). *T.* (*V.*) *survilai* **sp. nov**.
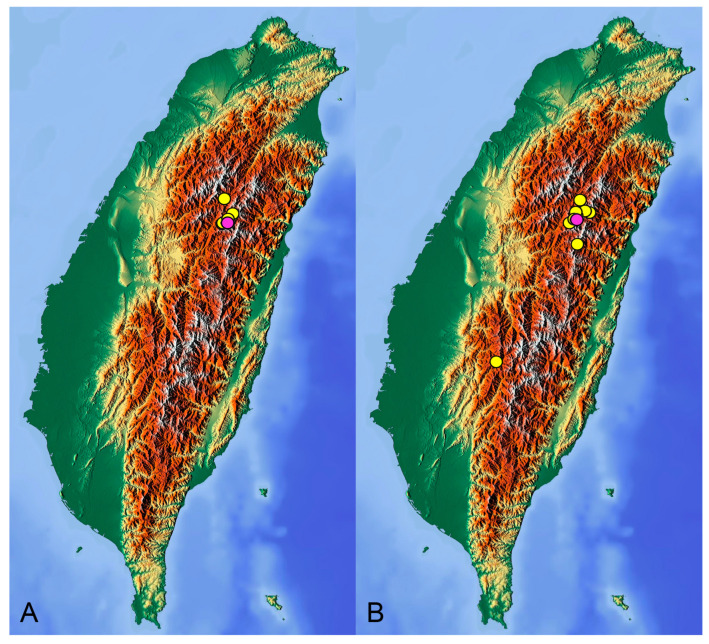




***Tipula* (*V.*) *coxitalis* species group**


The *coxitalis* group was proposed by Alexander [4] for *T.* (*V.*) *coxitalis* and *T.* (*V.*) *sternotuberculata* and also includes *T.* (*V.*) *bicalcarata* Savchenko, 1965, *T.* (*V.*) *leigongshanensis* Men and Young, 2017, and *T.* (*V.*) *maoershanensis* Men and Young, 2017. The maximum likehood tree based on a CO1 sequences tree also supports the close relationship of the listed species ([40]: Figure 39). *Tipula* (*V.*) *proboscelongata* Yang and Yang, 1991, *T.* (*V.*) *lushuiensis* Men and Strakevich, 2021, and *T.* (*V.*) *bispathifera* Savchenko, 1960 are placed here in the *coxitalis* species group according to the shape of the gonocoxite and inner gonostylus ([41]: Figures 7–9; [42]: Figure 2; [43]: Figures 1–14) and discussed below.

Males of the *coxitalis* species group can be easily recognized by their massive elongated gonocoxite which is broad at the base, apically produced into a long arm with the tip tapering or expanded and the inner gonostylus middorsally having a short or elongated lobe. They can be also characterized by tergite 9 divided along midline by pale membrane with dorsal portion occupying the majority of the area; ventral portion with a pair of well-developed plates; outer gonostylus relatively small; sternite 9 with dorsal lobe of A9s developed, claw- or finger-shaped.

Females are known for *T.* (*V.*) *coxitalis* and *T.* (*V.*) *sternotuberculata*. The cercus of both species is smooth, without visible serration. Sternite 8 with lateral parts flattened. Hypovalva developed blade-shaped, median incision between hypovalvae basally with slightly serrated edge and provided with setae.


***Tipula* (*Vestiplex*) *coxitalis* Alexander**


Figure 22, Figure 23, Figure 24 and Figure 25; Figure 4A

*Tipula* (*Oreomyza*) *coxitalis* Alexander, [44]: 532;

*Tipula *(*Oreomyza*) *coxitalis*: Alexander, [4]: 122;

*Tipula* (*Unplaced*) *coxitalis*: Alexander and Alexander, [34]: 66;

*Tipula* (*Vestiplex?*) *coxitalis*: Men, [35]: 801–804, 807;

*Tipula coxitalis*: Men et al. [45]: 94;

*Tipula* (*Vestiplex*) *coxitalis*: Men et al. [40]: 66;

*Tipula* (*Vestiplex*) *longarmata* Yang and Yang, [46]: 63 **syn. nov.**;

*Tipula* (*Oreomyza*) *subnata* Alexander, [47]: 437 **syn. nov.**;

*Tipula* (*Unplaced*) *subnata*: Alexander and Alexander, [34]: 67.


**Diagnosis**


*Tipula* (*V.*) *coxitalis* can be recognized by the gonocoxite having a long and straight outgrowth, and a pencil of long reddish setae on either side of the median line of sternite 9. The female has an elongated tergite 10 and short cercus whose ventral margin is smooth.

**Type of material examined**. *Tipula* (*Vestiplex*) *coxitalis* Alexander: **TAIWAN** • Holotype ♂; Formosa, Hassensan; alt. 6532 ft.; 26 June 1934; Gressit leg.; slide mounted antenna, wing, and genitalia; USNM • Paratype 1 ♀; same data as for preceding; alt. 4875 ft.; 22 June 1934; USNM • Paratype 1 ♂; same data as for preceding; alt. 6700 ft.; 26 June 1934; USNM • Paratype 1 ♀; same data as for preceding; Bukai; alt. 2775 ft.; 14 June 1934; USNM.

*Tipula subnata* Alexander: **CHINA** • Holotype ♂; Fukien [Fujian], Ta-chu-lan; alt. 1500 m; 18 May 1948; Joseph Fu leg.; antenna, leg, wing, and genitalia mounted on slide (USNM).

**Additional material examined. TAIWAN**—**Yilan Co.** • 2 ♂♂; Yuan-Shan Woods near Shuang-Lian-Pi, N24.7485, E121.6413; alt. 486 m; 20 May 2010; Chen Young leg.; CMNH543689; CO1 Barcode TIPTW070-10; CMNH543688; CMNH—**Nantou Co.** • 2 ♂♂; Ren-Ai Huei-Sun Forest Station; N24.0909, E121.0335; alt. 687 m; 6 May 2010; Chen Young leg.; CO1 Barcode TIPTW177-10; CMNH543795; CO1 Barcode SATIP1140-10; CMNH543618; CMNH • 4 ♂♂, 1 ♀; same data as for preceding; CO1 Barcode TIPTW176-10; CMNH543794; dissected; CO1 Barcode SATIP1136-10; CMNH543614; CO1 Barcode SATIP1083-10; CMNH543561; CMNH543616; CO1 Barcode TIPTW887-11; CMNH544805; dissected; CMNH—**Taichung Co.** • 1 ♀; Heping Twp Snow Mountains; N24.2453, E121.5368; alt. 2240; 30 May 2011; Yu-Chuan Chiu leg.; CMNH575719; CMNH—**Taipei Co.** • 2 ♂♂; Shihlin Yangmingshan DaTun Falls; N25.1270, E121.5885; alt. 235 m; 17 April 2010; Chen Young leg.; CMNH543641; CMNH543642; CMNH—**TaiTung Co.** • 2 ♂♂; Jhihben Li-Jia Forest Road 14K; N22.7986, E121.0334; alt. 1060 m; 2 May 2010; Chen Young leg.; CMNH455249; dissected; CMNH412399; NRC • 1 ♂, 1♀; same data as for preceding; CMNH544085; CMNH544078; CMNH. **CHINA** • 1 ♀; Fukien [Fujian], Kuatun; alt. 2500–3000 ft.; 26 May 1938; Klapperich leg.; Tipula sp. Det. C. P. Alexander, met. 193; Det. P. Starkevich 2021; ZFMK.


**Redescription**


**Male** (Figure 22A). Body length is 12.7–16.4 mm, wing length is 17.2–19.8 mm, length of antenna 3.7–4.4 mm. General body coloration is yellow.

Head. Yellow, with median vitta on vertex and occiput. Nasus distinct. Palpus dirty yellow. Antenna 13-segmented, if bent backward reaching base of the wing. Scape, pedicel and first flagellar segment yellow; flagellar segments 2–4 bicolorous, remaining segments brown. Each flagellomere, except first, with small, dark brown enlargement. Long verticils equal to length of corresponding flagellomeres.

*Thorax* (Figure 22B). Overall yellow. Pronotum yellow with brown median spot. Prescutum and presutural scutum with four distinct stripes. Central stripes yellowish brown, bordered by brown, separated by yellow. Lateral stripes dark brown bordered by brown. Interspaces yellow. Postsutural scutum yellow, each lobe with two spots. Anterior spot yellowish, bordered by brown, posterior spot brown, bordered by darker brown, twice as large. Scutellum and mediotergite yellowish brown with central vitta. Pleura yellow. Legs with coxae and trochanters yellow. Femora yellowish brown, darkened at tip. Tibiae basally yellowish brown, distal part brown, tarsal segments brown. Claw with tooth. Wing clear yellowish brown variegated with light brown clouds (Figure 22A). Costal and subcostal area yellowish, stigma dark brown. Brown spot at fork of R. Vein m-cu joining dm at fork of M_3+4_, M_1+2_ about 0.25× as long as length of cell m_1_. Halter pale yellow, knob basally brown.
Figure 22Habitus of *Tipula* (*Vestiplex*) *coxitalis* (**A**). Male, lateral view. (**B**). Male thorax, dorsal view. (**C**). Female, lateral view. Scale bars: a (**A**,**C**) = 5 mm, b (**B**) = 1 mm.
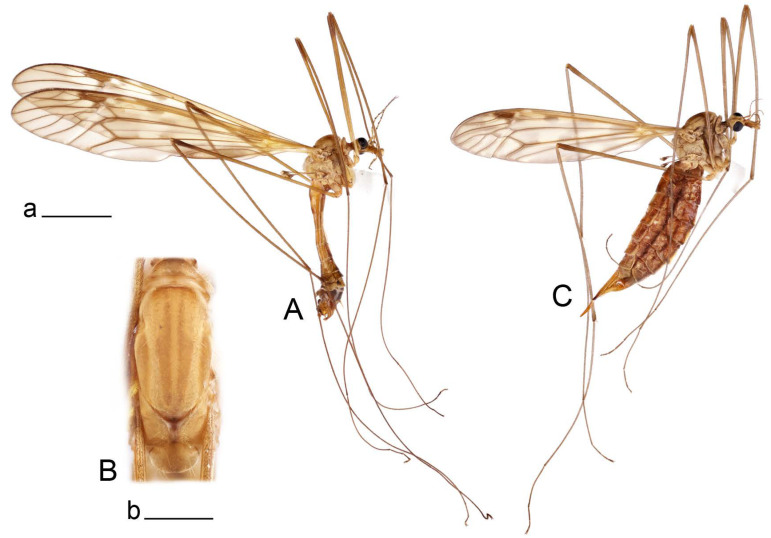



*Abdomen*. Tergites 1–5 yellow, with broad, dark brown median and narrow lateral stripes. Tergite 6 laterally yellow, dorsal area dark brown. Remaining tergites dark brown. Sternites 1–6 yellow, medially with dark brown stripe, remaining dark brown.

*Hypopygium* (Figure 23 and Figure 24). Male hypopygium with tergite 9 fused with sternite 9 basally (Figure 23A). Tergite 9 divided along midline by pale membrane, posteriorly with broad V-shaped notch (Figure 23B,C). Dorsal portion of tergite 9 posterolaterally produced into short, darkened obtuse lobes, posteromedian margin sligthly rounded; ventral portion with a pair of flattened, brown, nearly triangular plates. Gonocoxite separated from sternite 9 by suture, massive and elongated; basal portion broad, apically produced into long arm with tip expanded (Figure 23A,D,H,G). Outer gonostylus small, short, finger-shaped (Figure 23E). Inner gonostylus terminating into black and obtuse beak, dorsal margin brown, covered with long setae; middorsal margin with gently curved lobe, a round incision and obtuse projection below the lobe; a long rod along ventral margin terminating in round lower beak (Figure 23F). Sternite 9 near caudal end of either side of midline with a conspicuous hair pencil of reddish bristles (Figure 23G). Appendage of sternite 9 with ventral lobe darkened, narrow, covered with long setae (Figure 23I,H); dorsal lobe curved, claw-shaped (Figure 23J,H). Adminiculum relatively narrow, nearly triangular in ventral view, medially protruded into narrow sclerite (Figure 23G,K); tip slightly curved in ventral projection (Figure 23L). Sperm pump with central vesicle swollen (Figure 24A). Compressor apodeme flattened, with median incision (Figure 24B). Posterior immovable apodeme narrowed, anterior immovable apodeme flattened. Aedeagus about 2.5× as long as sperm pump (Figure 24A).
Figure 23Male terminalia of *Tipula* (*Vestiplex*) *coxitalis*. (**A**). Hypopygium, lateral view. (**B**). Tergite 9, dorsal view. (**C**) Tergite 9, caudal view. (**D**). Gonocoxite. (**E**) Outer gonostylus. (**F**). Left inner gonostylus, lateral view. (**G**). Sternite 9, tergite 9 and gonostyli removed, ventral view. (**H**). Gonocoxite and appendages of A9s, mesal view. (**I**). Ventral lobe of A9s. (**J**). Dorsal lobe of A9s. (**K**). Adminiculum, ventral view. (**L**). Adminiculum, lateral view. Abbreviations: dl—dorsal lobe of A9s; dp—dorsal portion of tergite 9; gcx—gonocoxite; vl—ventral lobe of A9s; vp—ventral portion of tergite. Scale bar: **a** (**A–L**) = 0.5 mm.
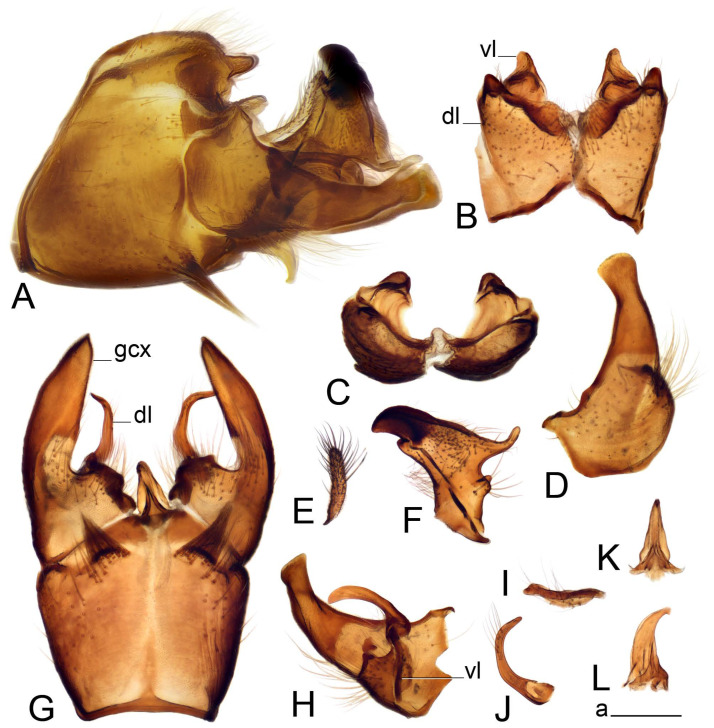



**Female** (Figure 22C). Body length is 14.6–19.1 mm, wing length is 15.0–17.7 mm, length of antenna is 2.1–2.5 mm. Generally similar to male in body coloration. Flagellar segments, except first one, bicolorous.

*Ovipositor* (Figure 25). Tergite 10 elongated, shiny yellow, distally blackened. Cercus short, about 2.5X as short as tergite 10, slightly curved, narrowing towards tip; ventral margin smooth, without visible serration (Figure 25A). Hypovalva extending slightly before base of cercus. Sternite 8 flattened from the middle into rounded lobe, imarginated at the lateral angle and reaching basal third of hypovalva (Figure 25B,C). Sternite 9 with posterior part wedge-shaped, rounded at margin; the surface covered with short filaments, anterior parts membranouos (Figure 25D). Furca in the shape of filament, posteriorly flattened (Figure 25E). Spermatheca broadened at base (Figure 25F).
Figure 24Male terminalia of *Tipula* (*Vestiplex*) *coxitalis*. (**A**). Semen pump and intromittent organ, lateral view. (**B**). Semen pump, dorsal view. Scale bar: a (**A**,**B**) = 0.5 mm.
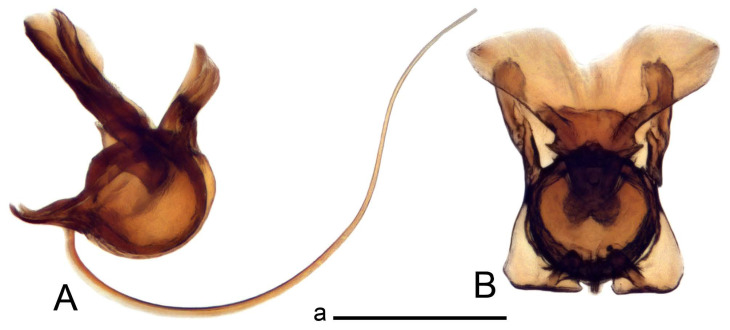



**Distribution.** Taiwan (Figure 4A), China (Anhui, Fujian and Hubei).

**Elevation range in Taiwan.** Adults were collected at altitudes ranging from 200 m to 2400 m.

**Period of activity.** Adults are active from April through to June.

**Remarks.** *Tipula* (*V.*) *coxitalis* was described by Alexander [4] based on both male and female specimens collected by J. L. Gressitt in 1934 from Hassensan (台中和平八仙山 2042–1486 m); aditional specimens of both sexes were collected by J. L. Gressett in 1934 from Bukai (屏東霧台 845 m). The female asssociation of this species was confirmed in a current DNA gene sequence study.

The synonymy *T.* (*V.*) *coxitalis* = *T.* (*V.*) *subnata*
**syn. nov.** was established after comparison of type specimens of both taxa. The synonymy of *T.* (*V.*) *coxitalis* = *T.* (*V.*) *longarmata*
**syn. nov.** is based on a comparison with an original description and illustration [46]. *Tipula* (*V.*) *longarmata* was described from a single male collected in Hubei, China. The shape of the inner gonostylus, gonocoxite, and tergite 9 ([46]: Figures 1–3) is undoubdetly specific to *T.* (*V.*) *coxitalis*.
Figure 25Female terminalia of *Tipula* (*Vestiplex*) *coxitalis*. (**A**). Ovipositor, left lateral view. (**B**). Sternite 8 with hypovalvae, ventral view. (**C**). Sternite 8 with hypovalvae, same sclerite pressed by cover glass, ventral view. (**D**). Sternite 9, dorsal view (**E**). Furca, dorsal view. (**F**). Spermatheca. Scale bars: a (**A**), b (**B**–**E**) = 0.5 mm; c (**F**) = 0.25 mm.
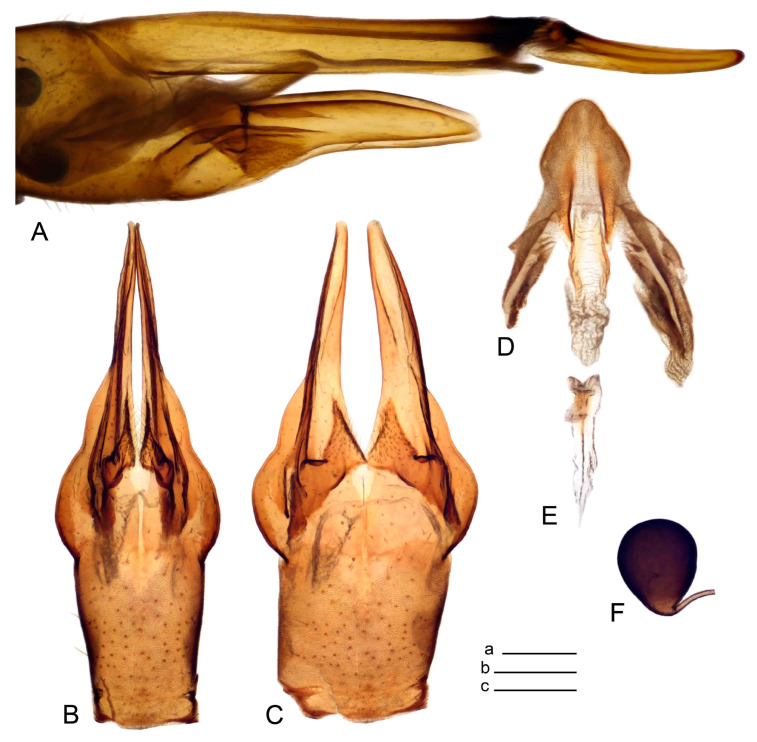



***Tipula* (*Vestiplex*) *sternotuberculata*** Alexander

Figure 26, Figure 27, Figure 28 and Figure 29; Figure 4B

*Tipula* (*Oreomyza*) *sternotuberculata* Alexander, [44]: 534;

*Tipula *(*Oreomyza*) *sternotuberculata*: Alexander, [4]: 122;

*Tipula* (*Unplaced*) *sternotuberculata*: Alexander and Alexander, [34]: 67;

*Tipula* (*Vestiplex*) *sternotuberculata:* Men et al. [40]: 66.


**Diagnosis**


*Tipula* (*V.*) *sternotuberculata* can be recognized by the gonocoxite having a long, bended outgrowth, and the black tubercles tufted with short, golden yellow setae on either side of the median line of sternite 9. The female has a smooth cercus, with a length subequal to that of tergite 10.

**Type of material examined. TAIWAN** • Holotype ♂; Arisan; alt. 7640 ft.; 24 May 1934; Gressit leg.; USNM • Paratype ♂; same data as for preceding; alt. 6200 ft.; 26 May 1934; USNM.

**Additional material examined. TAIWAN**—**Chiayi Co.** • 1 ♂; Alishan; [23°21′ N, 120°48′ E]; 12 April 1965; Yoshimoto; CMNH412948; CMNH • 1 ♂; same data as for preceding; 4 April 2011; CMNH450597; CMNH—**Hualien Co.** • 1 ♂; KuanYuan; [N24.1850667, E121.3421278]; alt. 2734 m; 16 May 2007; Ling-Chu Lin leg.; CO1 Barcode TIPTW217-10; CMNH543835; • 1 ♀; SiouLin, E. end of JinMa Tunnel; N24.1773; E121.3727; alt. 2508 m; 14 June 2011; Chen Young leg.; CMNH452046; dissected; NRC • 1 ♂; same data as for preceding; Han-Tzu Hsu leg.; CMNH454260; CMNH • 1 ♂; same data as for preceding; Yu-Da Chuan leg.; CMNH451291; CMNH • 1 ♂; same data as for preceding; Chen Young leg.CMNH543594; CMNH • 1 ♂; SiouLin, pi-Lu Sacred Tree Trail; N24.1739, E121.3975; alt. 2146 m; 3 May 2010; Chen Young leg.; CMNH544084; CMNH—**Kaohsiung Co.** • 1 ♂; Taoyuan Twp, TengZhih Station; N23.0848, E120.7877; alt. 1806 m; 4 May 2011; Yu-Chuan Chiu leg.; CMNH448673; NRC • 1♂; Taoyuan Twp, TengZhih Pk Mid-altitude Res. Inst.; N23.0853; E120.7884; alt. 1638 m; 13 April 2011; Chen Young leg.; CMNH544501; CMNH—**Taichung Co.** • 1 ♂; Bi-LuSi, Taiwan Forest Res. Inst.; N24.2226, E121.3050; alt. 2149 m; 12 May 2010; Chen Young leg.; CMNH452307; dissected; NRC • 1 ♂♂, 1 ♀; same data as for preceding; CMNH543602; dissected; CO1 Barcode SATIP1138-10; CMNH543616; CO1 Barcode SATIP1137-10; CMNH543615; CMNH • 1 ♂; Pilushi; [N24.24750, E121.30720]; alt. 2200 m; 1988.05.22-23; R. Davidson, J. Rawlins, C. Young leg.; CO1 Barcode TIPTW432-10; CMNH544050; CMNH.


**Redescription**


**Male** (Figure 26A). Body length is 13.2–15.5 mm, wing length is 18.0–18.9 mm, length of antenna is 4.0–4.6 mm. General body coloration is yellow.

Head. Yellow, with median line on vertex and occiput. Nasus distinct. Palpus yellow. Antenna 13-segmented, if bent backward reaching base of the wing. Scape, pedicel, and first flagellar segment yellow; flagellar segments 2–4 bicolorous, remaining segments brown. Each flagellomere, except first, with dark brown enlargement. Long verticils equal to length of corresponding flagellomeres.

*Thorax* (Figure 26B). Overall yellow. Pronotum yellow with brown median spot. Prescutum and presutural scutum with four brown stripes. Central stripes without distinct borders, separated by yellowish brown. Interspaces yellow. Postsutural scutum yellow, each lobe with two spots. Anterior spot brown, posterior spot yellowish. Scutellum and mediotergite yellow with central vitta. Pleura yellow. Legs with coxae and trochanters yellow. Femora yellow, darkened at tip. Tibiae and tarsal segments brownish black. Claw with tooth. Wing clear pale brown, cells with light spots. (Figure 26A). Costal and subcostal area yellowish, stigma dark brown. Brown spot at fork of Rs. Vein m-cu joining dm at fork of M_3+4_, M_1+2_ about 0.25× as long as length of cell m_1_. Halter pale-yellow, knob basally brown.

*Abdomen*. Abdominal segments 1–5 yellow, tergite and sternite 6 with black, broad line medially. Remaining segments brownish black.
Figure 26Habitus of *Tipula* (*Vestiplex*) *sternotuberculata*. (**A**). Male, lateral view. (**B**). Male thorax, dorsal view. (**C**). Female, lateral view. Scale bars: a (**A**,**C**) = 5 mm, b (**B**) = 1 mm.
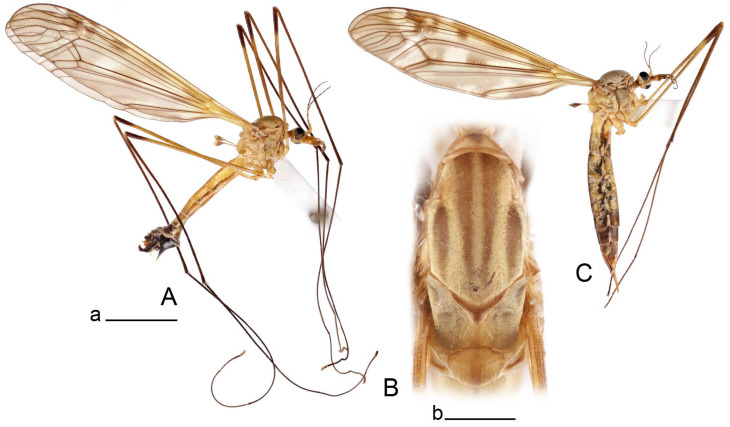


*Hypopygium* (Figure 27 and Figure 28). Male hypopygium with tergite 9 fused with sternite 9 basally (Figure 27A). Tergite 9 divided along midline by pale membrane, posteriorly broad U-shaped notch (Figure 27B–D). Dorsal portion of tergite 9 with posterior margin provided with a pair of tufts of long setae on either side of midline. Ventral portion blackened, terminating in a pair of brown, leaf-shaped plates. Gonocoxite separated from sternite 9 by suture, massive and elongated; basal portion broad, apically produced into gently curved arm with tip flattened (Figure 27A,B,E,F). Outer gonostylus small, short, finger-shaped (Figure 28A). Inner gonostylus a stocky plate, terminating into black and slender beak, dorsal margin with black round lobe, middorsal margin with short, horn-shaped lobe; posterior margin convex with round incision; lower beak an obscure round projection (Figure 28B,C). Sternite 9 near caudal end of either side of midline with a dark, long, basal tubercle tufted with golden yellow setae (Figure 27A and Figure 28D). Appendage of sternite 9 with ventral lobe well developed, provided with long setae (Figure 27F and Figure 28G); dorsal lobe finger-shaped flattened at apex (Figure 27F and Figure 28H). Adminiculum relatively narrow, medially protruded into narrow sclerite (Figure 28E); tip slightly curved in ventral projection (Figure 28F). Sperm pump with central vesicle spherical (Figure 28I).

Compressor apodeme flattened, with median rod (Figure 28J). Posterior immovable apodeme narrow at base, distally dilated, anterior immovable apodeme rounded. Aedeagus about 2.5× as long as sperm pump (Figure 28I).

**Female** (Figure 26C). Body length is 16.9–17.5 mm, wing length is 15.9–17.5 mm, length of antenna is 2.3–2.4 mm. Generally similar to male in body coloration. Antenna is brown.

*Ovipositor* (Figure 29). Tergite 10 shiny, dark brown. Cercus yellow, subequal to length of tergite 10, straight, with tip round; ventral margin smooth, without visible serration (Figure 29A). Hypovalva extending slightly beyond base of cercus. Sternite 8 with hypovalva developed, blade-shaped; median incision between hypovalvae basally with indistinctly serrated edge and covered with setae (Figure 29B,C). Lateral angle of sternite 8 produced into small rounded lobe. Sternite 9 nearly triangular in outline with posterior margin acute; anterior parts acute at base (Figure 29D). Furca in the shape of membranuous oblong sclerite (Figure 29E). Spermatheca pear-shaped (Figure 29F).
Figure 27Male terminalia of *Tipula* (*Vestiplex*) *sternotuberculata*. (**A**). Hypopygium, lateral view. (**B**). Hypopygium, dorsal view. (**C**). Tergite 9, dorsal view. (**D**). Tergite 9, dorso-caudal view. (**E**). Gonocoxite. (**F**). Gonocoxite and appendages of A9s, mesal view. Scale bars: a (**A**,**B**), b (**C**–**F**) = 0.5 mm.
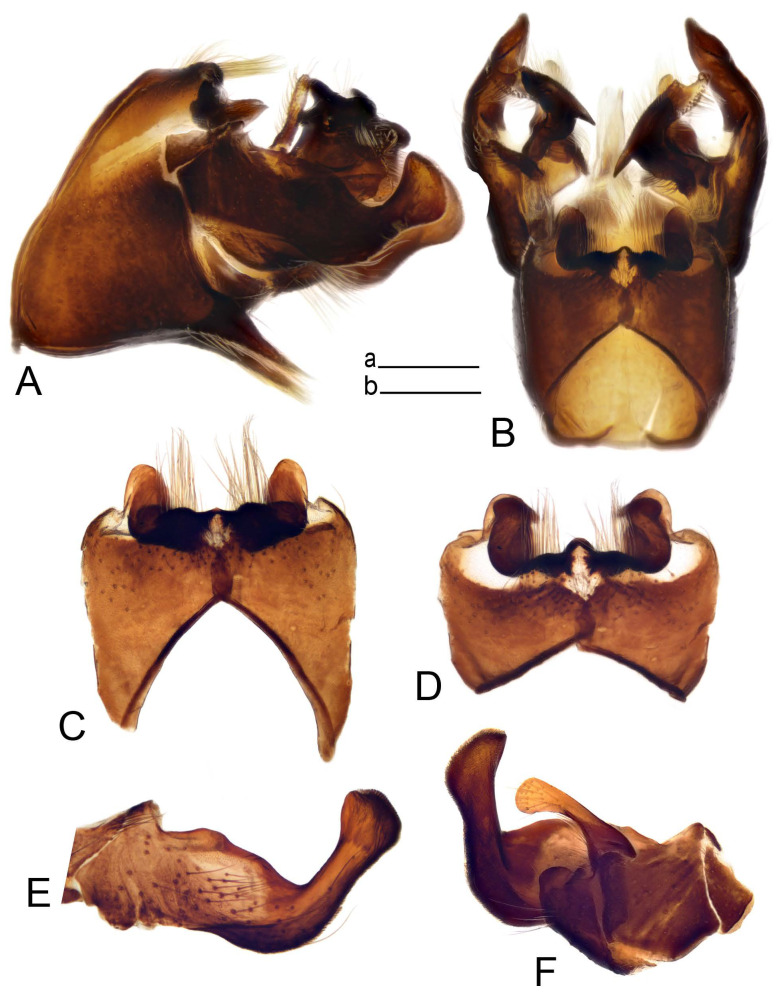

Figure 28Male terminalia of *Tipula* (*Vestiplex*) *sternotuberculata*. (**A**). Outer gonostylus. (**B**). Left inner gonostylus, mesal view. (**C**). Left inner gonostylus, lateral view. (**D**). Sternite 9, ventral view. (**E**). Adminiculum, ventral view. (**F**). Adminiculum, lateral view. (**G**). Ventral lobe of A9s**.** (**H**). Dorsal lobe of A9s**.** (**I**). Semen pump and intromittent organ, lateral view. (**J**). Semen pump, dorsal view. Scale bar: a (**A**–**J**) = 0.5 mm.
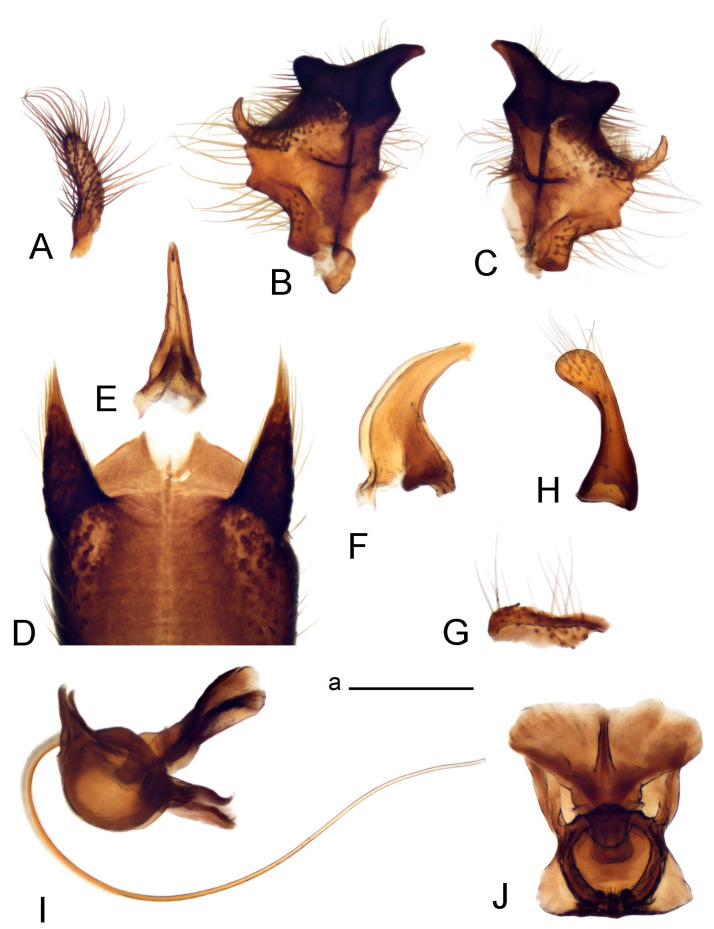

Figure 29Female terminalia of *Tipula* (*Vestiplex*) *sternotuberculata*. (**A**). Ovipositor, sternites removed, left lateral view. (**B**). Sternite 8 with hypovalvae, ventral view. (**C**). Hypovalvae pressed by cover glass, ventral view. (**D**). Sternite 9, dorsal view. (**E**). Furca, dorsal view. (**F**). Spermatheca. Scale bar: a (**A**–**F**) = 0.5 mm.
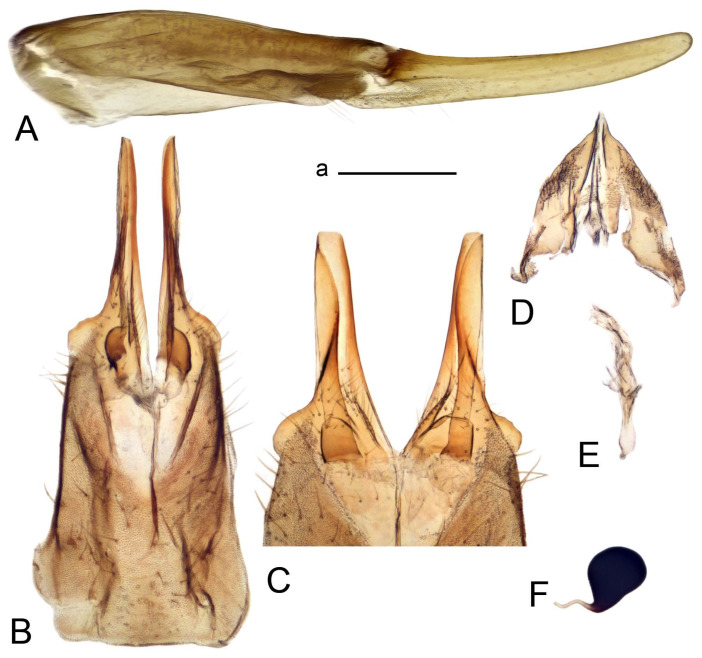



**Distribution.** Taiwan (Figure 4B), China (Guizhou).

**Elevation range in Taiwan.** Adults were collected at altitudes ranging from 1000 m to 2200 m.

**Period of activity.** Adults are active from April through to May.

**Remarks.** *Tipula sternotuberculata* was described by Alexander [44] based on male specimens collected by J. L Gressitt in 1934 from Arisan (嘉義阿里山 2328–1890 m). The female asssociation of this species is based on a current DNA gene sequence study.
insects-14-00616-sch004_Scheme 4Scheme 4Distribution maps of *Tipula* (*Vestiplex*) crane flies in Taiwan. (**A**). *T.* (*V.*) *coxitalis*. (**B**). *T.* (*V.*) *sternotuberculata*. Yellow circles represent paratypes.
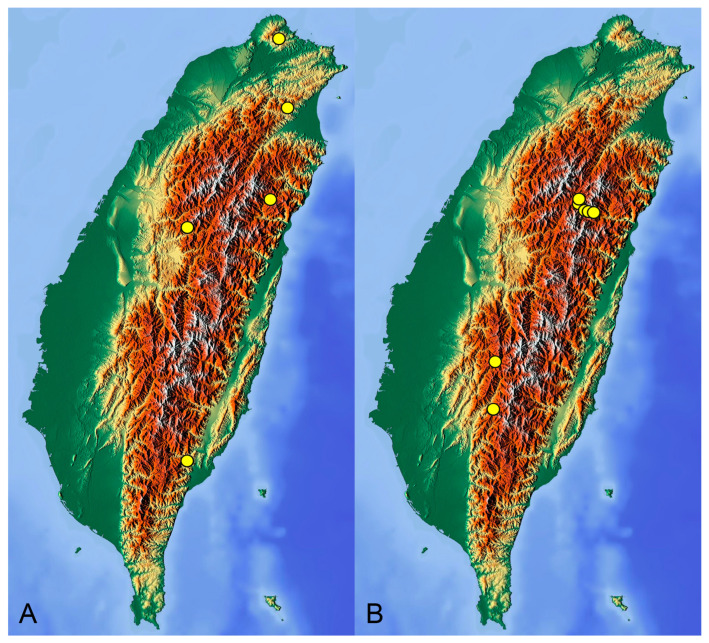




***Tipula* (*Vestiplex*)**
***deserrata* species group**


The *deserrata* species group was proposed by Starkevich [15] with a list of species revised in Men et al. [38]. Another two species have been described recently from Tibet, China [48]. Three species are recognized in Taiwan, including *T.* (*V.*) *biserra*, *T.* (*V.*) *formosae* **sp. nov**., and *T.* (*V.*) *pseudobiserra* **sp. nov**.

Males of the *deserrata* group can be recognized by the following features: tergite 9 divided at midline by pale membrane. Ventral portion with a pair of blackened and microscopically roughened or serrulated lobes shaped as finger, rod, spine, or knob. Gonocoxite bispinuous, apically produced into a strong black spine, lobe, or curved arm. Ventromesal portion of gonocoxite in the shape of short spine, horn, or black point. Sternite 9 with dorsal lobe of A9s reduced into a small, round, oval, finger-like, or spinuous sclerite located nearby margin of junction of gonocoxite with sternite 9 [38].

Females are illustrated for Taiwanese species and *T.* (*V.*) *serricauda* Alexander, 1914 ([49]: Figures 38–41). Hypovalvae short, plate-shaped, median incision between hypovalvae basally with slightly serrated or smooth edge and provided with setae.


***Tipula* (*Vestiplex*) *biserra* Edwards**


Figure 30, Figure 31, Figure 32 and Figure 33; Figure 5A

*Tipula biserra* Edwards, [33]: 108;

*Tipula* (*Vestiplex*) *biserra*: Edwards, [2]: 80;

*Tipula *(*Vestiplex*) *biserra*: Alexander, [4]: 118;

*Tipula* (*Vestiplex*) *biserra*: Alexander and Alexander, [34]: 63.


**Diagnosis**


*Tipula* (*V.*) *biserra* can be recognized by antenna reaching before the base of the wing, gonocoxite possesing a black spine, and tergite 9 having a posterior U-shaped notch and a pair of blackened, roughened lobes as well as an inner gonostylus with a rounded dorso-lateral crest.

**Type of material examined**. **TAIWAN** • Holotype ♀; Formosa, Arisan 24.v.1917; T. Shiraki leg.; BMNH(#)246078; BMNH.

**Additional material examined. TAIWAN**—**Hsinchu Co.** • 2 ♂♂; Jian-Shih Twp Smangus; N24.5788, E121.3344; alt. 1570 m; 31 March 2011; Chen Young leg.; CO1 Barcode TIPTW744-11; CMNH 544497; dissected; CO1 Barcode TIPTW745-11; CMNH 544498; CMNH—**Hualien Co.** • 1 ♀; KuanYuan; N24.1850667, E121.3421278; alt. 2734 m; 15 May 2007; Ling-Chu Lin leg.; • 2 ♂♂; same data as for preceding; 18 May 2007; CMNH543831; CMNH543832; dissected; CMNH • 1 ♀; Sinbaiyoung at night; N24.2005, E121.4247; alt. 1500 m; 7 October 2010; Chen Young leg.; CMNH449752; NRC • 5 ♀♀; same data as for preceding; CMNH544117; CMNH544116; CMNH449587; CMNH544102; CMNH544115; CMNH • 1 ♂; SiouLin, E. end of JinMa Tunnel; N24.1794, E121.3647; alt 2456 m; 13 May 2010; Chen Young leg.; CMNH543595; CMNH—**Kaohsiung Co.** • 1 ♀; Taoyuan Township, Tengjhih Park, Mid-altitude Res. Inst.; N23.085, E120.788; alt. 1638; 4 April 2011; Chen Young leg.; CO1 Barcode TIPTW755-11; CMNH544508; CMNH • 1 ♂; Taoyuan Twp, TengZhih Station; N23.0848, E120.7877; alt. 1806 m; 23 April 2011; Zheng-De Yao; 2007-000110; CMNH • 2 ♂♂, 1 ♀; same data as for preceding; 4 May 2011; Yu-Chuan Chiu leg.; 2007-000005; 2007-000015; 2007-000007; CMNH • 2 ♂♂; same data as for preceding; 5 May 2011; 2007-000066; 2007-000077; CMNH • 1 ♀; Taoyuan Twp, TengZhih Pk Mid-altitude Res. Inst.; N23.0853, E120.7884; alt. 1638 m; 4 April 2011; Chen Young leg.; CMNH544509; CMNH • 1 ♀; TaoYuan Twp. TengJhih Pk Mid-Altitude Res. Inst.; N23.0863, E120.7771; alt. 1644 m; 15 October 2011; J. T. Tao leg.; CMNH544499; CMNH—**Nantou Co.** • 1 ♂; Ren-Ai Hsing-Pai-Young; N24.1980, E121.4326; alt. 1486 m; 27 April 2011; Han-Tzu Hsu leg.; 808000415; CMNH • 1 ♂, 2 ♀♀; same data as for preceding; Wen-Chen Chu leg.; 808000414; 0808-000413; 0808-000412; CMNH • 1 ♂; same data as for preceding; 8 June 2011; Han-Tzu Hsu leg.; 0808-0000417; CMNH • 1 ♀; same data as for preceding; 14 November 2011; CMNH454487; dissected; NRC • 1 ♂, 2 ♀♀; same data as for preceding; CMNH551714; CO1 Barcode TIPTW1035-12; CMNH576503; CO1 Barcode TIPTW1034-12; CMNH576502; CMNH • 1 ♂; Ren-Ai Township, Auwanta; N23.9494, E121.1614; alt. 1200 m; 20 April 2011; S. J. Huang leg.; CMNH544548; CMNH • 2 ♀♀; Xin-yi Township, Zizhong; N23.4748, E120.8325; alt. 2275 m; 4 May 2011; L. C. Huang leg.; CMNH575750; CMNH575738; CMNH—**Taichung Co.** • 1 ♂; Bi-LuSi, Taiwan Forest Res. Inst.; N24.2226, E121.3050; alt. 2149 m; 4 October 2010; Chen Young leg.; CMNH544104; CMNH.


**Redescription**


**Male** (Figure 30A). Body length is 13.1–16.8 mm, wing length is 17.6–21.2 mm, length of antennae is 2.0–3.1 mm. General body coloration is brownish yellow.

Head. Head yellow, occiput with brown nearly diamond-shaped spot. Rostrum dorsally yellow, laterally brownish yellow, nasus long. Palpus brown. Antenna 12-segmented, if bent backward reaching before base of the wing. Scape and pedicel yellow, flagellum coloration varies from yellow to brown with cases of bicolor appearance. Longest verticils slightly longer than corresponding flagellomeres.

*Thorax* (Figure 30B). Pronotum brown with darker median spot. Prescutum and presutural scutum with four greenish brown stripes, bordered by darker brown. Lateral stripes darker than central. Central stripes confluent anteriorly, separated by brown. Interspaces yellowish. Postsutural scutum yellowish brown with brown median vitta. Each scutal lobe with two greenish brown spots, bordered by darker brown. Scutellum and mediotergite brown with darker median line. Pleura brownish yellow. Legs with coxae and trochanters yellow. Femora yellow with tip blackened. Tibiae and tarsal segments dark brown. Claw with tooth. Wing brown, wing cells marbled with darker and clear spots (Figure 30A). Costal area brownish yellow. Brown postarcular darkening in cells br and bm, at origin of Rs and stigma area. A faint brown cloud in the middle of br and bm and distal area of bm. Vein m-cu joining dm at fork of M_3+4_, M_1+2_ about 0.1× as long as length of cell m_1_. Halter pale yellow, knob brown with margin distally pale.
Figure 30Habitus of *Tipula* (*Vestiplex*) *biserra*. (**A**). Male, lateral view. (**B**). Male thorax, dorsal view. (**C**). Female, lateral view. Scale bars: a (**A**,**C**) = 5 mm, b (**B**) = 1 mm.
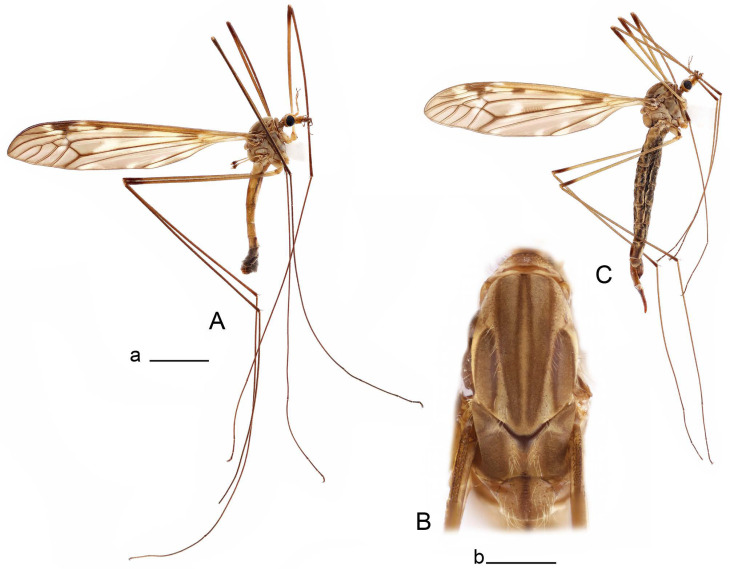



*Abdomen*. Tergites 1–5 yellow, trivittate. Dorsal line broad, lateral lines narrow, both lines are broader on tergite 5. Sternites 1–5 yellow. Remaining segments black.

*Hypopygium* (Figure 31 and Figure 32). Male hypopygium with tergite 9 fused with sternite 9 basally (Figure 31A). Tergite 9 divided along midline by pale membrane, posteriorly with U-shaped notch (Figure 31B–D). Dorsal portion brown, medially, provided with long setae on either side of midline. Ventral portion yellow with a pair of blackened, roughened lobes on either side of midline. Tip of lobe rounded, inner margin roughly serrated, base broadened. Posterolateral angle of both dorsal and lateral portions narrowed. Gonocoxite separated from sternite 9 by suture, apically produced into a strong, black spine; ventromesal portion in the shape of darkened obtuse angle (Figure 31E). Outer gonostylus flattened, base narrowed (Figure 31F). Inner gonostylus yellow, beak slender, lower beak small; both beaks separated by rounded incision (Figure 31G). Dorsal margin with dorso-lateral crest rounded. Sternite 9 with ventral lobe of A9s rounded, surface with setae (Figure 31J,K). Dorsal lobe of A9s reduced into small narrow sclerite. Adminiculum triangular in ventral view, basally dilated with lateral parts raised (Figure 31H). Medial portion distinctly protruded, base raised up, apex split with additional preapical incision (Figure 31I). Gonocoxal fragment with medial sclerites reduced, lateral sclerite plate-shaped with distal part flattened (Figure 31K). Sperm pump with central vesicle swollen (Figure 32A). Compressor apodeme flattened, with shallow emargination medially (Figure 32B). Posterior and anterior immovable apodeme narrow. Aedeagus about 3× as long as sperm pump, basally brown, becoming yellow towards apex (Figure 32A).
Figure 31Male terminalia of *Tipula* (*Vestiplex*) *biserra*. (**A**). Hypopygium, lateral view. (**B**). Hypopygium, dorsal view. (**C**). Tergite 9, dorsal view. (**D**). Tergite 9, caudal view. (**E**). Gonocoxite. (**F**). Outer gonostylus. (**G**). Left inner gonostylus, lateral view. (**H**). Adminiculum, ventral view. (**I**). Adminiculum, lateral view. (**J**). Sternite 9, tergite 9 removed, ventral view. (**K**). Sternite 9, tergite 9 and gonostyli removed, caudal view. Abbreviations: dl—dorsal lobe of A9s; ls—lateral sclerite of gonocoxal fragment; vl—ventral lobe of A9s. Scale bars: a (**A**,**B**), b (**C**–**K**) = 0.5 mm.
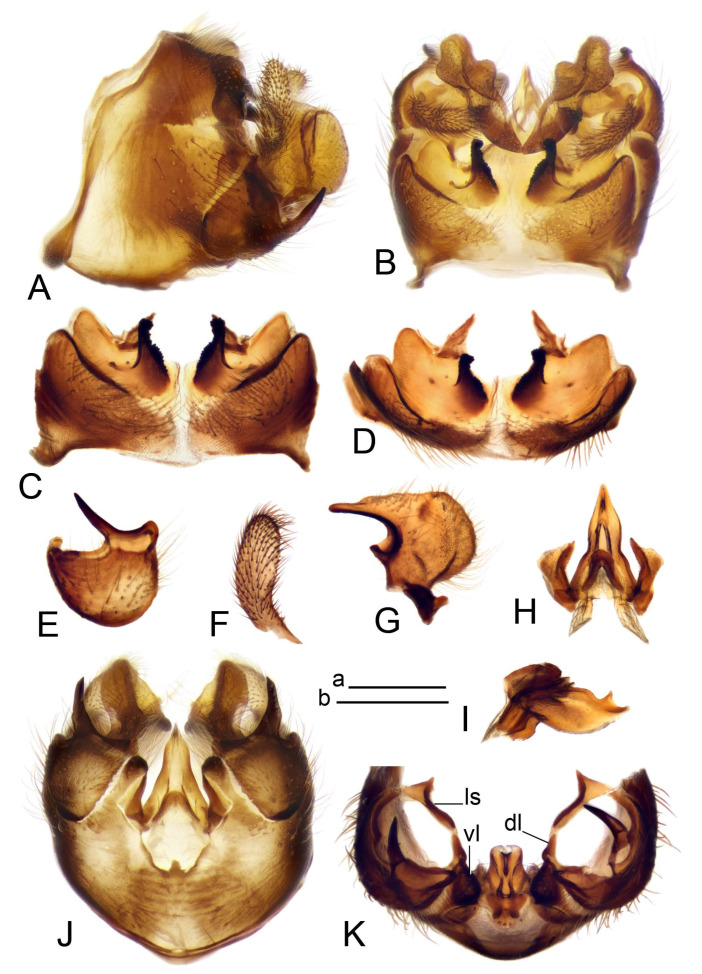

Figure 32Male terminalia of *Tipula* (*Vestiplex*) *biserra*. (**A**). Semen pump and intromittent organ, lateral view. (**B**). Semen pump, dorsal view. Scale bar: a (**A**,**B**) = 0.5 mm.
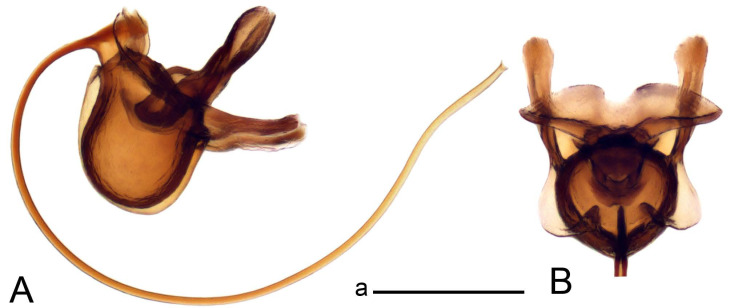

Figure 33Female terminalia of *Tipula* (*Vestiplex*) *biserra*. (**A**)**.** Ovipositor, left lateral view. (**B**). Sternite 8 with hypovalvae, ventral view. (**C**). Sternite 8 with hypovalvae, same sclerite pressed by cover glass, ventral view. (**D**). Sternite 9, dorsal view (**E**). Furca, dorsal view. (**F**). Spermatheca. Scale bars: a (**A**), b (**B**–**F**) = 0.5 mm.
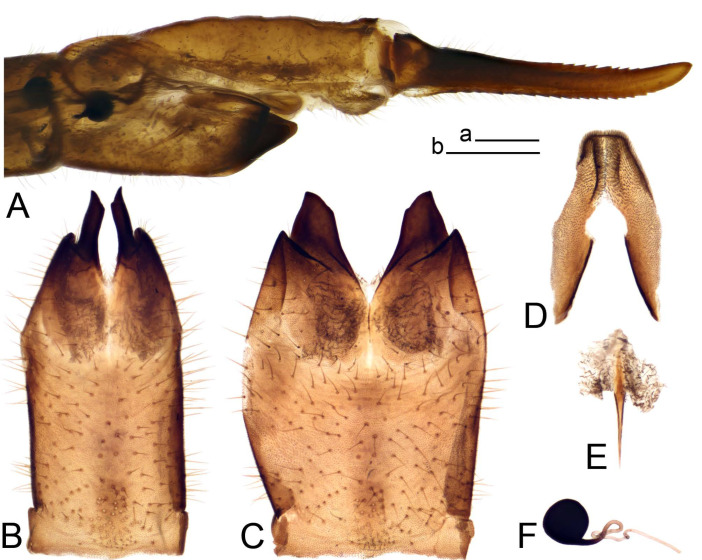



**Female** (Figure 30C). Body length is 16.5–27.1 mm, wing length is 18.5–21.7 mm, length of antenna is 2.2–3.9 mm. Female generally similar to male in body coloration.

*Ovipositor* (Figure 33). Tergite 10 shiny brown. Cercus brown, slender, longer than tergite 10, tip narrowed and slightly up-turned; ventral and dorsal margins with visible serration (Figure 33A). Hypovalva short, plate-shaped, posterior margin obliquely incised (Figure 33B,C). Sternite 9 with posterior half shovel-shaped, surface covered with short filaments, with median groove; anterior parts nearly straight, narrowed at base (Figure 33D). Furca narrow (Figure 33E). Spermatheca nearly spherical (Figure 33F).

**Distribution.** Taiwan (Figure 5A).

**Elevation range in Taiwan.** Adults were collected at altitudes ranging from 1200 m to 2700 m.

**Period of activity.** Adults are active from March through to November.

**Remarks.** *Tipula biserra* was described by Edwards [33] based on single female specimen collected by T. Shiraki in 1917 from Arisan (嘉義阿里山), Taiwan. The male association of this species is based on a current DNA gene sequence study.


***Tipula* (*Vestiplex*) *formosae* Starkevich & Young sp. nov.**


Figure 34, Figure 35, Figure 36 and Figure 37; Figure 5B


**Differentional diagnosis**


*Tipula* (*V.*) *formosae* **sp. nov**. can be recognized by 13-segmented antenna reaching beyond the base of the wing, gonocoxite apically produced into a strong, black spine, and tergite 9 having a posterior U-shaped notch and a pair of ventral, blackened, microscopically roughened lobes. The female has a slender cercus with a tip slightly up-turned and with visible serration on the ventral and dorsal margin. The hypovalva is short and plate-shaped, and the posterior margin is obliquely incised with a shallow median emargination. *Tipula* (*V.*) *formosae* **sp. nov**. is closely related to *T*. (*V*.) *grahami* Alexander, 1933 (Alexander, 1933; China, Sichuan) in having a similar shape of tergite 9, gonocoxite, inner and outer gonostyli, adminiculum, and appendages of sternite 9. The new species can be differentiated by details of the ventral armatures of tergite 9: *T.* (*V.*) *formosae* **sp. nov**. has a relatively narrower armature, while armatures of *T*. (*V*.) *grahami* are dilated at the base and relatively shorter. *Tipula* (*V.*) *formosae* **sp. nov**. can be easily separated by the 13-segmented antenna reaching the base of the wing if bent backward, while the antenna of *T*. (*V*.) *grahami* is 12-segmented, reaching the pronotum.


**Type of material**



**Holotype**


**TAIWAN** • ♂; Hualien Co.; GuanYu woods behind gas station; N24.1853, E121.3426; alt. 2355 m; 14 May 2010; Chen Young leg.; CO1 Barcode SATIP1045-10; CMNH543523; CMNH.


**Paratypes**


**TAIWAN**—**Hualien Co.** • 1 ♂; GuanYu woods behind gas station; N24.1853, E121.3426; alt. 2355 m; 14 May 2010; Chen Young leg.; CMNH455778; dissected; CMNH; • 1 ♀; Sioulin Highway 8 at 131 km mark; N24.1794, E121.3647; alt. 2039 m; 14 May 2014; Li-Tzen Shih leg.; CO1 Barcode SATIP1023-10; CMNH543501; CMNH; • 1 ♀; Xiulin Township Guan Yuan; N24.1850; E121.30424; alt. 2333 m; 25 April 2011; Wen-Chen Chu leg.; CMNH575751; CMNH—**Nantou Co.** • 2 ♂♂; Ren-Ai, Jui-Yen; N24.1063, E121.1981; alt. 2294 m; 24 October 2011; Yu-Da Chuan leg.; CO1 Barcode TIPTW1076-12; CMNH576544; 1420-000081; CMNH

• 2 ♀♀; same data as for preceding; 2 August 2011; CO1 Barcode TIPTW1124-12; CMNH551710; 1420-000077; CMNH—**Taichung Co.** • 2 ♂♂, 1 ♀; Bi-LuSi, Taiwan Forest Res. Inst.; N24.2226, E121.3050; alt. 2149 m; 12 May 2010; Chen Young leg.; CO1 Barcode SATIP1126-10; CO1 Barcode SATIP1132-10; CMNH543610;CMNH543604; dissected; CMNH543596; dissected; CMNH • 2 ♂♂, 2 ♀♀; same data as for preceding; 4 October 2010; CMNH544159; CMNH544162; CO1 Barcode TIPTW554-10; CMNH544172; CMNH544825; CMNH• 1 ♀; same data as for preceding; N24.2265, E121.3088; alt. 2149 m; 26 April 2011; Wen-Chen Chu leg.; CMNH.


**Description**


**Male** (Figure 34A). Body length is 12.3–13.7 mm, wing length is 15.8–18.4 mm, length of antenna is 3.7–5.3 mm. General body coloration is yellow.
Figure 34Habitus of *Tipula* (*Vestiplex*) *formosae*
**sp. nov**. (**A**). Male, lateral view. (**B**). Male thorax, dorsal view. (**C**). Female, lateral view. Scale bars: a (**A**,**C**) = 5 mm, b (**B**) = 1 mm.
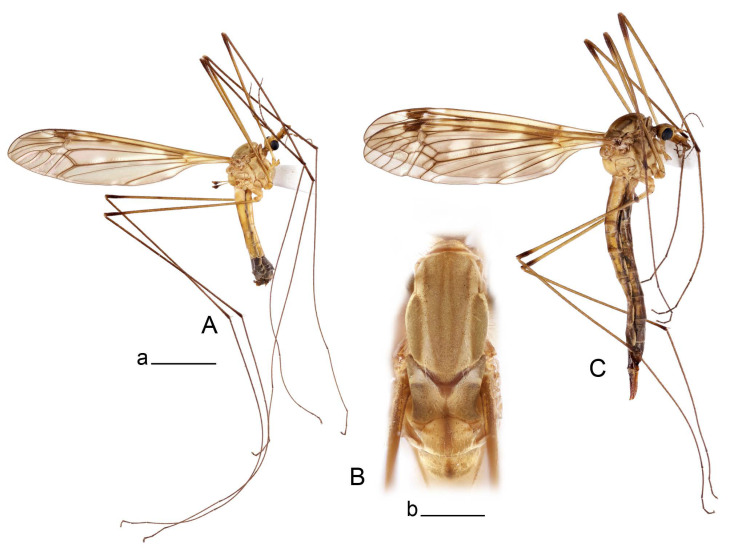



Head. Yellow, vertex and occiput with brown median vitta. Rostrum dorsally yellow, laterally yellowish brown, nasus distinct. Palpus brown. Antenna 13-segmented, if bent backward slightly reaching beyond the base of the wing. Scape and pedicel yellow, flagellum brown. Each flagellomere, except first, with basal enlargement. Apical flagellomere small, distinctly shorter than preceding flagellomere. Long verticils about the same length as corresponding flagellomeres.

*Thorax* (Figure 34B). Pronotum yellowish brown. Prescutum and presutural scutum with four greenish brown stripes, narrowly bordered by brown. Central stripes narrowly confluent anteriorly and at base, separated by greenish brown line. Interspaces yellow. Postsutural scutum greenish yellow with brown median line. Each scutal lobe with two greenish brown spots bordered by brown. Scutellum yellow, with broad brown line. Mediotergite yellow, with narrow median vitta. Pleura yellow. Legs with coxae and trochanters yellow. Femora basally yellow, becoming brownish yellow, tip darkened. Remaining segments brown. Claw with tooth. Wing clear pale brown, tinged with brown (Figure 34A). Brown postarcular darkening in cells br and bm, brown darkening at origin of Rs, and brown spot in stigma area. Vein m-cu joining dm at fork of M_3+4_, M_1+2_ about 0.35× as long as length of cell m_1_. Halter yellow, knob brown, except pale distal margin.

*Abdomen*. Tergites 1–4 yellow, with broad pale median and narrow lateral line. Sternites 1–4 yellow, remaining segments black.

*Hypopygium* (Figure 35 and Figure 36). Male hypopygium with tergite 9 fused with sternite 9 basally (Figure 35A). Tergite 9 divided along midline by pale membrane, posteriorly, with U-shaped notch (Figure 35B). Dorsal portion brown, medially, provided with long setae on either side of midline. Ventral portion yellow with a pair of blackened, microscopically roughened lobes on either side of midline. Tip of lobe rounded, base dilated. Gonocoxite separated from sternite 9 by suture, apically produced into a strong, black spine; ventromesal portion a black point (Figure 35A,C). Outer gonostylus flattened, base narrowed (Figure 35D). Inner gonostylus yellow, beak slender and blackened, lower beak small, acute; both beaks separated by rounded incision (Figure 35E). Dorsal margin with indistinct shallow emargination, dorso-lateral crest rounded. Sternite 9 with ventral lobe of A9s rounded, surface with setae (Figure 35F). Dorsal lobe of A9s reduced into small rounded sclerite. Adminiculum triangular in ventral view, basally dilated with lateral parts raised (Figure 35H). Medial portion indistinctly protruded, base raised up, apex splitted (Figure 35I). Gonocoxal fragment with medial sclerites reduced, lateral sclerite plate-shaped, expanded at base (Figure 35F). Sperm pump with central vesicle swollen (Figure 36A). Compressor apodeme flattened, with shallow emargination medially (Figure 36B). Posterior and anterior immovable apodeme narrow. Aedeagus about 3× as long as sperm pump, basally brown, becoming yellow towards apex (Figure 36A).
Figure 35Male terminalia of *Tipula* (*Vestiplex*) *formosae*
**sp. nov**. (**A**). Hypopygium, lateral view. (**B**). Tergite 9, dorsal view. (**C**). Gonocoxite. (**D**). Outer gonostylus. (**E**). Left inner gonostylus, lateral view. (**F**). Sternite 9, tergite and gonostyli removed. (**G**). Lateral sclerite of gonocoxal fragment. (**H**). Adminiculum, ventral view. (**I**). Adminiculum, lateral view. Abbreviations: dl—dorsal lobe of A9s; ls—lateral sclerite of gonocoxal fragment; vl—ventral lobe of A9s. Scale bars: a (**A**), b (**B**–**I**) = 0.5 mm.
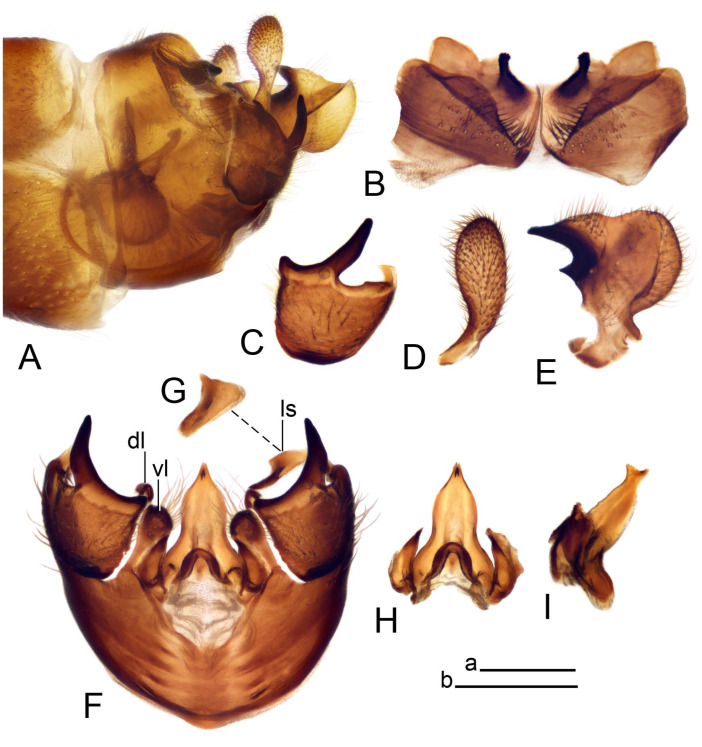

Figure 36Male terminalia of *Tipula* (*Vestiplex*) *formosae*
**sp. nov**. (**A**). Semen pump and intromittent organ, lateral view. (**B**). Semen pump, dorsal view. Scale bar: a (**A**,**B**) = 0.5 mm.
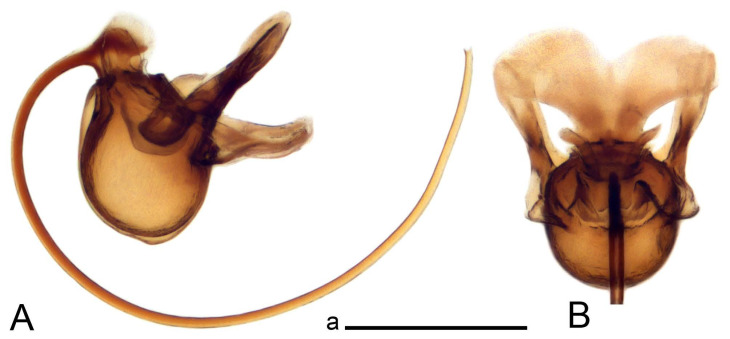



**Female** (Figure 34C). Body length is 17.8–25.3 mm, wing length is 15.5–22.0 mm, length of antenna is 2.4–3.2 mm. Female generally similar to male in body coloration.

*Ovipositor* (Figure 37). Tergite 10 shiny brown. Cercus brown, longer than tergite 10, tip narrowed and slightly up-turned; dorsal margin in distal third and ventral margin with visible serration (Figure 37A). Hypovalva short, plate-shaped, posterior margin obliquely incised with shallow median emargination. Median incision between hypovalvae basally with setae (Figure 37B,C). Sternite 9 with posterior half shovel-shaped, surface covered with short filaments, with median groove; anterior parts nearly straight (Figure 37D). Furca narrow (Figure 37E). Spermatheca nearly oval (Figure 37F).

**Etymology.** The species’ epithet is derived from an old name for Taiwan.

**Distribution.** Taiwan (Figure 5B).

**Elevation range in Taiwan.** Adults were collected at altitudes ranging from 2000 m to 2400 m.

**Period of activity.** Adults are active from April through to October.
Figure 37Female terminalia of *Tipula* (*Vestiplex*) *formosae*
**sp. nov**. (**A**). Ovipositor, sternites removed, left lateral view. (**B**). Sternite 8 with hypovalvae, ventral view. (**C**). Posterior part of sternite 8 with hypovalvae, pressed by cover glass, ventral view. (**D**). Sternite 9, dorsal view (**E**). Furca, dorsal view. (**F**). Spermatheca. Scale bars: a (**A**), b (**B**–**F**) = 0.5 mm.
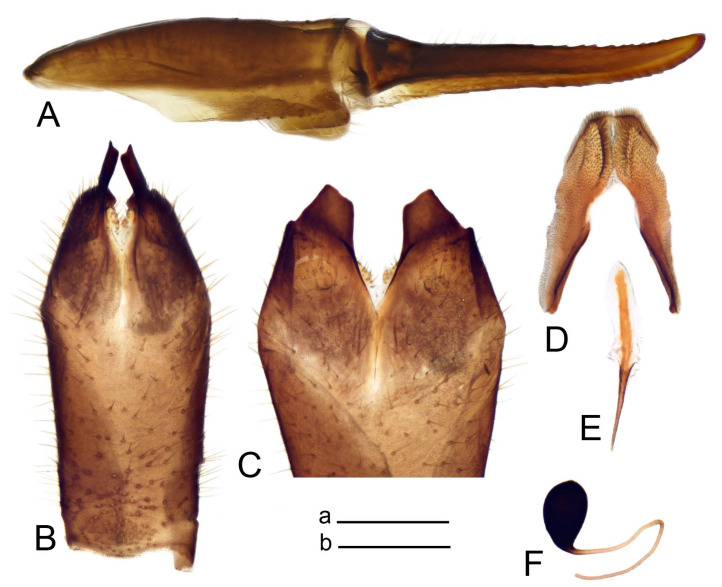

insects-14-00616-sch005_Scheme 5Scheme 5Distribution maps of *Tipula* (*Vestiplex*) crane flies in Taiwan. (**A**). *T.* (*V.*) *biserra*. (**B**). *T.* (*V.*) *formosae* **sp. nov**. Purple circle represents holotype, yellow circles represent paratypes.
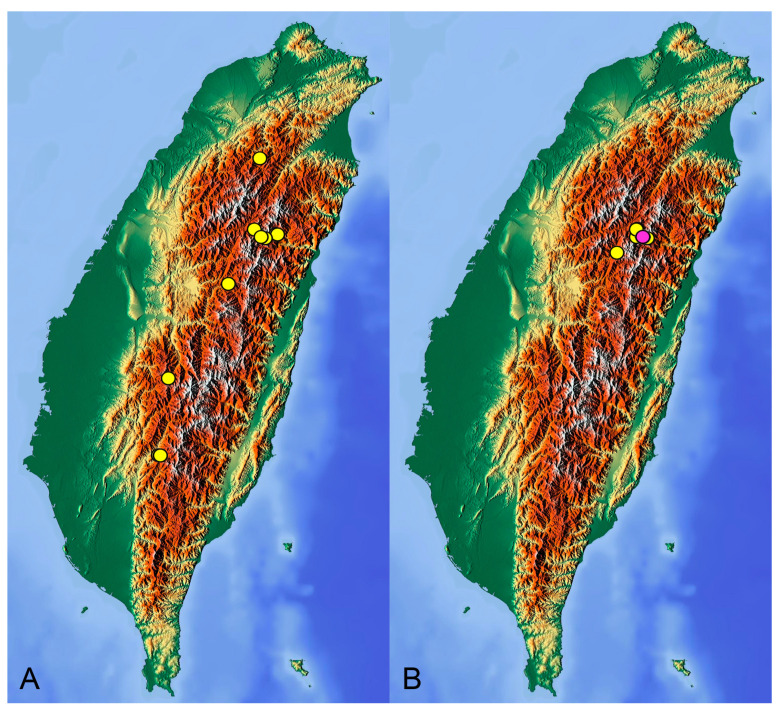




***Tipula* (*Vestiplex*) *pseudobiserra* Starkevich & Young sp. nov.**


Figure 38, Figure 39, Figure 40 and Figure 41; Figure 6A


**Differentional diagnosis**


*Tipula* (*V.*) *pseudobiserra* **sp. nov**. can be recognized by antenna reaching to before the base of the wing, gonocoxite possesing a black spine, and tergite 9 having posterior U-shaped notch and a pair of slender ventral, blackened, microscopically roughened lobes. The inner gonostylus has a specific shallow emargination dorsally, a nearly rectangular dorso-lateral crest, and blackened ridge along the medial surface. *Tipula* (*V.*) *pseudobiserra* **sp. nov**. is closely related to *T*. (*V*.) *biserra* in having a similar body coloration and shape of both male and female terminalia. The male of the new species can be separated by the details of its inner gonostylus: *T.* (*V.*) *pseudobiserra* **sp. nov**. has an inner gonostylus dorsally with a shallow emargination, a nearly rectangular dorsolateral cres, and a medial surface with a specific blackened ridge which is absent, while in *T*. (*V*.) *biserra*, the gonostylus is dorsally without emargination, the crest is rounded, and the ridge is absent. The female of the new species can be separated by the hypovalva which is wavy and slightly longer in a lateral view, while *T*. (*V*.) *biserra* has a shorter, not wavy hypovalva.


**Type of material**



**Holotype**


**TAIWAN** • ♂; Taichung Co.; Bi-LuSi, Taiwan Forest Res. Inst.; N24.2226, E121.3050; alt. 2149 m; 4 October 2010; Chen Young leg.; CO1 Barcode TIPTW490-10; CMNH544108; dissected; CMNH.


**Paratypes**


**TAIWAN**—**Hualien Co.** • 1 ♂; SiouLin, E. end of JinMa Tunnel; N24.1794, E121.3647; alt. 2456 m; 5 October 2010; Chen Young leg.; CMNH544114; CMNH • 1 ♀; same locality; N24.1773, E121.3727; alt. 2508 m; 14 June 2011; Hon-Po Chen leg.; CO1 Barcode TIPTW1137-12; CMNH551723; CMNH—**Nantou Co.** • 1 ♀; Ren-Ai Hsing-Pai-Young; N24.1980, E121.4326; alt. 1486 m; 17 April 2011; Yu-Da Chuan leg.; 0808-000617; CMNH • 1 ♂; same data as for preceding; 14 November 2011; Yu-Da Chuan leg.; CMNH448868; dissected; NRC • 2 ♂♂, 1 ♀; same data as for preceding; CO1 Barcode CMNH446999; TIPTW1139-12; CMNH551725; CMNH551724; CMNH • 1 ♀; same data as for preceding; Han-Tzu Hsu leg.; CMNH576511; CMNH—**Taichung Co.** • 3 ♂♂, 2 ♀♀; Bi-LuSi, Taiwan Forest Res. Inst.; N24.2226, E121.3050; alt. 2149 m; 4 October 2010; Chen Young leg.; CO1 Barcode TIPTW493-10; CMNH544111; CMNH544105; CMNH544113; dissected; TIPTW492-10; CMNH544110; dissected; CO1 Barcode TIPTW493-10; CMNH544111; CMNH.
Figure 38Habitus of *Tipula* (*Vestiplex*) *pseudobiserra*
**sp. nov**. (**A**). Male, lateral view. (**B**). Male thorax, dorsal view. (**C**). Female, lateral view. Scale bars: a (**A**,**C**) = 5 mm, b (**B**) = 1 mm.
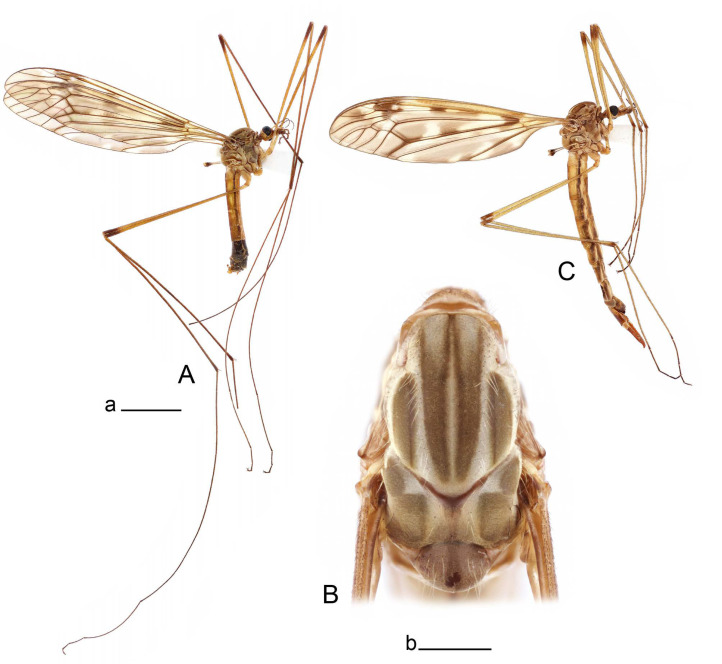




**Description.**


**Male** (Figure 38A). Body length is 14.8–17.3 mm, wing length is 18.5–23.6 mm, length of antennae is 2.4–2.9 mm. General body coloration is brownish yellow.

Head. Head yellow, occiput brownish yellow with gray spot and narrow brown vitta reaching yellow vertex. Rostrum brownish yellow with long nasus. Palpus brown except last yellowish brown segment. Antenna 12-segmented, if bent backward reaching before base of the wing. Scape and pedicel yellow, flagellum brown. The last two flagellar segments small. Flagellar segments 4–10 slightly enlarged and darkened at base. Long verticils slightly longer than corresponding flagellomeres.

*Thorax* (Figure 38B). Pronotum anteriorly brown, posterior margin yellow. Prescutum and presutural scutum with four greenish brown stripes, bordered by darker brown. Central stripes narrowly confluent anteriorly, separated by pale yellow line. Interspaces pale yellow with yellow setae. Postsutural scutum yellowish brown with light brown median vitta. Each scutal lobe with two greenish brown spots, bordered by darker brown. Scutellum brown with dark brown median line. Mediotergite brown with narrow brown vitta. Pleura brownish yellow with anepisternum and katepisternum brown, thinly dusted with yellow. Legs with coxae and trochanters yellow. Femora yellow with tip blackened. Tibiae dark brown, tip blackened. Tarsal segments dark brown. Claw with tooth. Wing brown, wing cells marbled with darker and clear spots (Figure 38A). Costal area yellow. Yellowish postarcular darkening, brown darkening in cells br and bm, at origin of Rs and stigma area. A faint brown cloud in the middle of br and bm and distal area of bm. Vein m-cu joining dm at fork of M_3+4_, M_1+2_ about 0.2× times as long as length of cell m_1_. Halter pale yellow, knob brown with margin distally pale.

*Abdomen*. Tergites 1–5 yellow, trivittate. Dorsal line broad, broader on tergite 5, lateral lines narrow. Sternites 1–4 yellow, sternite yellowish brown, posterior third blackened. Remaining segments black.

*Hypopygium* (Figure 39 and Figure 40). Male hypopygium with tergite 9 fused with sternite 9 basally (Figure 39A). Tergite 9 divided along midline by pale membrane, posteriorly with U-shaped notch (Figure 39B–D). Dorsal portion brown, medially, provided with long setae on either side of midline. Ventral portion yellow with a pair of blackened, slender, microscopically roughened lobes on either side of midline. Tip of lobe rounded, inner margin roughly serrated. Posterolateral angle of ventral portion rounded. Gonocoxite separated from sternite 9 by suture, apically produced into a strong, black spine; ventromesal portion in the shape of darkened obtuse point (Figure 39A,E,K). Outer gonostylus flattened, base narrowed (Figure 39F). Inner gonostylus yellow, beak slender, lower beak blackened, nearly triangular; both beaks separated by rounded incision (Figure 39G). Dorsal margin with broad shallow emargination, dorso-lateral crest nearly rectangular; a blackened ridge along medial surface. Sternite 9 with ventral lobe of A9s rounded, surface covered with setae (Figure 39H,J). Dorsal lobe of A9s reduced into small narrow sclerite (Figure 39K). Adminiculum triangular in ventral view, basally dilated with lateral parts raised (Figure 39H). Medial portion protruded, base raised up, apex splitted (Figure 39I). Gonocoxal fragment with medial sclerites reduced, lateral sclerite plate-shaped (Figure 39K). Sperm pump (Figure 40A–D) with central vesicle swollen. Compressor apodeme flattened, with distinct emargination medially (Figure 40C). Posterior immovable apodeme narrow, anterior apodeme flattened and. Aedeagus about 3.4× as long as sperm pump, basally brown, becoming yellow towards apex (Figure 40A). Apex terminating in three short filaments (Figure 40D).

**Female** (Figure 38C). Body length is 24.8–32.6 mm, wing length is 21–27.8 mm, length of antenna is 2.5–2.8 mm. Female generally similar to male in body coloration. Thorax with brown longitudinal stripes and darker lateral stripes.

*Ovipositor* (Figure 41). Tergite 10 shiny brown. Cercus brown, slender, longer than tergite 10, tip narrowed and slightly up-turned; ventral and dorsal margins with rough, visible serration (Figure 41A). Hypovalva short, plate-shaped, posterior margin obliquely incised, indistinctly wavy in lateral view (Figure 41B,C). Sternite 9 with posterior half shovel-shaped, surface covered with short filaments, with median groove; anterior parts nearly straight (Figure 41D). Furca elongated and narrow (Figure 41E). Spermatheca nearly oval (Figure 41F).
Figure 39Male terminalia of *Tipula* (*Vestiplex*) *pseudobiserra*
**sp. nov**. (**A**). Hypopygium, lateral view. (**B**). Hypopygium, dorsal view. (**C**). Tergite, dorsal view. (**D**). Tergite, caudal view. (**E**). Gonocoxite. (**F**). Outer gonostylus. (**G**). Left inner gonostylus, lateral view. (**H**). Adminiculum and ventral lobes of A9s, ventral view. (**I**). Adminiculum, lateral view. (**J**). Sternite 9, tergite 9 and gonostyli removed, ventral view. (**K**). Gonocoxite, distal lobe of A9s and lateral sclerite of gonocoxal fragment. (**L**). Lateral sclerite of gonocoxal fragment. Abbreviations: dl—dorsal lobe of A9s; ls—lateral sclerite of gonocoxal fragment; vl—ventral lobe of A9s. Scale bars: a (**A**), b (**B**–**L**) = 0.5 mm.
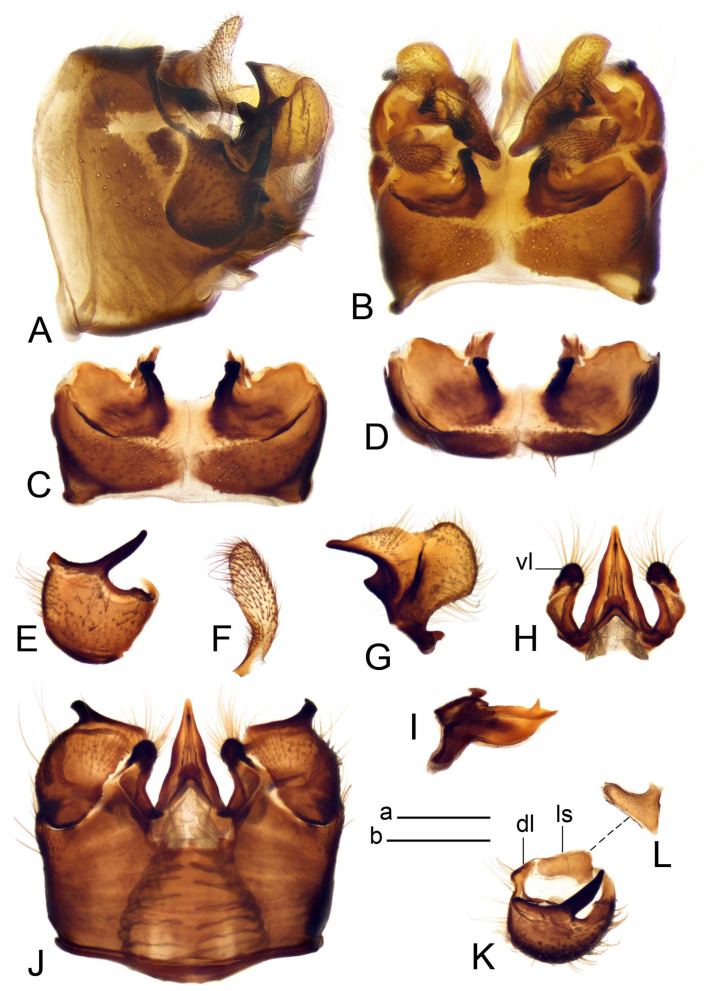

Figure 40Male terminalia of *Tipula* (*Vestiplex*) *pseudobiserra*
**sp. nov**. (**A**). Intromittent organ, lateral view. (**B**). Semen pump, lateral view. (**C**). Semen pump, dorsal view. (**D**). Apical part of intromittent organ. Scale bars: a (**A**,**C**) = 0.5 mm, a (**D**) = 0.25 mm.
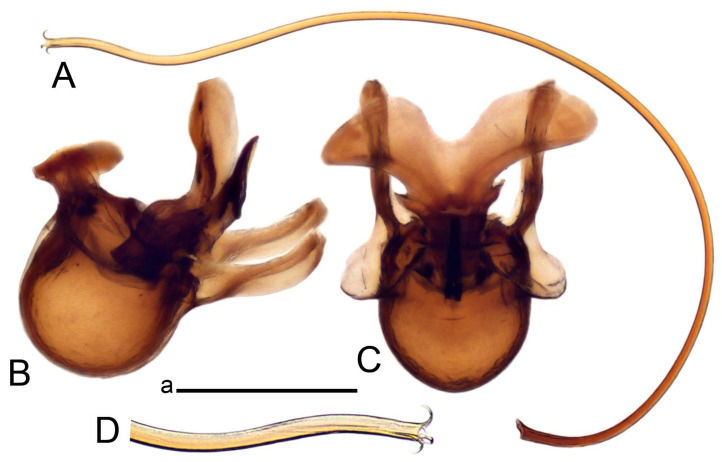

Figure 41Female terminalia of *Tipula* (*Vestiplex*) *pseudobiserra*
**sp. nov**. (**A**). Ovipositor, left lateral view. (**B**). Sternite 8 with hypovalvae, ventral view. (**C**). Sternite 8 with hypovalvae, same sclerite pressed by cover glass, ventral view. (**D**). Sternite 9, dorsal view (**E**). Furca, dorsal view. (**F**). Spermatheca. Scale bars: a (**A**), b (**B**–**F**) = 0.5 mm.
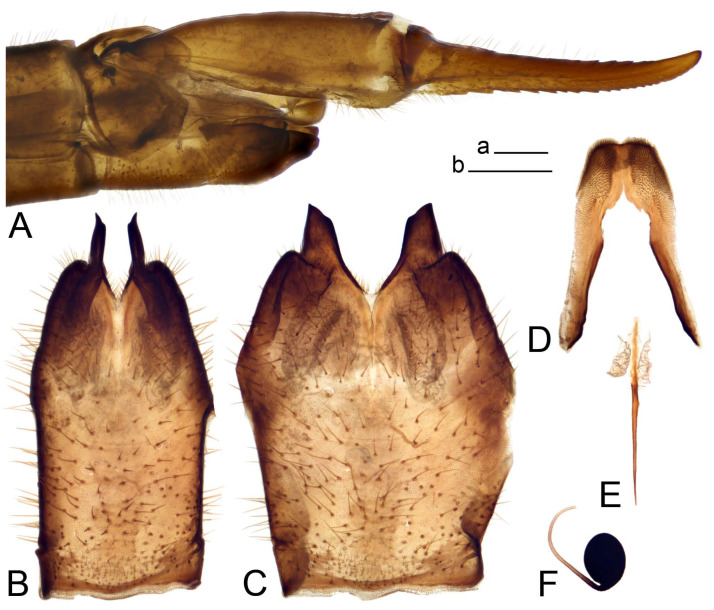


**Etymology.** The new species is named after closely related species *T.* (*V.*) *biserra* with Latin prefix “pseudo” refering to the morphological similarity.

**Distribution.** Taiwan (Figure 6A).

**Elevation range in Taiwan.** Adults were collected at altitudes ranging from 1500 m to 2500 m.

**Period of activity.** Adults are active from June through to September.


***Tipula* (*Vestiplex*) *divisotergata* species group**


The *divisotergata* species group was proposed by Savchenko [10] and the range of Chinese members was revised by Starkevich et al. [50]. Another two species have been described recently from Tibet, China [48,51]. The following species are recognized in Taiwan: *T.* (*V.*) *nestor*, *T.* (*V.*) *niitakensis*, *T.* (*V.*) *subapterogyne*, *T.* (*V.*) *taiwanica* **sp. nov**., and *T.* (*V.*) *terebrata*.

Males of the *divisotergata* species group can be recognized by an unarmed gonocoxite.

They can be also characterized by an inner gonostylus which has a flattened lower beak or with a roundly swollen area; sternite 9 with a dorsal lobe of A9s usually developed, although still reduced in *T.* (*V.*) *xinduqiaoensis* Starkevich, Men and Saldaitis ([50]: Figure 27); tergite 9 with a ventral portion usually provided by a pair of blackened, serrulatred lobes, the latter with additional points, teeth, and incisions, except in *T.* (*V.*) *xinduqiaoensis* and *T.* (*V.*) *rongtoensis* Alexander, 1963, which ventrally have membranous lobes or pale rods [52].

Females are illustrated for Taiwanese species and *T.* (*V.*) *verecunda* Alexander, 1924 ([49]: Figures 80–85). The cercus is usually smooth, without visible serration, yet still with a dorsal and ventral margin with a small serration in *T.* (*V.*) *subapterogyne*. Hypovalvae are developed and blade-shaped with a median incision between hypovalvae basally with a slightly serrated or a smooth edge and provided with setae.

***Tipula* (*Vestiplex*) *nestor*** Alexander

Figure 42, Figure 43, Figure 44 and Figure 45; Figure 6B

*Tipula* (*Vestiplex*) *nestor* Alexander, [32]: 28;

*Tipula* (*Vestiplex*) *nestor*: Alexander and Alexander, [34]: 64.


**Diagnosis**


*Tipula* (*V.*) *nestor* can be recognized by brown thoracic stripes, femora having a prepaical yellow ring and a yellow abdomen with black terminal segments. The male can also be recognized by the gonocoxite which has no outgrowth, black tergite 9 having a broad and deep V-shaped notch, slender posterolateral lobes, and black, ventral lobes bearing an obtuse point and shallow preapical incision. The female has a smooth brown cercus, blade-shaped hypovalva, and sternite 8 with lateral angles indistinctly flattened.
Figure 42Habitus of *Tipula* (*Vestiplex*) *nestor*. (**A**). Male, lateral view. (**B**). Male thorax, dorsal view. (**C**). Female, lateral view. Scale bars: a (**A**,**C**) = 5 mm, b (**B**) = 1 mm.
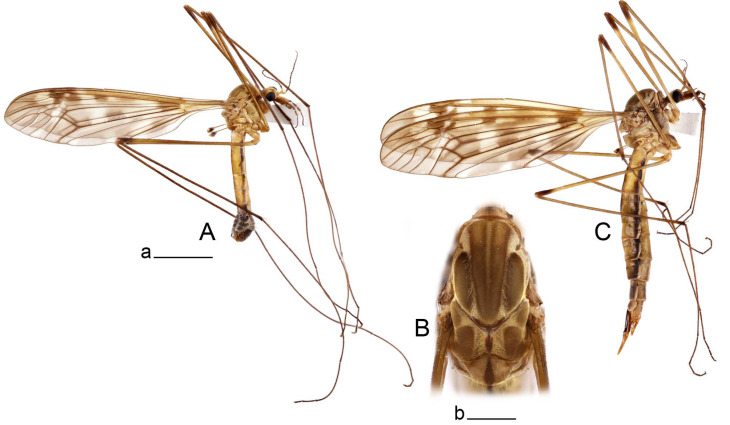



**Type of material examined. TAIWAN** • Holotype ♂; Formosa, Kanzangoe, Taito-cho Kaimosu to Kanzananbu; 19 August 1932; Teiso Esaki leg.; antenna, leg, wing, and genitalia mounted on slide; USNM; other slide at the collection of Kyushu University (Japan) with images available online [53]; the main body is possibly destroyed (the holotype is absent in USNM).

**Additional material examined. TAIWAN**—**Hualien Co.** • 1 ♂, 1 ♀; SiouLin, E. end of JinMa Tunnel; N24.1773, E121.3727; alt. 2508 m; 20 September 2011; Wen-Chen Chu leg.; CMNH216370; dissected; CMNH451352; dissected; NRC • 1 ♂; same data as for preceding; 9 August 2011; Han-Tzu Hsu leg.; 0808-000807; CMNH • 2 ♂♂, 1 ♀; 20 September 2011; Yu-Chuan Chiu leg.; CO1 Barcode TIPTW1131-12; CMNH551717; CO1 Barcode TIPTW1133-12; CMNH551719; dissected; CMNH • 1 ♂, 2 ♀♀; same data as for preceding; Wen-Chen Chu leg.; CO1 Barcode TIPTW1129-12; CMNH551715; CO1 Barcode TIPTW1132-12; CMNH551718; CMNH559004; CMNH • 2 ♀♀; SiouLin, E. end of JinMa Tunnel; N24.1794, E121.3647; alt. 2456 m; 5 October 2010; Chen Young leg.; CMNH544158; CO1 Barcode TIPTW494-10; CMNH544112; CMNH • 1 ♂; SiouLin, Hsio-Fon-Ko parking area, N24.1618, E121.2882; alt. 2999 m; 21 September 2011; Han-Tzu Hsu leg.; 0808-0011787; CMNH—**Nantou Co.** • 1 ♂; Ren-Ai HeHuan-Shan Main Peak; N24.1425, E121.2726; alt. 3398 m; 27 June 2011; Han-Tzu Hsu leg.; CMNH453333; CMNH • 1 ♂; same data as for preceding; Wen-Chen Chu leg.; 0808-000026; CMNH • 1 ♂; Ren-Ai Yuan-Feng; N24.1181, E121.2374; alt. 2761 m; 9 August 2012; HuiSheng Fang leg.; CMNH551711; CMNH • 1 ♀; same data as for preceding; 19 July 2011; Yu-Chuan Chiu leg.; 0808-001283; CMNH.


**Redescription**


**Male** (Figure 42A). Body length is 13.2–15.0 mm, wing length is 16.2–18.6 mm, length of antenna is 3.5–4.5 mm. General body coloration is yellow.

Head. Yellow, vertex and occiput with brown median line. Rostrum dorsally yellow, laterally yellowish brown, nasus distinct. Palpus brown. Antenna 13-segmented, if bent backward reaching base of the wing. Scape and pedicel yellow, flagellum brown. Each flagellomere, except first, slightly enlarged at base. Apical flagellomere small, distinctly shorter than preceding flagellomere. Long verticils about the same length as corresponding flagellomeres.

Thorax (Figure 42B). Pronotum brown, anterior margin yellow. Prescutum and presutural scutum with four brown stripes, bordered by brown. Central stripes narrowly confluent anteriorly and at base, separated by pale yellow line. Interspaces pale yellow. Postsutural scutum pale yellow with brown median line. Each scutal lobe with two brown spots bordered by brown. Scutellum and mediotergite yellow, with broad brown median line. Pleura yellow. Legs with coxae and trochanters yellow. Femora basally yellow, becoming brownish yellow towards midline, tip blackened, preceeded by yellow subterminal ring. Remaining segments brown. Claw with tooth. Wing brown, wing cells marbled with darker and clear spots (Figure 42A). Brown postarcular darkening in cells br and bm, at origin of Rs and stigma area. A faint brown cloud in the middle of br and bm and distal area of bm. Vein m-cu joining dm at fork of M_3+4_, M_1+2_ about 0.3× as long as length of cell m_1_. Halter yellow, knob basally brown, distally yellowish brown.

Abdomen. Tergites 1–5 yellow, with broad brown median and narrow lateral line. Sternites 1–5 yellow, remaining segments brownish black.

Hypopygium (Figure 43 and Figure 44). Male hypopygium with tergite 9 fused with sternite 9 basally (Figure 43A). Tergite 9 divided along midline by pale membrane, posteriorly with broad and deep V-shaped notch (Figure 43B,C). Dorsal portion occupying the tergal area, blackish, covered with setae, posterolateral angle extended into slender lobe. Ventral portion terminating into slender, blackened, microscopically roughened lobes; the inner margin of lobe with black obtuse point and shallow preapical incision. Gonocoxite separated from sternite 9 by suture, unarmed, obtuse at apex (Figure 43A,D). Outer gonostylus blade-shaped (Figure 43E). Inner gonostylus nearly crescent plate, terminating into a blackened, short upper beak, lower beak flattened yellow plate (Figure 43F). Dorsal margin round, edge basally bent outward. Sternite 9 with ventral lobe of A9s narrow, oblong, surface covered with setae (Figure 43G). Dorsal lobe of A9s short, finger-shaped, the tip with setae (Figure 43H). Adminiculum nearly triangular in both ventral and lateral views, medial protrusion indistinct, apex slightly curved (Figure 43I,J). Gonocoxal fragment with medial sclerites narrow, fused at base, nearly V-shaped; lateral sclerite a flattened uniform plate (Figure 43K). Sperm pump with central vesicle swollen (Figure 44A). Compressor apodeme flattened, with median incision (Figure 44B). Posterior immovable apodeme narrow, anterior immovable apodeme flattened. Aedeagus about 4.3× as long as sperm pump, brown, distally becoming yellow with apex funnel-shaped (Figure 44C,D).

**Female** (Figure 42C). Body length is 21.0–24.7 mm, wing length is 18.3–21.1 mm, length of antenna is 2.5–3.0 mm.

Ovipositor (Figure 45). Tergite 10 shiny black. Cercus brown, nearly the same length as tergite 10, tip round; ventral margin smooth, without visible serration (Figure 45A). Hypovalva extending before base of cercus. Sternite 8 with hypovalva developed, blade-shaped (Figure 45B,C). Lateral angle indistinctly flattened. Median incision between hypovalvae basally with setae. Sternite 9 wedge-shapedwith posterior margin narrowing (Figure 45D). Furca flattened (Figure 45D). Spermatheca pear-shaped (Figure 45E).
Figure 43Male terminalia of *Tipula* (*Vestiplex*) *nestor*. (**A**). Hypopygium, lateral view. (**B**). Hypopygium, dorsal view. (**C**). Tergite 9, dorsal view. (**D**). Gonocoxite (**E**). Outer gonostylus. (**F**). Left inner gonostylus, lateral view. (**G**). Ventral lobe of A9s. (**H**). Dorsal lobe of A9s. (**I**). Adminiculum, ventral view. (**J**). Adminiculum, lateral view. (**K**). Gonocoxal fragment. Abbreviations: ls—lateral sclerite of gonocoxal fragment; ms—medial sclerite of gonocoxal fragment. Scale bars: a (**A**), b (**B**–**K**) = 0.5 mm.
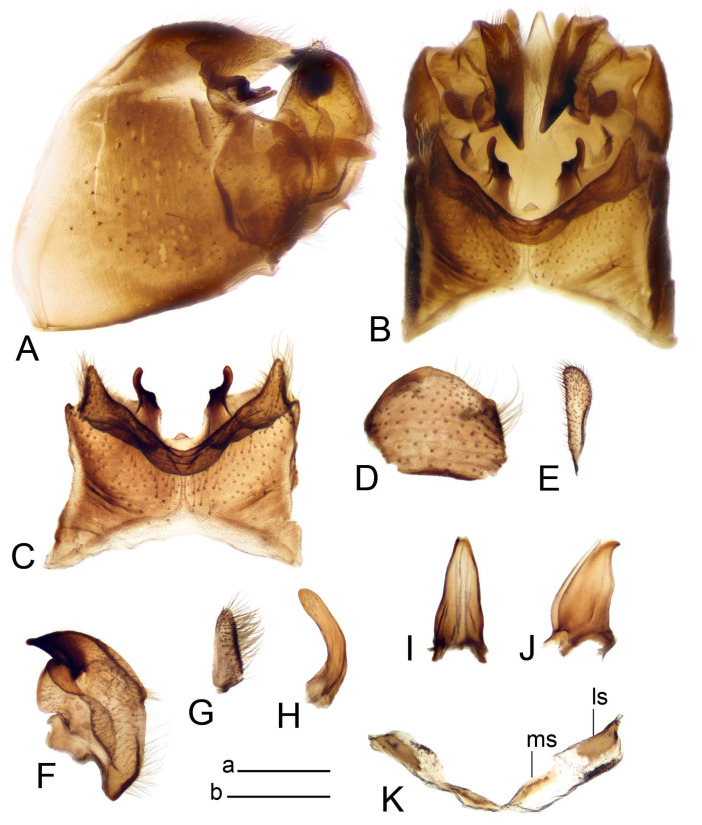

Figure 44Male terminalia of *Tipula* (*Vestiplex*) *nestor*. (**A**). Semen pump, lateral view. (**B**). Semen pump, dorsal view. (**C**). Intromittent organ, lateral view. (**D**). Apical part of intromittent organ. Scale bar: a (**A**–**C**) = 0.5 mm, a (**D**) = 0.25 mm.
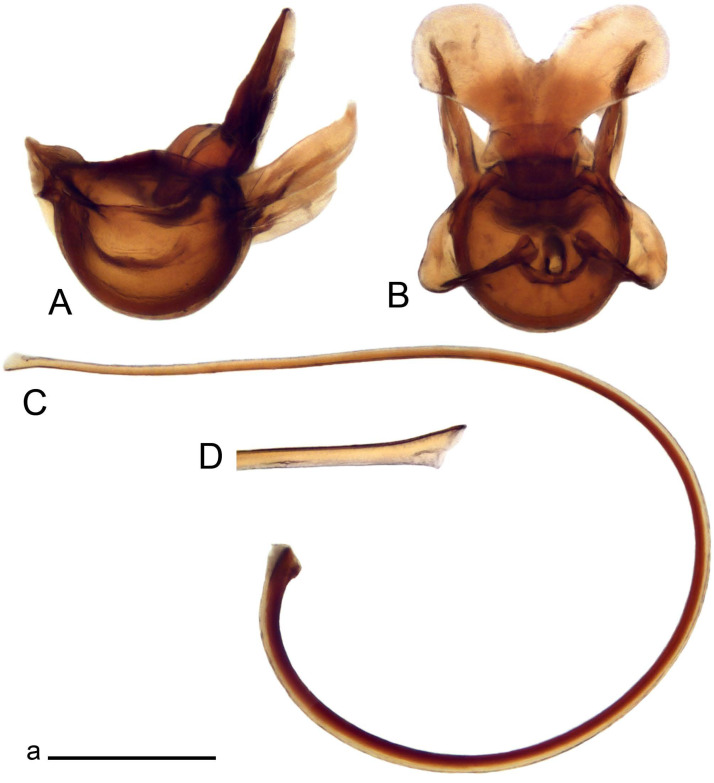

Figure 45Female terminalia of *Tipula* (*Vestiplex*) *nestor*. (**A**). Ovipositor, left lateral view. (**B**). Sternite 8 with hypovalvae, ventral view. (**C**). Sternite 8 with hypovalvae, same sclerite pressed by cover glass, ventral view. (**D**). Sternite 9 and furca, dorsal view (**E**). Spermatheca. Scale bars: a (**A**), b (**B**–**E**) = 0.5 mm.
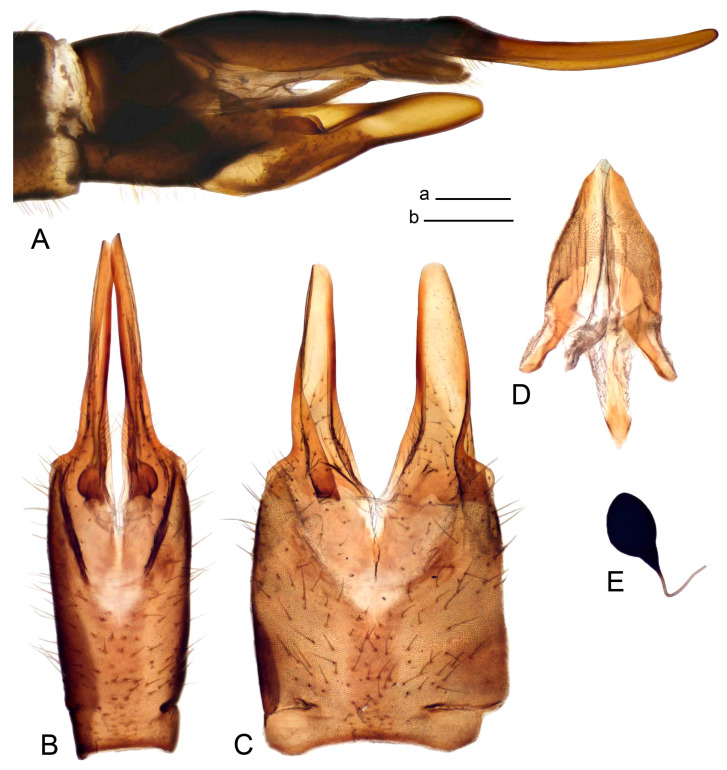



**Distribution.** Taiwan (Figure 6B), China (Yunnan).

**Elevation range in Taiwan.** Adults were collected at altitudes ranging from 2500 m to 3400 m.

**Period of activity.** Adults are active from June through to September.

**Remarks.** *Tipula* (*V.*) *nestor* was described by Alexander [32] based on single male specimen collected by T. Esaki in 1932 from Kanzagoe, Taito-cho (Taitung County). The female asssociation of this species is based on a current DNA gene sequence study.
insects-14-00616-sch006_Scheme 6Scheme 6Distribution maps of *Tipula* (*Vestiplex*) crane flies in Taiwan. (**A**). *T.* (*V.*) *pseudobiserra* **sp. nov**. Purple circle represents holotype, yellow circles represent paratypes. (**B**). *T.* (*V.*) *nestor*.
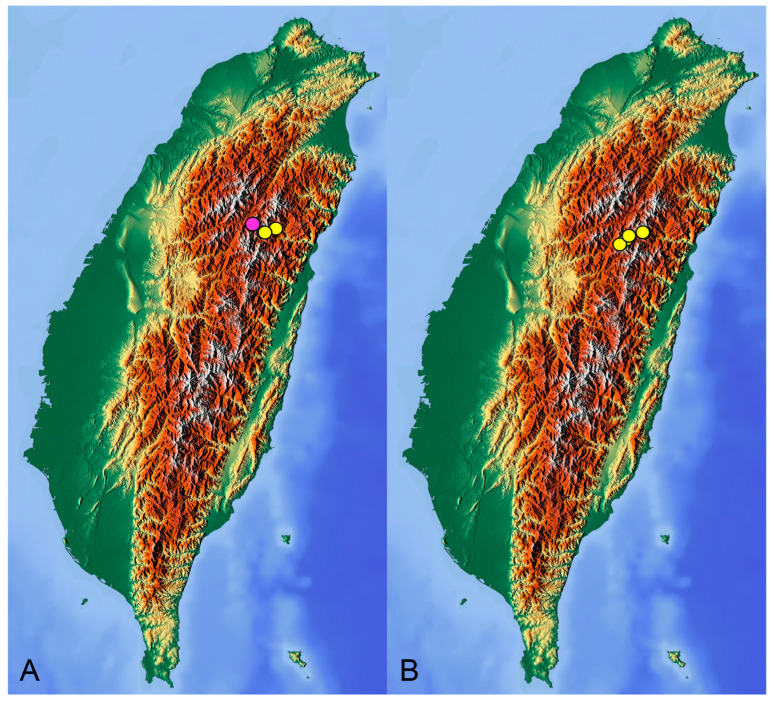




***Tipula* (*Vestiplex*) *niitakensis* Alexander**


Figure 46, Figure 47, Figure 48 and Figure 49; Figure 7A

*Tipula* (*Pterelachisus*) *niitakensis* Alexander, [39]: 104;

*Tipula* (*Pterelachisus*) *niitakensis*: Alexander and Alexander, [34]: 52;

*Tipula* (*Vestiplex*) *niitakensis* Alexander, 1938 **subgen. comb. nov.**


**Diagnosis**


*Tipula* (*V.*) *niitakensis* can be recognized by its thoracic greenish brown stripes and yellow abdomen with brown terminal segments. The male can be also recognized by the gonocoxite which has no outgrowth, the brown tergite 9 having a broad and deep V-shaped notch, the extended posterolateral lobes, and black, crescent-shaped ventral lobes. The female is brachypterous, with non-functional short wings, an ovipositor with a smooth cercus and blade-shaped hypovalva.
Figure 46Habitus of *Tipula* (*Vestiplex*) *niitakensis*. (**A**). Male, lateral view. (**B**). Male thorax, dorsal view. (**C**). Female, lateral view. Scale bars: a (**A**,**C**) = 5 mm, b (**B**) = 1 mm.
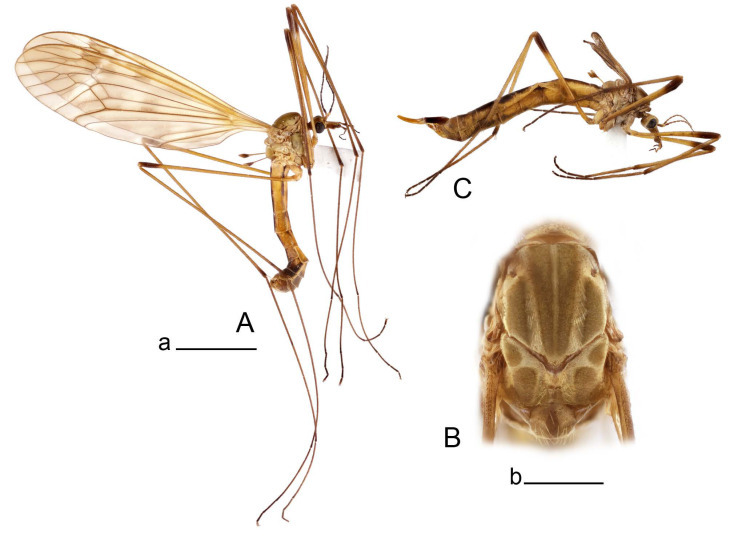



**Type of material examined. TAIWAN** • Holotype ♂; Formosa, Mount Niitaka; 25 August 1936; Takahashii leg.; USNM.

**Additional material examined. TAIWAN**—**Hualien Co.** • 1 ♂; Sioulin Township, Sung-Hsueh-Lou; N24.1539, E121.2839; alt. 3026 m; 9 August 2011; Yu-Chuan Chiu leg.; 0808-000399; CMNH—**Nantou Co.** • 1 ♂; Ren-Ai Chi-Lai Mountain Entrance; N24.1388, E121.2874; alt. 3110 m; 22 July 2011; Wen-Chen Chu leg.; 0808-000289; CMNH • 1 ♂; same data as for preceding; 10 August 2011; 0808-000292; CMNH • 1 ♀; Ren-Ai He Huan-Shan-Chuang; N24.1425, E121.2840; alt 3138 m; 18 May 2011; Wen-Chen Chu leg.; 0808-000373; CMNH • 1 ♂; Ren-Ai HeHuan-Shan Main Peak; N24.1425, E121.2726; alt. 3398 m; 8 June 2011; Wen-Chen Chu leg.; 0808-000082; CMNH • 3 ♂♂; same data as for preceding; 27 June 2011; Han-Tzu Hsu leg.; 0808-002084; 0808-002163; CO1 Barcode TIPTW1107-12; CMNH576575; CMNH • 2 ♂♂, 1 ♀; same data as for preceding; Wen-Chen Chu leg.; 0808-002085; CMNH453121; 0808-000023; CMNH • 1 ♀; same data as for preceding; 6 July 2011; Wen-Chen Chu leg.; 0808-000032 • 2 ♂♂, 3 ♀♀; same data as for preceding; 13 July 2011; Wen-Chen Chu leg.; 0808-000100; 0808-000105; 0808-000098; 0808-000099; 0808-000037; CMNH • 1 ♂, 1 ♀; same data as for preceding; 14 July 2011; Wen-Chen Chu leg.; 0808-000215; 0808-002143; CMNH • 1 ♂; same data as for preceding; 20 July 2011; Han-Tzu Hsu leg.; 0808-000040; CMNH • 1 ♂, 2 ♀♀; same data as for preceding; Wen-Chen Chu leg.; 0808-000038; 0808-000047; 0808-000044; CMNH • 9 ♂♂, 4 ♀♀; same data as for preceding; 22 July 2011; Han-Tzu Hsu leg.; 0808-002127; 0808-002146; 0808-002113; 0808-002092; CO1 Barcode TIPTW1106-12; CMNH576574; 0808-002144; TIPTW1108-12; CMNH576576; CMNH569802; dissected; 0808-002159; CO1 Barcode TIPTW1048-12; CMNH576516; 0808-002160; CMNH450365; TIPTW1046-12; CMNH576514; CMNH • 1 ♂; same data as for preceding; HuiSheng Fang leg.; CMNH409336; dissected; NRC • 3 ♂♂, 2 ♀♀; same data as for preceding; 0808-002130; 0808-000238; 0808-002095; 0808-002162; 0808-002145; CMNH • 1 ♂, 1 ♀; same data as for preceding; Wen-Chen Chu leg.; 0808-002109; 0808-002120; CMNH • 3 ♂♂, 3 ♀♀; same data as for preceding; 28 July 2011; Han-Tzu Hsu leg.; 0808-000115; 0808-000132; 0808-000133; 0808-000120; 0808-000112; 0808-000117; CMNH • 4 ♂♂; same data as for preceding same data as for preceding; Wen-Chen Chu leg.; 0808-000129; 0808-000126; 0808-000130; 0808-000123; CMNH • 1 ♂; same data as for preceding; 5 August 2011; Wen-Chen Chu leg.; CMNH576560; CMNH • 3 ♂♂, 1 ♀; same data as for preceding; 11 August 2011; Wen-Chen Chu leg.; 0808-000143; 0808-000144; 0808-000142; 0808-000139; CMNH • 1 ♂; same data as for preceding; Wen-Chen Chu leg.; CO1 Barcode TIPTW1091-12; CMNH576559; CMNH • 1 ♂; same data as for preceding; 7 September 2011; Wen-Chen Chu leg.; 0808-000067; CMNH • 2 ♂♂, 1 ♀; Ren-Ai HeHuan-Shan North Peak; N24.1815, E121.2816; alt. 3424 m; 14 July 2011; Han-Tzu Hsu leg., CMNH453587; CMNH414936; dissected; CMNH431726; NRC • 2 ♂♂, 5 ♀♀; same data as for preceding; 0808-000248; CMNH451694; CO1 Barcode TIPTW1047-12; CMNH576515; dissected; CMNH455510; 0808-000235; 0808-000213; 0808-000247; CMNH • 1 ♂; same data as for preceding; Wen-Chen Chu leg.; CMNH569310; NRC • 6 ♂♂, 4 ♀♀; same data as for preceding; Wen-Chen Chu leg.; CMNH572524; 0808-000249; 0808-000214; 0808-000254; 0808-000242; 0808-000221; CMNH407300; 0808-000251; 0808-000263; 0808-000255; CMNH • 1 ♂; same data as for preceding; 22 July 2011; Han-Tzu Hsu leg.; 0808-000270; CMNH • 1 ♂; same data as for preceding; 27 July 2011; Wen-Chen Chu leg.; 0808-000264; CMNH • 1 ♂; Ren-Ai Twp, He-Huan-Shan W. Peak; N24.1777, E121.2445; alt. 3415 m; 27 July 2011; Wen-Chen Chu leg.; CMNH0808-000206; CMNH • 1 ♂; Ren-Ai, Wuling rest area; N24.1376, E121.2757; alt. 3276 m; 19 July 2011; HuiSheng Fang leg.; 0808-000294; CMNH.


**Redescription**


**Male** (Figure 46A). Body length is 11.5–13.9 mm, wing length is 13.2–16.4 mm, length of antenna is 3.5–5.0 mm. General body coloration is brownish yellow.

Head. Yellow, vertex and occiput thin with blackish median line. Rostrum yellow, nasus conspicuous. Palpus brown. Antenna 13-segmented, if bent backward reaching base of the wing. Scape and pedicel yellow, remaining segments blackish brown. Each flagellomere, except first, with indistinct basal enlargement. Apical flagellomere small, distinctly shorter than preceding flagellomere. Long verticils nearly equal to length of corresponding flagellomeres.

Thorax (Figure 46B). Pronotum yellow with brown median spot. Prescutum and presutural scutum with four greenish brown stripes bordered by darker brown. Central stripes confluent anteriorly and posteriorly, separated by yellow vitta. Interspaces bright yellow. Postsutural scutum yellow, with darker median line. Each scutal lobe with two greenish brown spots. Scutellum brownish yellow with broad brown line. Mediotergite brownish yellow with narrow vitta. Pleura ochreous yellow. Legs with coxae and trochanters yellow. Femora yellow, darkened at tip. Tibiae yellow, distally becoming brown. Tarsal segments brown. Claw without tooth. Wing brownish yellow, variegated with pale brown clouds (Figure 46A). Costal and subcostal area yellow. Small brown spot at origin of Rs and stigma area. Vein m-cu joining dm at fork of M_3+4_, M_1+2_ about 0.1× as long as length of cell m_1_. Halter yellow, knob basally brown.

Abdomen. Tergites 1–6 yellow, tergites with broad blackish median and narrow lateral lines. Sternites 1–7 yellow with broad line. Remaining segments brown.
Figure 47Male terminalia of *Tipula* (*Vestiplex*) *niitakensis*. (**A**). Hypopygium, lateral view. (**B**). Hypopygium, dorsal view. (**C**). Tergite 9, dorsal view. (**D**). Gonocoxite (**E**). Outer gonostylus. (**F**). Left inner gonostylus, lateral view. (**G**). Ventral lobe of A9s. (**H**). Dorsal lobe of A9s. (**I**). Adminiculum, ventral view. (**J**). Adminiculum, lateral view. (**K**). Gonocoxal fragment. Abbreviations: ls—lateral sclerite of gonocoxal fragment; ms—medial sclerite of gonocoxal fragment. Scale bars: a (**A**), b (**B**–**K**) = 0.5 mm.
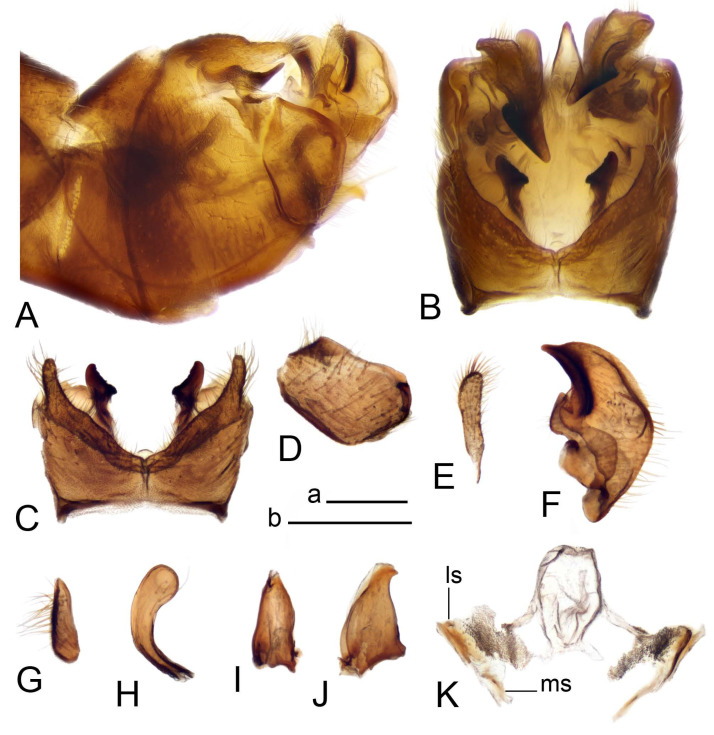



Hypopygium (Figure 47 and Figure 48). Male hypopygium with tergite 9 fused with sternite 9 except on posterior corners (Figure 47A). Tergite 9 divided along midline by pale membrane, posteriorly with broad and deep V-shaped notch (Figure 47B,C). Dorsal portion brown, narrow at midlength, covered with setae, posterolateral angle extended into slender lobe. Ventral portion terminating into blackened, microscopically roughened crescent-shaped lobes. Gonocoxite separated from sternite 9 by suture, unarmed (Figure 47A,D). Outer gonostylus finger-shaped (Figure 47E). Inner gonostylus nearly crescent plate, terminating into a blackened, short upper beak, lower beak flattened yellow plate (Figure 47F). Dorsal margin with narrow band along its margin; ventral edge basally bent outward. Sternite 9 with ventral lobe of A9s oblong, narrowed at tip, surface covered with setae (Figure 47G). Dorsal lobe of A9s curved, base narrowed, the tip roundly dilated, covered with setae (Figure 47H). Adminiculum nearly triangular in both ventral and lateral views, medial protrusion indistinct, apex slightly curved (Figure 47I,J). Gonocoxal fragment membranuous, with medial sclerites narrow, separated, lateral sclerite, flattened uniform (Figure 47K). Sperm pump with central vesicle spherical (Figure 48A). Compressor apodeme flattened, with shallow median incision (Figure 48B). Posterior and anterior immovable apodemes narrow. Aedeagus about 3.5× as long as sperm pump (Figure 48A).
Figure 48Male terminalia of *Tipula* (*Vestiplex*) *niitakensis*. (**A**). Semen pump and intromittent organ, lateral view. (**B**). Semen pump, dorsal view. Scale bar: a (**A**,**B**) = 0.5 mm.
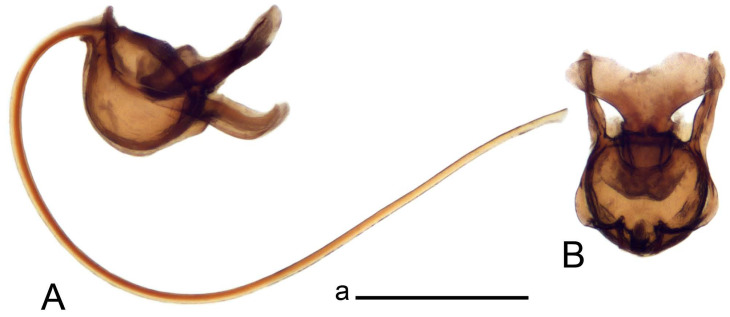



**Female** (Figure 46C). Body length is 15.0–19.1 mm, wing length is 3.8–5.7 mm, length of antenna is 2.2–3.8 mm. Legs short and stout, with a length about half those of males. Their wings are brachypterous, short and narrow, with a rudimentary trace of venation.

Ovipositor (Figure 49). Tergite 10 shiny dark brown. Cercus brown, about the same length as tergite 10, tip round; ventral margin without visible serration (Figure 49A). Hypovalva extending to just beneath base of cercus. Sternite 8 with hypovalva developed, blade-shaped (Figure 49B,C). Lateral angle sloping. Median incision between hypovalvae basally serrated, provided with setae. Sternite 9 nearly wedge-shaped with posterior margin acute; anterior parts acute at base (Figure 49D). Furca flattened, membranuous except darkened base (Figure 49D). Spermatheca nearly oval (Figure 49E).

**Distribution.** Taiwan (Figure 7A).

**Elevation range in Taiwan.** Adults were collected at altitudes ranging from 3000 m to 3400 m.

**Period of activity.** Adults are active from July through to August.

**Remarks.***Tipula niitakensis* was described by Alexander as a member of the former subgenus *T.* (*Oreomyza*) in 1938 based on a single male specimen collected by R. Takahashi in 1936 from Mount Niitaka, recently known as Yu Shan (玉山). The female asssociation of this species is confirmed in a current DNA gene sequence study. The species was placed in subgenus *T.* (*Vestiplex*) according to the morphology of male and female terminalia and DNA.
Figure 49Female terminalia of *Tipula* (*Vestiplex*) *niitakensis*. (**A**). Ovipositor, left lateral view. (**B**). Sternite 8 with hypovalvae, ventral view. (**C**). Sternite 8 with hypovalvae, same sclerite pressed by cover glass, ventral view. (**D**). Sternite 9 and furca, dorsal view (**E**). Spermatheca. Scale bars: a (**A**), b (**B**–**E**) = 0.5 mm.
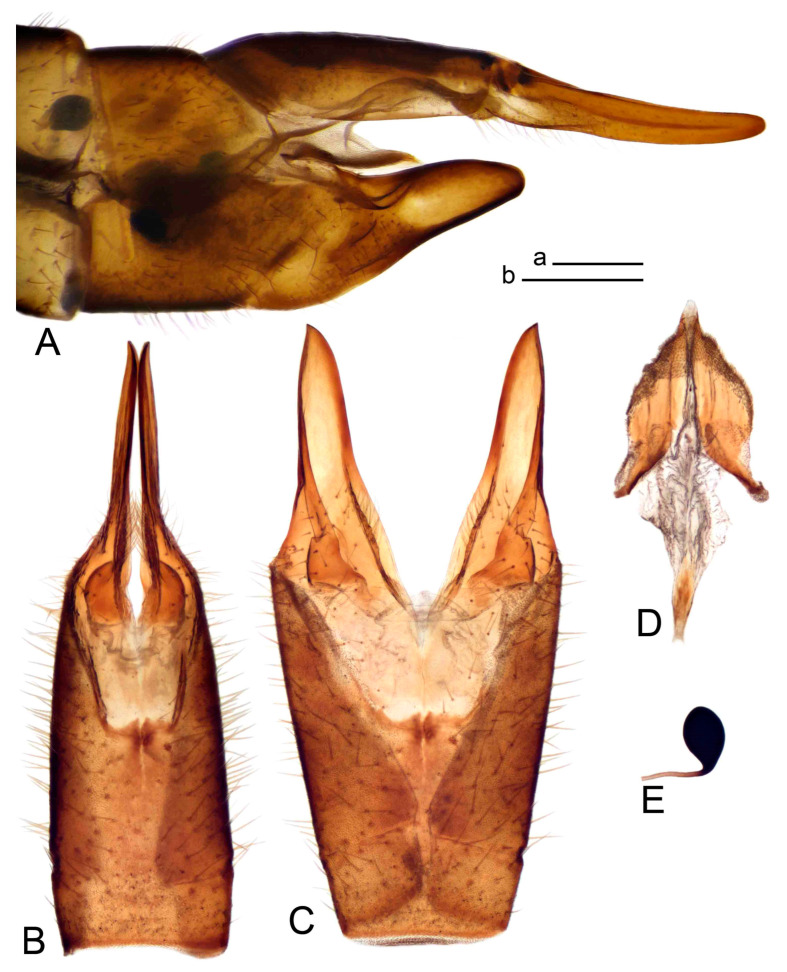




***Tipula* (*Vestiplex*) *subapterogyne* Alexander**


Figure 50, Figure 51, Figure 52 and Figure 53; Figure 7B

*Tipula subapterogyne* Alexander, [54]: 266;

*Tipula *(*Vestiplex*) *subapterogyne*: Alexander, [4]: 118; Alexander and Alexander, [34]: 65.


**Diagnosis**


*Tipula* (*V.*) *subapterogyne* can be recognized by the gonocoxite having no outgrowth, and the posterior margin of tergite 9 with a deep and broad U-shaped notch. The female is flightless with non-functional short wings. The ovipositor has a stocky cercus, with the ventral and dorsal margin serrated.

**Type of material examined**. **TAIWAN** • Holotype ♂; Formosa, Noko; 11 May 1919; T. Shiraki leg.; USNM; • Paratypes 1 ♂, 1 ♀; same data as for preceding; USNM.

**Additional material examined. TAIWAN**—**Hualien Co.** • 2 ♂♂, 2 ♀♀; Xiulin Township Chilai Mts; N24.1388; E121.2874; alt. 3110 m; 26 May 2011; Wen-Chen Chu leg.; CMNH575730; CO1 Barcode TIPTW926-11; CMNH575732; CMNH575733; CMNH575731; CMNH—**Nantou Co.** • 1 ♂, 2 ♀♀; Yuanfeng; [N24.12083, E121.24333]; IV/10---V/8/2007; Malaise trap (KCN); C. S. Lin & W. T. Yang leg.; CMNH544549; CMNH544578; CMNH544577; CMNH • 16 ♂♂; Ren-Ai Endemic Species Res. Centr.; N24.1813, E121.2866; alt. 2939 m; 13 May 2010; Chen Young leg.; CMNH432295; CMNH452623; CMNH413573; dissected; CMNH543496; CMNH • 19 ♂♂; same data as for preceding; 14 May 2010; CMNH455555; CMNH543518; CO1 Barcode SATIP1035-10; CMNH543513; CMNH543515; CO1 Barcode SATIP1038-10; CMNH543516; CMNH • 5 ♂♂; same data as for preceding; CMNH427190; dissected; CMNH450151; CMNH449673; CMNH448119; CMNH449046; NRC • 1 ♀; Ren-Ai He Huan-Shan-Chuang; N24.1425, E121.2840; alt. 3138 m; 11 May 2011; Wen-Chen Chu leg.; 0808-002263; CMNH • 2 ♂♂, 2 ♀♀; Ren-Ai He Huan-Shan-Chuang; N24.1425, E121.2840; alt. 3138 m; 18 May 2011; Wen-Chen Chu leg.; 0808-000372; 0808-000349; 0808-000348; 0808-000371; CMNH • 2 ♂♂; Ren-Ai He Huan-Shan-Chuang; N24.1425, E121.2840; alt. 3138 m; 19 May 2011; Wen-Chen Chu leg.; 0808-000370; 0808-000351; CMNH • 1 ♀; same data as for preceding; Yu-Da Chuan leg., 0808-000352; CMNH • 3 ♂♂, 1 ♀; Ren-Ai HeHuan Mts. Jui-Yen CK; N24.1456, E121.2838; alt. 3256 m; 12 May 2009; H. S. Fang; CMNH; CMNH544156; CMNH544120; CMNH544119; CMNH544118; CMNH • 2 ♀♀; Ren-Ai HeHuan-Shan Main Peak; N24.1423, E121.2715; alt. 3416 m; 14 May 2010; Chen Young leg,; CO1 Barcode SATIP1041-10; CMNH543519; CO1 Barcode SATIP1042-10; CMNH543520; CMNH • 1 ♀; same data as for preceding; 19 May 2010; Wen-Chen Chu leg.; CO1 Barcode TIPTW921-11; CMNH575727; CMNH • 2 ♀♀; same data as for preceding; N24.1423, E121.2715; alt. 3416 m; Chen Young leg.; SATIP1041-10; CMNH543519; SATIP1042-10; CMNH543520; CMNH • 2 ♂♂, 3 ♀♀; same data as for preceding; 19 May 2011; Wen-Chen Chu leg.; CMNH575726; CO1 Barcode TIPTW921-11; CMNH575727; CMNH575728; CMNH575729; CMNH575735; CMNH • 1 ♂; same data as for preceding; alt. 3398 m; CMNH575734; CMNH • 1 ♂; same data as for preceding; 3 June 2011; alt. 3398 m; 0808-000090; CMNH • 1 ♂; same data as for preceding; 8 June 2011; alt. 3398 m; 0808-000088; CMNH • 2 ♂♂, 1 ♀; N24.1425, E121.2726; 19 May 2011; Wen-Chen Chu leg.; 0808-002001; 0808-002002; 8080-02000; CMNH • 1 ♀; same data as for preceding; CMNH568903; NRC • 1 ♀; same data as for preceding; 2 June 2011; HuiSheng Fang leg.; CMNH454573; CMNH • 1 ♂, 1 ♀; 3 June 2011; Wen-Chen Chu leg.; 0808-002063; CMNH451153; CMNH • 2 ♀♀; 8 June 2011; Han-Tzu Hsu leg.; 0808-000077; dissected; 0808-000075; dissected; CMNH • 6 ♂♂; Ren-Ai Wu-Ling Area; N24.1385, E121.2841; alt. 3146 m; 14 May 2010; Chen Young; leg.; CMNH449584; CMNH • 14 ♂♂; Ren-Ai Wu-Ling Rest Area; N24.1385, E121.2841; alt. 3146 m; 14 May 2010; Chen Young leg.; CMNH543505; CMNH543508; CMNH544504; CMNH543500; CMNH543503; CMNH543563; CMNH543499; CMNH543497; CMNH543498; CMNH452113; CMNH452884; CMNH—**Taichung Co.** • 1 ♂, 1 ♀; Heping Twp Snow Mountains; N24.2453; E121.5368; alt. 3568 m; 13 October 2011; Wen-Chen Chu leg.; 0001-000002; CMNH455375; CMNH.


**Redescription**


**Male** (Figure 50A). Body length is 13.0–16.4 mm, wing length is 16.7–20.6 mm, length of antenna is 3.3–4.3 mm. General body coloration is yellowish brown.

Head. Yellowish brown, vertex and occiput with brown median line. Rostrum yellow, dorsally thinly dusted with gray, nasus short. Palpus brown. Antenna 13-segmented, if bent backward reaching base of the wing. Scape and pedicel brownish yellow, flagellum brown. Each flagellomere, except first, with indistinct basal enlargement. Apical flagellomere small, distinctly shorter than preceding flagellomere. Long verticils shorter than length of corresponding flagellomeres.

Thorax (Figure 50B). Pronotum brown, laterally gray. Prescutum and presutural scutum with four brown-gray pruinose stripes, bordered by darker brown. Central stripes separated in the middle by thin vitta. Interspaces yellowish gray. Postsutural scutum brown, gray pruinose. Each scutal lobe with two brown-bordered spots. Scutellum brown, gray pruinose, with brown line. Mediotergite yellowish brown, pruinose, with broad brown line. Pleura brown, pruinose. Legs with coxae brown, gray pruinose. Trochanters yellow, femora yellow, darkened at tip. Tibiae brownish yellow with darkened tip. Tarsal segments brown. Claw without tooth. Wing clear pale brown, tinged with brown (Figure 50A). Wing cells marbled with darker and clear spots. Costal and subcostal area yellow. Brown postarcular spot in cells br and bm, at origin of Rs and stigma area. A faint brown cloud in the middle of br and bm and distal area of bm. Vein m-cu joining dm before fork of M_3+4_, M_1+2_ about 0.3× as long as length of cell m_1_. Halter yellow, knob basally brown.
Figure 50Habitus of *Tipula* (*Vestiplex*) *subapterogyne*. (**A**). Male, lateral view. (**B**). Male thorax, dorsal view. (**C**). Female, lateral view. Scale bars: a (**A**,**C**) = 5 mm, b (**B**) = 1 mm.
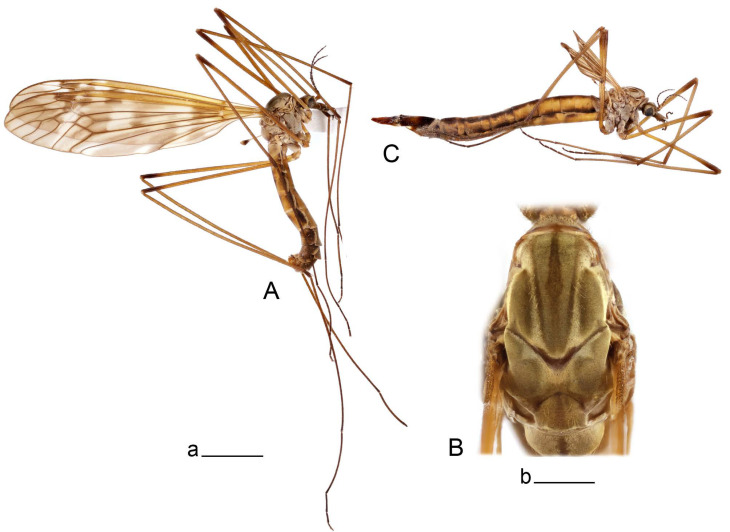



Abdomen. Abdominal segments 1–5 yellow, tergites with broad brownish black median and lateral lines. Remaining tergites brown. Sternites 1–4 yellow with broad line. Remaining sternites brown.

Hypopygium (Figure 51 and Figure 52). Male hypopygium with tergite 9 fused with sternite 9 basally (Figure 51A). Tergite 9 divided along midline by pale membrane, posteriorly with deep U-shaped notch (Figure 51B,D). Dorsal portion laterally rounded. Ventral portion terminating into narrow, blackened, microscopically serrulated lobes. Gonocoxite separated from sternite 9 by suture, unarmed (Figure 51A,C,E). Outer gonostylus leaf-shaped (Figure 51F). Inner gonostylus compressed, rounded plate terminating into obtuse beak; lower beak flattened, with obtuse point at tip; dorsal margin round; ventral margin with lobe bent outwardly (Figure 51G). Sternite 9 with ventral lobe of A9s nearly round, surface with setae (Figure 51C,H). Dorsal lobe of A9s slightly curved, finger-shaped, dilated at basal halt, tip with setae (Figure 51I). Adminiculum nearly triangular in ventral view (Figure 51C). Basal part of adminiculum broadened, protruded medially in basal half; apex curved, funnel-shaped (Figure 51C,J). Gonocoxal fragment reduced, membranuous with small vestiges of medial sclerites. Sperm pump with central vesicle swollen (Figure 52A). Compressor apodeme with median incision (Figure 52B). Posterior and anterior immovable apodemes narrow. Aedeagus about 2.5× as long as sperm pump (Figure 52A).

**Female** (Figure 50C). Body length is 18.7–23.7 mm, wing length is 4.1–7.2 mm, length of antenna is 2.3–3.2 mm. Head gray pruinose. Thorax and coxae heavily gray pruinose. Legs short and stout, length about half those of males. Wing brachypterous, short and narrow, with rudimentary trace of venation.

Ovipositor (Figure 53). Tergite 10 shiny dark brown, darker distally. Cercus brown, stocky about the same lenght as tergite 10; dorsal and ventral margins with small serration (Figure 53A). Hypovalva extending before base of cercus. Sternite 8 with hypovalva developed, blade-shaped (Figure 53B,C). Lateral angle slightly flattened. Median incision between hypovalvae basally serrated, provided with setae. Sternite 9 nearly shovel-shaped, round at posterior margin (Figure 52D). Furca in the shape of narrow stripe (Figure 52E). Spermatheca spherical, base slightly broadened (Figure 53F).
Figure 51Male terminalia of *Tipula* (*Vestiplex*) *subapterogyne*. (**A**). Hypopygium, lateral view. (**B**). Hypopygium, dorsal view. (**C**). Hypopygium, ventral view. (**D**). Tergite 9, dorsal view. (**E**). Gonocoxite. (**F**). Outer gonostylus. (**G**). Left inner gonostylus, lateral view. (**H**). Ventral lobe of A9s. (**I**). Dorsal lobe of A9s. (**J**). Adminiculum, lateral view. Scale bar: a (**A**–**J**) = 0.5 mm.
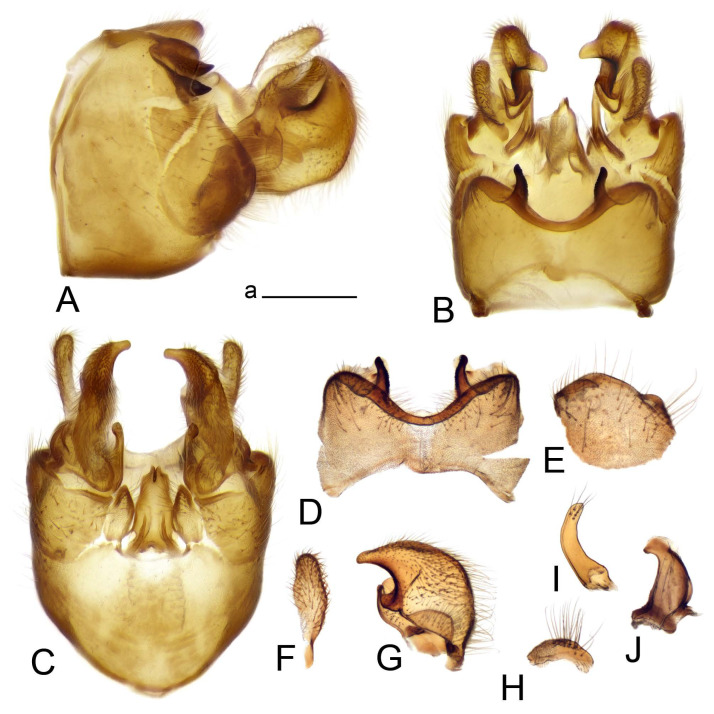

Figure 52Male terminalia of *Tipula* (*Vestiplex*) *subapterogyne*. (**A**). Semen pump and intromittent organ, lateral view. (**B**). Semen pump, dorsal view. Scale bar: a (**A**,**B**) = 0.5 mm.
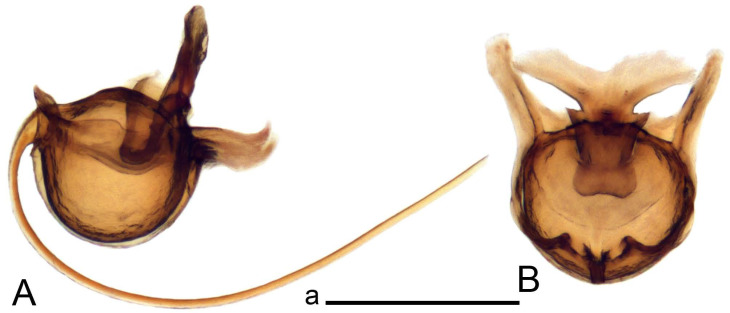

Figure 53Female terminalia of *Tipula* (*Vestiplex*) *subapterogyne*. (**A**). Ovipositor, left lateral view. (**B**). Sternite 8 with hypovalvae, ventral view. (**C**). Sternite 8 with hypovalvae, same sclerite pressed by cover glass, ventral view. (**D**). Sternite 9, dorsal view. (**E**). Furca, dorsal view (**F**). Spermatheca. Scale bars: a (**A**), b (**B**–**F**) = 0.5 mm.
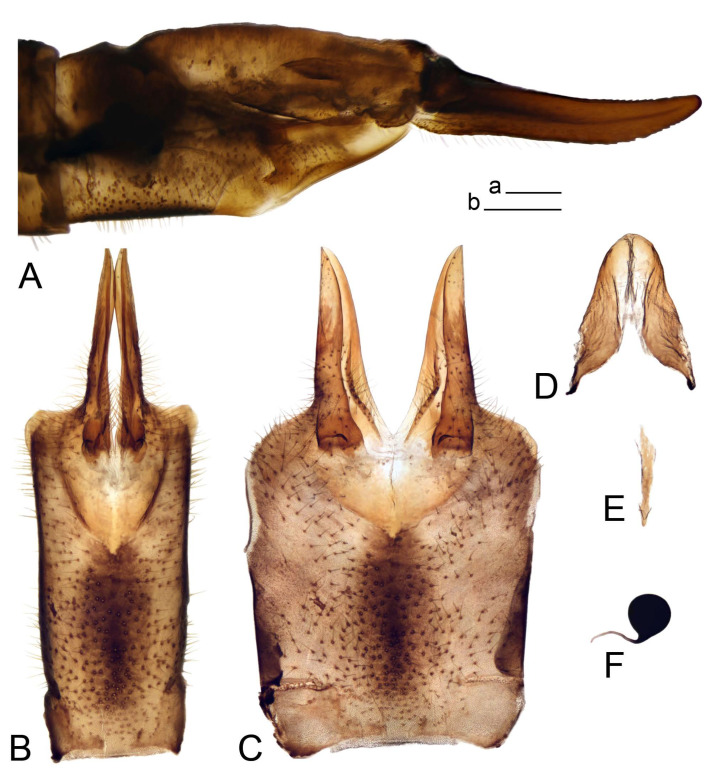



**Distribution.** Taiwan (Figure 7B).

**Elevation range in Taiwan.** Adults were collected at altitudes ranging from 2900 m to 3600 m.

**Period of activity.** Adults are active from April through to October.

**Remarks.***Tipula subapterogyne* was described by Alexander [54] based on both male and female specimens collected by T. Shiraki in 1919 from Noko (南投能高山). The female asssociation of this species is confirmed in a current DNA gene sequence study.
insects-14-00616-sch007_Scheme 7Scheme 7Distribution maps of *Tipula* (*Vestiplex*) crane flies in Taiwan. (**A**). *T.* (*V.*) *niitakensis*. (**B**). *T.* (*V.*) *subapterogyne*. Yellow circles represent paratypes.
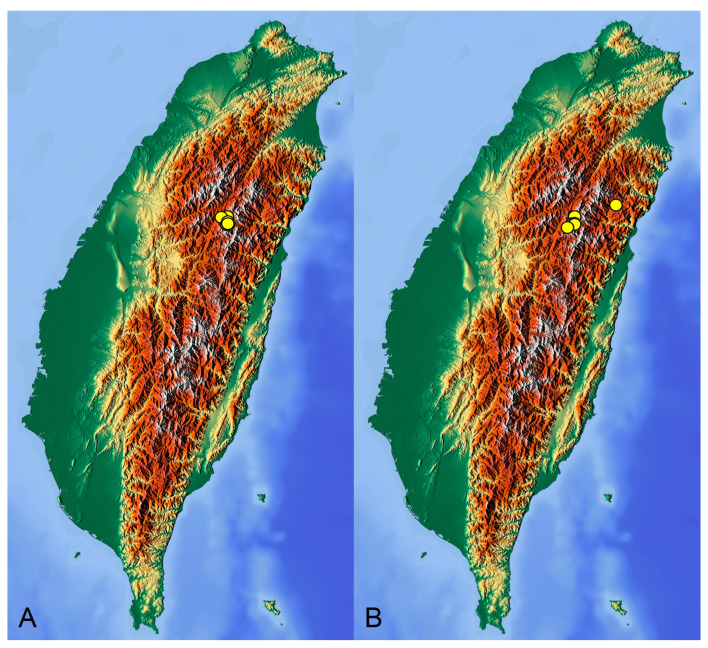



***Tipula* (*Vestiplex*) *taiwanica* Starkevich and Young sp. nov.**


Figure 54, Figure 55, Figure 56 and Figure 57; Figure 8A


**Differentional diagnosis**


*Tipula* (*V.*) *taiwanica*
**sp. nov**. can be recognized by greenish brown thoracic stripes and femora having yellow prepaical ring. The male can be also recognized by the gonocoxite which has no outgrowth, the black tergite 9 having a U-shaped notch, and black ventral lobes. The female has a smooth brown cercus, blade-shaped hypovalva, and sternite 8 with a lateral angle slightly flattened and rounded. *Tipula* (*V.*) *taiwanica* **sp. nov**. is closely related to *T*. (*V*.) *nestor* and *T*. (*V*.) *niitakensis* in having a similar, nearly crescent inner gonostylus and similar shaped ventral armatures of tergite 9, bearing a tooth and prepapical incision. The new species can be easily separated by the rounded posterolateral angle of tergite 9 which is extended into a distinct lobe in the both of related species.


**Type of material**



**Holotype**


**TAIWAN** • ♂; Hualien Co.; SiouLin, E. end of JinMa Tunnel; N24.1794, E121.3647; alt. 2456 m; 5 October 2010; CO1 Barcode TIPTW558-10; CMNH544176; CMNH.


**Paratypes**


**TAIWAN**—**Hualien Co.** • 2 ♂♂; GuanYu Youth Activities Center; N24.1879, E121.3377; alt. 2355 m; 3 May 2010; Chen Young leg.; CO1 Barcode TIPTW442-10; CMNH544060; CMNH544059; CMNH • 1 ♂; same data as for preceding; CMNH407086; dissected; NRC • 1 ♂; GuanYu woods behind gas station; N24.1853, E121.3426; alt. 2355 m; 14 May 2010; Chen Young leg.; CMNH • 2 ♂♂, 1 ♀; KuanYuan; [N24.1850667, E121.3421278]; alt. 2734 m; 16 May 2007; Ling-Chu Lin leg.; CMNH543821; CMNH544039; CMNH543822; CMNH • 1 ♀; Sioulin Highway 8 at 131 km mark; N24.1794, E121.3647; alt. 2039 m; 14 May 2014; Li-Tzen Shih leg.; CMNH544818; CMNH • 1 ♂; SiouLin, E. end of JinMa Tunnel; N24.1773, E121.3727; alt. 2508 m; 14 June 2011; Chen Young leg.; 0808-001092; CMNH • 1 ♂; same data as for preceding; 18 September 2011; Yu-Chian Chiu leg.; 0808-000881; CMNH • 1 ♀; same data as for preceding; 20 September 2011; Wen-Chen Chu leg.; 0808-000836; CMNH • 1 ♀; same data as for preceding; 18 October 2011; Han-Po Chen leg.; CMNH551709; CMNH • 1 ♂, 2 ♀♀; same data as for preceding; 14 November 2011; Han-Po Chen leg.; CMNH449407; 0808-001070; 0808-001066; CMNH • 1 ♂; same data as for preceding; 12 December 2011; Han-Po Chen leg.; CMNH551708; CMNH • 4 ♂♂, 1 ♀; SiouLin, E. end of JinMa Tunnel; N24.1794, E121.3647; alt. 2456 m; Chen Young leg.; CMNH543590; CMNH543557; CMNH543558; CMNH543591; CMNH • 2 ♂♂; same data as for preceding; CMNH450809; dissected; CMNH449321; NRC • 22 ♂♂, 38 ♀♀; same data as for preceding; 5 October 2010; CMNH544128; CMNH544125; CMNH544126; CMNH544132; CMNH449008; dissected; CMNH544127; CMNH448813; CMNH544153; CMNH544144; CMNH455117; CMNH569023; CMNH452294; CMNH544141; CMNH544181; CMNH544179; CMNH544143; CMNH544130; CMNH544139; CMNH544138; CMNH544131; CMNH544133; CMNH544136; CMNH544124; CMNH544061; CMNH544129; dissected; CMNH544123; CMNH544182; CMNH454671; CO1 Barcode TIPTW562-10; CMNH544180; CMNH544175; CMNH • 1 ♀; same data as for preceding; CMNH455012; NRC—**Nantou Co.** • 2 ♂♂, 1 ♀; Ren-Ai Endemic Species Res. Centr.; N24.1063, E121.1981; alt. 2294 m; 2 May 2011; Yu-Da Chuan leg.; CMNH544818; CMNH544819; CMNH544820; CMNH • 1 ♂, 1 ♀; same data as for preceding; 2 August 2011; 1420-000079; 1420-000078; CMNH • 1 ♂, 1 ♀; Ren-Ai Hsing-Pai-Young; N24.1980, E121.4326; alt. 1486 m; 14 November 2011; Han-Tzu Hsu leg.; CMNH576487; CO1 Barcode TIPTW1025-12; CMNH576493; CMNH • 4 ♀♀; same data as for preceding; 12 December 2011; Yu-Chian Chiu leg.; 0808-000506; 0808-000509; 0808-000511; 0808-000513; CMNH • 1 ♀; Ren-Ai Jui-Yen; N24.1063, E121.1981; alt. 2294 m; 2 May 2011; Han-Tzu Hsu leg.; CMNH575720; CMNH • 2 ♂♂, 1 ♀; Ren-Ai Township, Mei-Feng Farm; N24.0961, E121.7169; alt. 2088 m; 1 September 2011; 1420-000033; 1420-000037; 1420-000028; CMNH • 1 ♂; SinYi Twp Dan-Da Trial 18 m mark; N23.7852, E121.1743; alt. 2470 m; 3 April 2012; Han-Tzu Hsu leg.; 1420-000109; CMNH—**Taichung Co.** • 1 ♂, 5 ♀♀; Bi-LuSi, Taiwan Forest Res. Inst.; N24.2226, E121.3050; alt. 2149 m; 12 May 2010; Chen Young leg.; CMNH543569; CMNH543565; CMNH543609; CMNH543607; CMNH543566; CMNH544808; CMNH • 8 ♂♂, 12 ♀♀; same data as for preceding; 4 October 2010; CO1 Barcode TIPTW549-10; CMNH544167; CMNH544174; CO1 Barcode TIPTW555-10; CMNH544173; CO1 Barcode TIPTW545-10; CMNH544163; CMNH • 1 ♂, 1 ♀; same data as for preceding; CMNH569259; CMNH449067; NRC • 1 ♀; Heping Twp Snow Mountains; N24.2453, E121.5368; alt. 2240 m; 30 May 2011; Yu-Chuan Chiu leg.; CMNH544823; CMNH • 3 ♂♂, 3 ♀♀; Pilushi; [N24.24750, E121.30720]; alt. 2200 m; 1988.05.22-23; R. Davidson, J. Rawlins, C. Young leg.; CMNH544029; CMNH544032; CMNH544047; CMNH544045; CMNH.


**Description**


**Male** (Figure 54A). Body length is 10.8–12.6 mm, wing length is 13.7–15.8 mm, length of antenna is 3.2–4.3 mm. General body coloration is yellow.
Figure 54Habitus of *Tipula* (*Vestiplex*) *taiwanica*
**sp. nov**. (**A**). Male, lateral view. (**B**). Male thorax, dorsal view. (**C**). Female, lateral view. Scale bars: a (**A**,**C**) = 5 mm, b (**B**) = 1 mm.
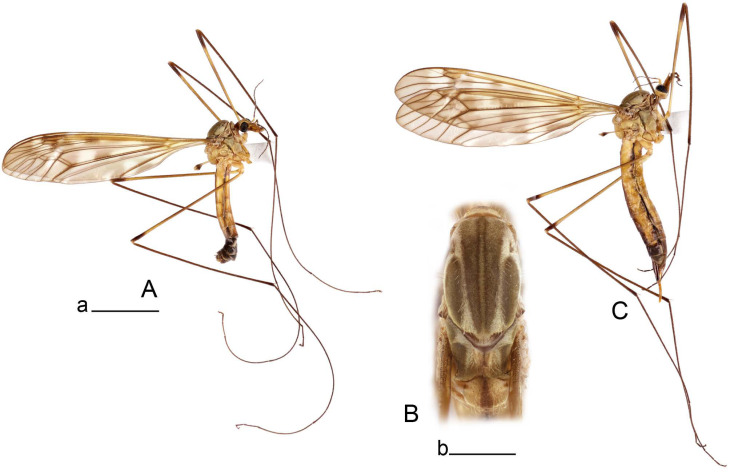



Head. Yellow, vertex and occiput with brown median line. Rostrum yellow with distinct nasus. Palpus brown. Antenna 13-segmented, if bent backward reaching base of the wing. Scape and pedicel yellow, flagellum brown. Each flagellomere, except first, slightly enlarged at base. Apical flagellomere small, distinctly shorter than preceding flagellomere. Long verticils shorter than corresponding flagellomeres.

Thorax (Figure 54B). Pronotum brown, anterior margin narrowly yellow. Prescutum and presutural scutum with four greenish brown stripes, bordered by brown. Central stripes narrowly confluent anteriorly, separated by yellow line. Interspaces yellow pollinose. Postsutural scutum yellow pollinose with brown median line. Each scutal lobe with two greenish brown spots bordered by brown. Scutellum and mediotergite yellow, with broad brown median line. Pleura yellow except brown, yellow pollinose episternum. Legs with coxae and trochanters yellow. Femora dirty yellow, blackened tip preceeded by yellow subterminal ring. Remaining segments brown. Claw with tooth. Wing clear pale brown, wing cells marbled with darker and clear spots (Figure 54A). Costal and subcostal area yellow. Brown postarcular darkening in cells br and bm, at origin of Rs and stigma area. A faint brown cloud in the middle of br and bm and distal area of bm. Vein m-cu joining dm at fork of M_3+4_, M_1+2_ about 0.2× as long as length of cell m_1_. Halter yellow, knob basally brown, distally pale yellow.

Abdomen. Tergites 1–5 yellow, with broad brown median and narrow lateral line. Sternites 1–5 yellow, remaining segments brownish black.
Figure 55Male terminalia of *Tipula* (*Vestiplex*) *taiwanica*
**sp. nov**. (**A**). Hypopygium, lateral view. (**B**). Tergite 9, dorsal view. (**C**). Tergite 9, caudal view. (**D**). Gonocoxite. (**E**). Outer gonostylus. (**F**). Left inner gonostylus, lateral view. (**G**). Ventral lobe of A9s. (**H**). Dorsal lobe of A9s. (**I**). Adminiculum, ventral view. (**J**). Adminiculum, lateral view. (**K**). Gonocoxal fragment. Abbreviations: ls—lateral sclerite of gonocoxal fragment; ms—medial sclerite of gonocoxal fragment. Scale bars: a (**A**), b (**B**–**K**) = 0.5 mm.
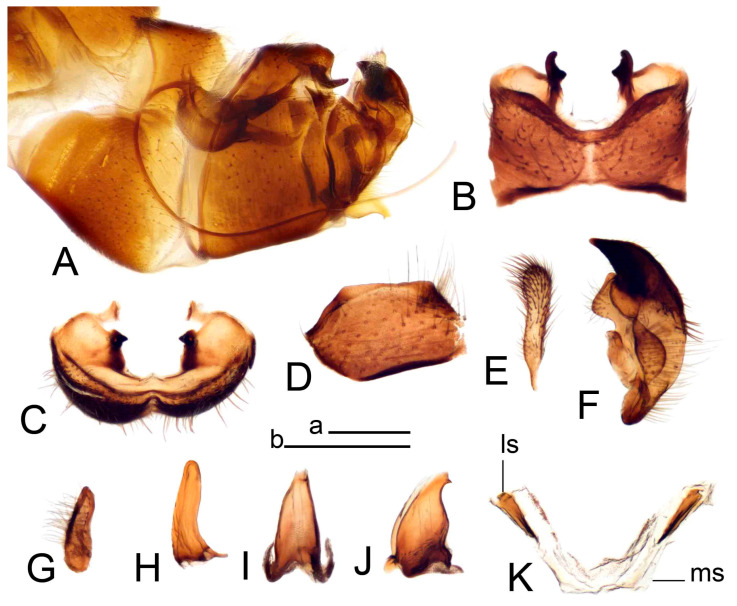



Hypopygium (Figure 55 and Figure 56). Male hypopygium with tergite 9 fused with sternite 9 basally (Figure 55A). Tergite 9 divided along midline by pale membrane, posteriorly with U-shaped notch (Figure 55B,C). Dorsal portion occupying the tergal area, covered with setae. Ventral portion terminating into slender, blackened lobes. The lobe with microscopic serrulation, dilated at base, medially with distinct tooth and shallow preapical incision. Gonocoxite separated from sternite 9 by suture, unarmed, apically with long setae (Figure 55A,D). Outer gonostylus relatively short, finger-shaped (Figure 55E). Inner gonostylus nearly crescent plate terminating into blackened obtuse beak; lower beak flattened; ventral margin with round lobe bent outwardly (Figure 55F). Sternite 9 with ventral lobe of A9s oblong, tip narrowed, surface with setae (Figure 55G). Dorsal lobe of A9s nearly finger-shaped (Figure 55H). Adminiculum nearly triangular in both ventral and lateral views, medial protrusion indistinct, apex slightly curved preceded by shallow incision (Figure 55I,J). Gonocoxal fragment with medial sclerites membranous, fused at base; lateral sclerite a narrowed plate (Figure 55K). Sperm pump with central vesicle nearly spherical (Figure 56A). Compressor apodeme flattened, with narrow median incision (Figure 56B). Posterior and anterior immovable apodeme narrow. Aedeagus about 4× as long as sperm pump, basally brown, becoming yellow towards apex (Figure 56A). Distal part funnel-shaped (Figure 56C).
Figure 56Male terminalia of *Tipula* (*Vestiplex*) *taiwanica*
**sp. nov**. (**A**). Semen pump and intromittent organ, lateral view. (**B**). Semen pump, dorsal view. (**C**). Apical part of intromittent organ. Scale bar: a (**A**,**B**) = 0.5 mm, a (**C**) = 0.25 mm.
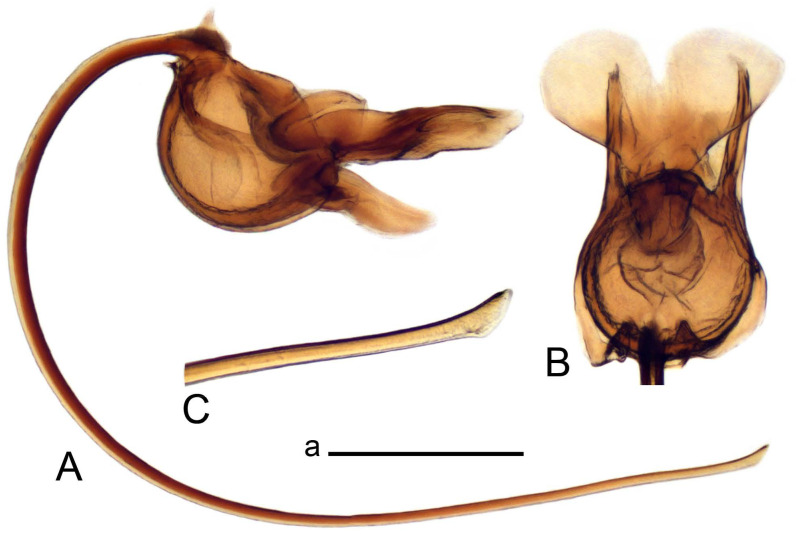



**Female** (Figure 54C). Body length is 16.2–19.4 mm, wing length is 16.0–18.3 mm, length of antenna is 2.3–2.7 mm. Generally similar to male in body coloration.

Ovipositor (Figure 57). Tergite 10 shiny brown, darker distally. Cercus brown, about the same length as tergite 10, with tip round; ventral margin without visible serration (Figure 57A). Hypovalva extending base of cercus. Sternite 8 with hypovalva developed, blade-shaped (Figure 57B,C). Lateral angle slightly flattened and rounded. Median incision between hypovalvae basally with setae. Sternite 9 nearly triangular in outline, posteriorly narrowing, with short tongue-shaped extension, anterior parts acute at base (Figure 57D). Furca in the shape of narrowed stripe (Figure 57E). Spermatheca nearly pear-shaped (Figure 57F).

**Etymology.** The new species is named after the country of Taiwan in which it was found.

**Distribution.** Taiwan (Figure 8A).

**Elevation range in Taiwan.** Adults were collected at altitudes ranging from 1500 m to 2700 m.

**Period of activity.** Adults are active from May through to December.
Figure 57Female terminalia of *Tipula* (*Vestiplex*) *taiwanica*
**sp. nov.** (**A**). Ovipositor, left lateral view. (**B**). Sternite 8 with hypovalvae, ventral view. (**C**). Sternite 8 with hypovalvae, same sclerite pressed by cover glass, ventral view. (**D**). Sternite 9, dorsal view. (**E**). Furca, dorsal view (**F**). Spermatheca. Scale bars: a (**A**), b (**B**–**F**) = 0.5 mm.
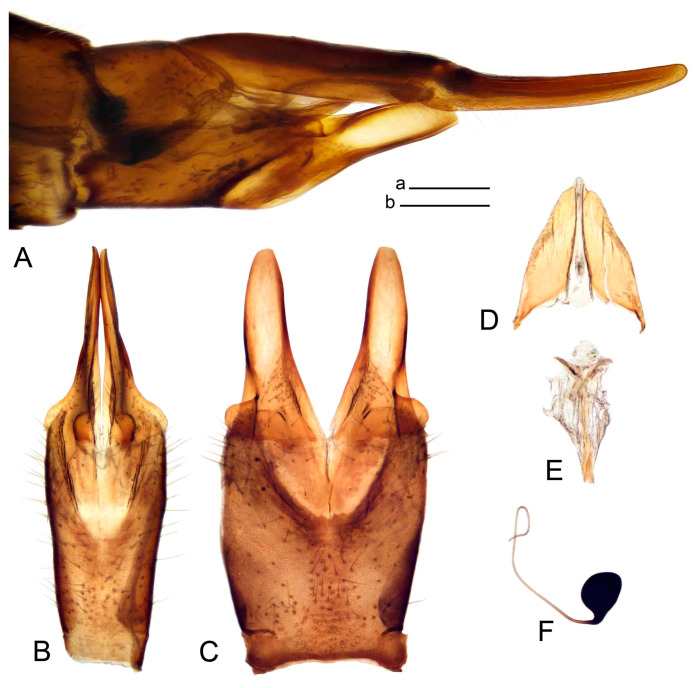




**
*Tipula*
**
**(*Vestiplex*) *terebrata* Edwards**


Figure 58, Figure 59, Figure 60 and Figure 61; Figure 8B

*Tipula terebrata* Edwards, [33]: 109;

*Tipula* (*Unplaced*) *terebrata*: Alexander and Alexander, [34]: 67;

*Tipula nokonis* Alexander, 1928: 459 [55] **syn. nov.**;

*Tipula *(*Vestiplex*) *nokonis*: Alexander, [4]: 118; [39]: 100; Alexander and Alexander, [34]: 64.


**Diagnosis**


*Tipula* (*V.*) *terebrata* can be easily recognized by its sternite 8 laterally having a group of yellow setae and sternite 9 ventrally being produced into a distinct tubercle. Males can be also recognized by their unarmed gonocoxite and tergite 9 with a U-shaped notch. Females can be recognized by a straight cercus, broadened at base, without visible serration, as well as sternite 8 with the lateral angle flattened and rounded and blade-shaped hypovalva.

**Type of material examined.** *Tipula* (*Vestiplex*) *terebrata* Edwards: **TAIWAN** • Holotype ♀; Formosa. Dr. T. Shiraki. 1920-70; Musha 10.v.1917 T. Shiraki leg.; BMNH(#)246106; BMNH.

*Tipula* (*Vestiplex*) *nokonis* Alexander: **TAIWAN** • Holotype ♂; Formosa, Noko; alt. 9800 ft.; 27 June 1927; S. Issiki leg.; antenna and wing on slide mounted; USNM.

**Additional material examined. TAIWAN**—**Hualien Co.** • 1 ♂; SiouLin, Hsio-Fon-Ko parking area; N24.1618, E121.2882; alt. 2999 m; 9 June 2011; HuiSheng Fang leg.; CMNH418070; CMNH—**Nantou Co.** • 1 ♂; Ren-Ai Endemic Species Res. Centr.; N24.1813, E121.2866; alt. 2939 m; 6 July 2011; Chen Young leg.; CMNH451687; CMNH • 1 ♂; same data as for preceding; 11 May 2010; Han-Tzu Hsu leg.; CMNH575740; CMNH • 2 ♂♂; same data as for preceding; 13 May 2010; Chen Young leg.; CMNH452563; CMNH451620; dissected; NRC • 3 ♂♂; same data as for preceding; CMNH542006; CMNH543495; CMNH451205; dissected; CMNH • 3 ♂♂; same data as for preceding; 14 May 2010; CMNH543517; CMNH543510; CMNH445416; CMNH • 1 ♂; Ren-Ai HeHuan-Shan North Peak; N24.1815, E121.2816; alt. 3424 m; 14 July 2011; Han-Tzu Hsu leg.; CMNH451211; CMNH—**Taichung Co.** • 1 ♂; Bi-LuSi, Taiwan Forest Res. Inst.; N24.2226, E121.3050; alt. 2149 m; 12 May 2010; Chen Young leg.; CMNH543606; dissected; CMNH • 1 ♀; Highway 8 at 108 mark; N24.0035, E121.2916; alt. 2410 m; 13 May 2010; Chen Young leg.; CMNH543597; dissected; CMNH • 1 ♀; Pilushi; [N24.24750, E121.30720]; alt. 2200 m; 22 May 1988; R. Davidson, J. Rawlins, C. Young leg.; CMNH544049; CMNH.


**Redescription**


**Male** (Figure 58A). Body length is 14.8–17.0 mm, wing length is 18.2–22.7 mm, length of antenna is 4.3–5.3 mm.
Figure 58Habitus of *Tipula* (*Vestiplex*) *terebrata*. (**A**). Male, lateral view. (**B**). Male thorax, dorsal view. (**C**). Female, lateral view. Scale bars: a (**A**,**C**) = 5 mm, b (**B**) = 1 mm.
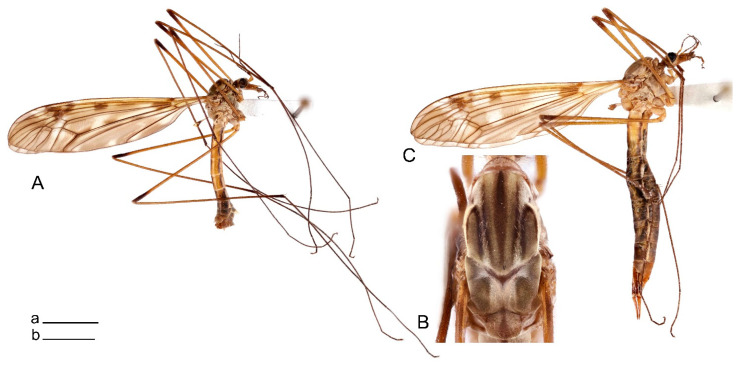



*Head*. Orange yellow, vertex and occiput with brown median line. Rostrum orange yellow with distinct nasus. Palpus brown. Antenna 13-segmented, if bent backward almost reaching base of the wing. Scape and pedicel yellow; remaining segments brown. Each flagellomere, except first, slightly enlarged at base. Apical flagellomere small, distinctly shorter than preceding flagellomere. Long verticils about the same length as corresponding flagellomeres.

Thorax (Figure 58B). Overall brown, gray pruinose. Pronotum dark brown. Prescutum and presutural scutum with four brown-gray dusted stripes, bordered by brown. Central stripes fused into broad brown line. Interspaces gray. Postsutural scutum gray. Each scutal lobe with two grayish brown spots bordered by brown. Scutellum brown, with median line. Mediotergite brown, gray pruinose, with brown line. Pleura brown. Legs with coxae and gray pruinose. Trochanters yellow. Femora dirty yellow, tip blackened. Tibiae dirty yellow at basal half, posterior half brown, tip blackened. Remaining segments brownish black. Claw without tooth. Wing brown, wing cells marbled with darker and clear spots (Figure 58A). Costal and subcostal area yellowish. Brown postarcular darkening in cells br and bm, brown spot at origin of Rs and stigma area. A faint brown cloud in the middle of br and bm and distal area of bm. Vein m-cu joining dm at fork of M_3+4_, M_1+2_ about 0.25× as long as length of cell m_1_. Halter yellow, knob basally brown, distally yellow.

Abdomen. Tergites 1–5 yellow, with broad brown median and narrow lateral line. Sternites 1–5 yellow, remaining segments brown.
Figure 59Male terminalia of *Tipula* (*Vestiplex*) *terebrata*. (**A**). Hypopygium, lateral view. (**B**). Tergite 9, dorsal view. (**C**). Tergite 9, caudal view. (**D**). Hypopygium, gonostyli removed, caudal view. (**E**). Ventral part of sternite 8, lateral view. (**F**). Gonocoxite. (**G**). Outer gonostylus. (**H**). Left inner gonostylus, lateral view. (**I**). Adminiculum, lateral view. (**J**). Ventral and dorsal lobes of A9s. Scale bars: a (**A**), b (**B**–**J**) = 0.5 mm.
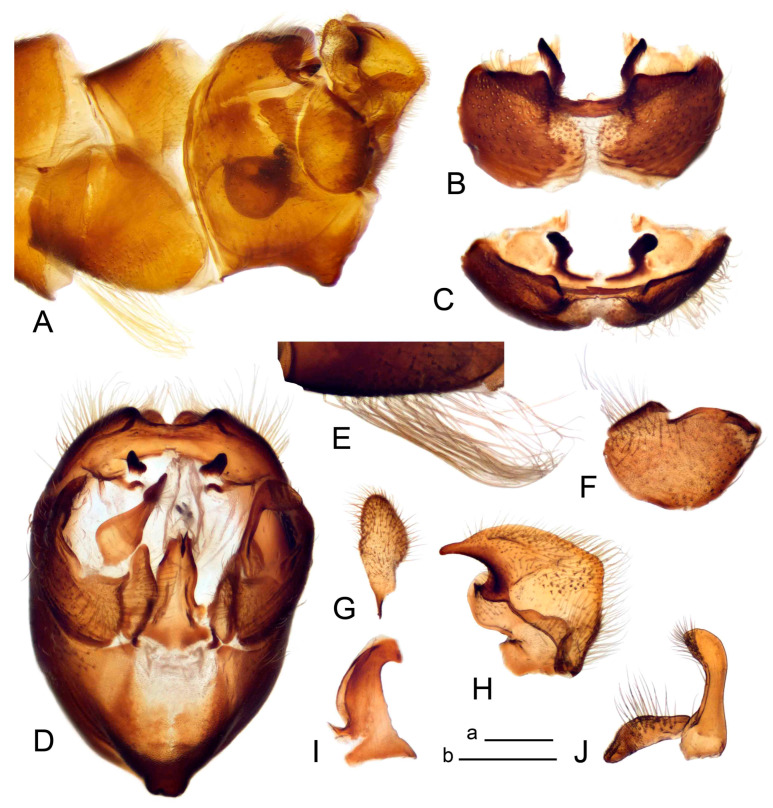



Hypopygium (Figure 59 and Figure 60). Male hypopygium with tergite 9 fused with sternite 9 basally (Figure 59A). Tergite 9 divided along midline by pale membrane, posteriorly with U-shaped notch (Figure 59B,C). Dorsal portion occupying the tergal area, covered with setae, posterior margin with obtusely rounded projection on either side of midline. Ventral portion terminating with short, blackened, microscopically roughened lobes. The lobes narrowed at base and dilated at apex. Gonocoxite separated from sternite 9 by suture, unarmed, irregular in outline, apically with long setae (Figure 59A,F). Outer gonostylus narrowed at base and dilated in the middle (Figure 59G). Inner gonostylus compressed, terminating into a short upper beak, lower beak surrounded by roundly swollen area (Figure 59H). Dorsal margin nearly rectangular in the middle, edge basally bent outward. Sternite 9 ventrally produced into prominent tubercle (Figure 59A,D). Sternite 9 with ventral lobe of A9s oblong, tip narrowed, surface with setae (Figure 59D,J). Dorsal lobe of A9s nearly finger-shaped, base slightly dilated, curved in the middle, tip narrow, on inner surface provided with setae. Adminiculum slightly dilated laterally, medially protruded (Figure 59D). Basal part of adminiculum broadened; apex curved, funnel-shaped (Figure 59I). Sternite 8 laterally provided with group of yellow setae reaching 1.05 mm long (Figure 59D,E). Gonocoxal fragment reduced. Sperm pump with central vesicle swollen (Figure 60A). Compressor apodeme flattened (Figure 60B). Posterior and anterior immovable apodemes narrowed. Aedeagus about 2.7× as long as sperm pump, basally brown, becoming yellow towards apex (Figure 60A).
Figure 60Male terminalia of *Tipula* (*Vestiplex*) *terebrata*. (**A**). Semen pump and intromittent organ, lateral view. (**B**). Semen pump, dorsal view. Scale bar: a (**A**,**B**) = 0.5 mm.
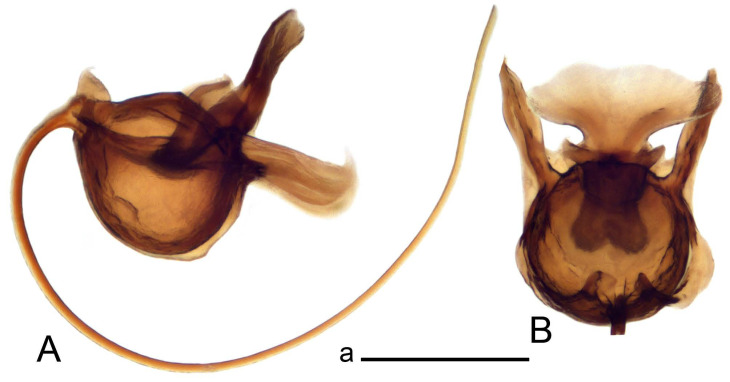



**Female** (Figure 58C). Body length is 25.4–25.6 mm, wing length is 21.2–22.1 mm, length of antenna is 2.6–3.0 mm.

Ovipositor (Figure 61). Tergite 10 shiny brown, darker distally. Cercus brown, slightly shorter than length of tergite 10, straight, broadedned at base with tip round; ventral margin without visible serration (Figure 61A). Hypovalva, extending before base of cercus. Sternite 8 with hypovalva developed, blade-shaped (Figure 61B). Lateral angle distinctly flattened and rounded. Median incision between hypovalvae basally serrated and covered with setae. Sternite 9 posteriorly flattened, shovel-shaped (Figure 61C). Furca narrowed (Figure 61D). Spermatheca pear-shaped (Figure 61E).
Figure 61Female terminalia of *Tipula* (*Vestiplex*) *terebrata*. (**A**). Ovipositor, left lateral view. (**B**). Sternite 8 with hypovalvae, ventral view. (**C**). Sternite 9, dorsal view. (**D**). Furca, dorsal view (**E**). Spermatheca. Scale bars: a (**A**), b (**B**–**E**) = 0.5 mm.
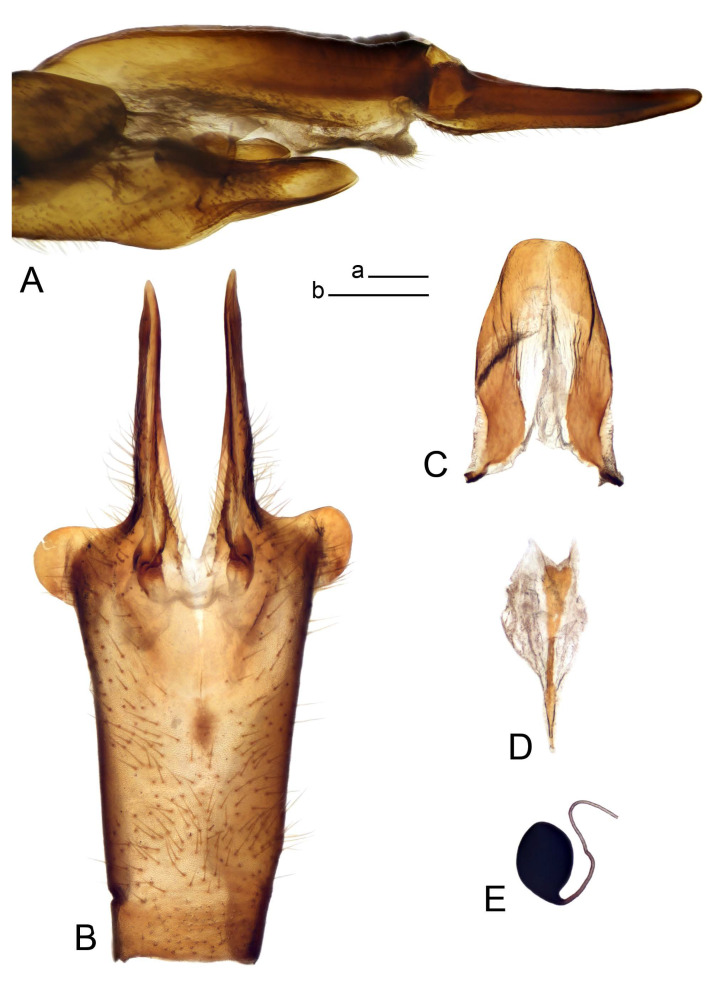



**Distribution.** Taiwan (Figure 8B).

**Elevation range in Taiwan.** Adults were collected at altitudes ranging from 2100 m to 3400 m.

**Period of activity.** Adults are active from May through to July.

**Remarks.***Tipula nokonis* was described by Alexander [55] based on a single male specimen collected by S. Issiki in 1927 from Noko (南投能高 2987 m), Taiwan. In 1938, Alexander designated a female allotype of *T. nokonis* [39] collected by R. Takahashi in 1936 from Kwanzan (高雄關山 2743 m), Taiwan (Alexander, 1938). Alexander expressed his association of the two sexes of this species based mainly on the coloration of the external morphology without giving details of their specified similarity. The male and female associations in this paper were confirmed by a DNA gene sequence study.

*Tipula terebrata* was described by Edwards [33] based on single female specimen collected by T. Shiraki in 1917 from Musha (Wushe), Taiwan. The synonymy *T.* (*V.*) *terebrata* = *T.* (*V.*) *nokonis*
**syn. nov.** was established after a comparison of type specimens of both taxa and additional non-type material collected from Taiwan.
insects-14-00616-sch008_Scheme 8Scheme 8Distribution maps of *Tipula* (*Vestiplex*) crane flies in Taiwan. (**A**). *T.* (*V.*) *taiwanica* **sp. nov**. Purple circle represents holotype, yellow circles represent paratypes. (**B**). *T.* (*V.*) *terebrata*.
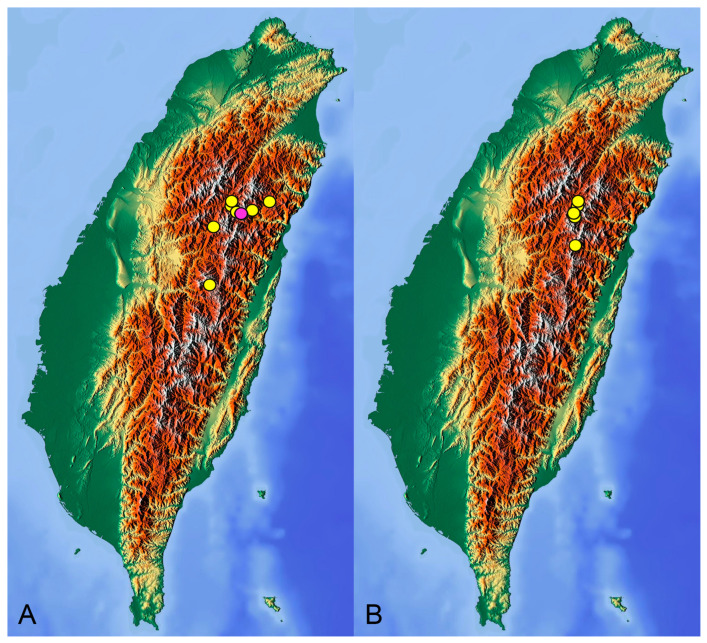




***Tipula* (*Vestiplex*) *immota* species group**


The *immota* species was proposed by Men et al. [16] and includes the following species: *T.* (*V.*) *bicornuta*; *T.* (*V.*) *dashahensis* Yang, Zhu and Liu, 2005; *T.* (*V.*) *immota* Alexander, 1935; *T.* (*V.*) *gongdangensis* Men, Starkevich and Saldaitis, 2021; and *T.* (*V.*) *dechangensis* Men, Starkevich and Saldaitis, 2021.

The males of the *immota* species group can be recognized by tergite 9 divided at the midline by a pale membrane with s pair of brown, inconspicuously protruded rounded lobes dorsally, located on either side of the midline; a dorsal portion with a posterior margin terminating into a pair of short, heavily blackened processes, and a ventral portion partially darkened, with additional microscopically darkened dots or terminating into blackened processes. Males can be also recognized by a gonocoxite dorsally produced into a horn or spine; an inner gonostylus terminating into extended upper beak; a dorsal crest nearly rectangular or angular with a blackened ridge generated from the dorsal corner and a black rim suffuse along the dorsal surface; a lower beak absent; sternite 9 with a short dorsal lobe of A9s, fused basally with a ventral lobe; and adminiculum basally flattened and dilated with a median portion distinctly protruded [16].

The female is known only for *T.* (*V.*) *dechangensis* and characterized by a smooth cercus, without visible serration, and a short, blackened, plate-shaped hypovalva [16].


**
*Tipula*
**
**(*Vestiplex*) *bicornuta* Alexander**


Figure 62, Figure 63 and Figure 64; Figure 9A

*Tipula bicornuta* Alexander, [54]: 267;

*Tipula *(*Vestiplex*) *bicornuta*: Alexander, [4]: 118; Alexander and Alexander, [34]: 62.


**Diagnosis**


*Tipula* (*V.*) *bicornuta* can be recognized by its yellow body coloration, elongated antenna reaching the base of abdomen if bent backward, and gonocoxite with a horn-shaped apical part. In addition, dorsally, tergite 9 has a pair of brown, inconspicuously protruded, rounded areas located on either side of midline and short blackened ventral processes. The female of this species is unknown.

**Type of material examined. TAIWAN** • Holotype ♂; Formosa, Funkiko; 21 April 1917; T. Shiraki leg.; antenna, wing, and genitalia mounted on slide; USNM.

**Additional material examined. TAIWAN**—**Taichung Co.** • 1 ♂; Bi-LuSi, Taiwan Forest Res. Inst.May; N24.2226, E121.3050; alt. 2149 m; 12 May 2010; Chen Young leg.; CMNH544183; CMNH—**Taoyuan Co.** • 1 ♂; Fushing Township, Lalashan Cottage; N24.7094, E.121.4332; alt. 1581 m; 6 April 2012; Wen-Chen Chu leg.; 2025-000020; dissected; CMNH.


**Redescription**


**Male** (Figure 62A). Body length is 10.2–10.6 mm, wing length is 14.1–14.3 mm, length of antenna is 5.4–5.5 mm. General body coloration is yellow.
Figure 62Habitus of *Tipula* (*Vestiplex*) *bicornuta*. (**A**). Male, lateral view. (**B**). Male thorax, dorsal view. Scale bars: a (**A**) = 5 mm, b (**B**) = 1 mm.
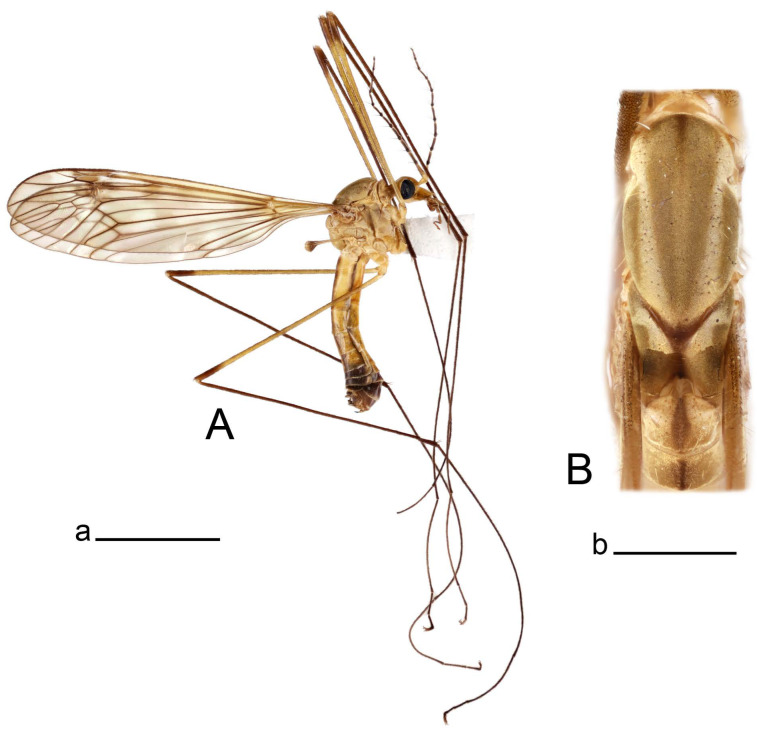



Head. Yellow, vertex and occiput with narrow median line. Rostrum yellow with short nasus. Palpus brownish yellow. Antenna 13-segmented, if bent backward reaching base abdomen. Scape pedicel and first flagellar segment yellow, remaining brown. Each flagellomere, except first, slightly enlarged at base. Apical flagellomere small, distinctly shorter than preceding flagellomere. Long verticils shorter than corresponding flagellomeres.

Thorax (Figure 62B). Pronotum yellow with median vitta. Prescutum and presutural scutum with four olive brown stripes, bordered by yellowish brown. Central stripes fused to anterior third, with the rest separated by greenish area. Interspaces yellow. Postsutural scutum yellow, with brown median line. Each scutal lobe has two olive brown spots bordered by brown. Scutellum and mediotegite brown, with broad brown median line. Pleura yellow. Legs with coxae and trochanters yellow. Femora yellow with tip blackened. Remaining segments brown. Claw with tooth. Wing subhyaline, wing cells marbled with darker and clear spots (Figure 62B). Brown postarcular darkening in cells br and bm, at origin of Rs and stigma area. A faint brown cloud in the middle of br and bm and distal area of bm. Vein m-cu joining dm at fork of M_3+4_, M_1+2_ short, about 0.2× as long as length of cell m_1_. Halter yellow, knob basally brown, distally yellowish brown.

Abdomen. Tergites 1–5 yellow, with broad brown median and narrow lateral line. Sternites 1–6 yellow, rest of abdomen brown.
Figure 63Male terminalia of *Tipula* (*Vestiplex*) *bicornuta*. (**A**). Hypopygium, lateral view. (**B**). Hypopygium, dorsal view. (**C**). Hypopygium, ventral view. (**D**). Tergite 9, dorsal view. (**E**). Tergite 9, caudal view. (**F**). Gonocoxite. (**G**). Outer gonostylus. (**H**). Left inner gonostylus, lateral view. (**I**). Left inner gonostylus, dorsal view. (**J**). Adminiculum with appendages of A9s, ventral view. (**K**). Adminiculum, lateral view. Abbreviations: dl—dorsal lobe of A9s; vl—ventral lobe of A9s. Scale bars: a (**A**,**B**), b (**C**–**K**) = 0.5 mm.
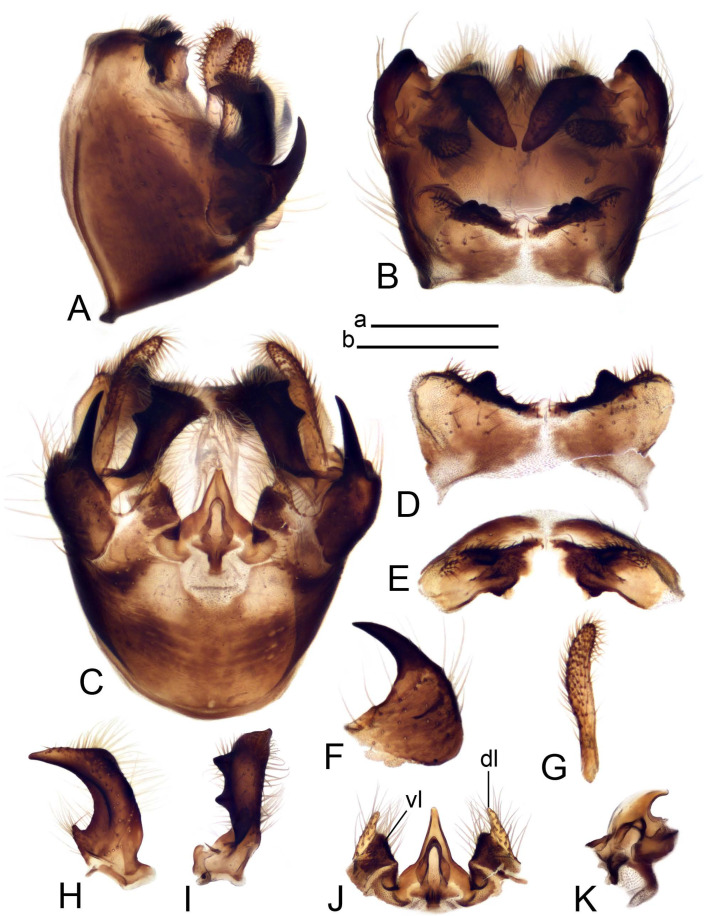



Hypopygium (Figure 63 and Figure 64). Male hypopygium with tergite 9 fused with sternite 9 basally (Figure 63A). Tergite 9 completely divided at midline by a pale membrane (Figure 63B,D,E). Dorsal portion of tergite 9 laterally pale yellow, medially with a pair of brown, inconspicuously protruded, rounded areas located on either side of midline; posterior margin covered with setae, medially with a U-shaped notch and a pair of black and short, apically rounded processes; posterolateral margin rounded. Ventral portion laterally membranous, median part darkened, covered with microscopic dots, terminating into a pair of short, blackened processes on either side of midline. Gonocoxite separated from sternite 9 by suture, apically horn-shaped (Figure 63A,F). Outer gonostylus narrowed, finger-shaped (Figure 63G). Inner gonostylus, terminating in an extended upper beak, lower beak absent (Figure 63H,I). Dorsal crest nearly rectangular with two large, triangular lobes; a suffused black rim along dorsal surface. Sternite 9 with ventral lobe of A9s dark brown, nearly triangular, covered with setae (Figure 63C,J). Dorsal lobe of A9s yellow, densely covered with setae, short, fused with ventral lobe at base (Figure 63J). Adminiculum basally expanded in basal half and dilated (Figure 63C,J); median portion, before apex, distinctly protruded with margin raised at base; apex narrowed and curved with a rounded preapical incision in lateral view (Figure 63K). Sperm pump with central vesicle swollen (Figure 64A). Compressor apodeme flattened, with median incision (Figure 64B). Posterior immovable apodeme narrowed, distally flattened, anterior immovable apodeme flattened. Aedeagus about 3.5× as long as sperm pump, basally brown, becoming yellow towards apex (Figure 64A).
Figure 64Male terminalia of *Tipula* (*Vestiplex*) *bicornuta*. (**A**). Semen pump and intromittent organ, lateral view. (**B**). Semen pump, dorsal view. Scale bar: a (**A**,**B**) = 0.5 mm.
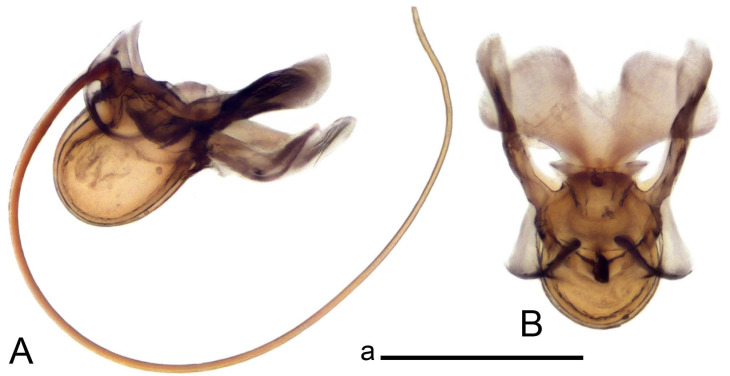



**Female**. Unknown.

**Distribution.** Taiwan (Figure 9A).

**Elevation range in Taiwan.** Adults were collected at altitudes ranging from 1600 m to 2100 m.

**Period of activity.** Adults are active from April through to May.

**Remarks.***Tipula* (*V.*) *bicornuta* was described by Alexander [54] based on single male specimen collected by T. Shiraki in 1917 from Funkiko (嘉義奮起湖).


**
*Tipula*
**
**(*Vestiplex*) *subtincta* species group**


The *subtincta* species group was proposed by Savchenko [10] for *T.* (*V.*) *subtincta* Brunetti, 1912, *T.* (*V.*) *distifurca* Alexander, 1942, and *T.* (*V.*) *optanda* Alexander, 1935. The following species belongs to the group ([15]): *T.* (*V.*) *apicifurcata* Yang and Yang, 1992, *T.* (*V.*) *dobrotworskyana* Alexander, 1968, *T.* (*V.*) *exechostyla* Alexander, 1964, *T.* (*V.*) *gandharva* Alexander, 1951, *T.* (*V.*) *hasyia* Pilipenko and Starkevich, 2020, *T.* (*V.*) *parvapiculata*, *T.* (*V.*) *quasimarmoratipennis* Brunetti, 1912, *T.* (*V.*) *tillyardana* Alexander, 1970, *T.* (*V.*) *tuta* Alexander, 1936, and *T.* (*V.*) *wahlgrenana* Alexander, 1968. Another three species have been described recently from Tibet and Yunnan, China [48].

Males of the *subtincta* species group can be recognized by tergite 9 with a dorsal portion having a pair of lobes on the posterior margin and ventral portion terminating into blackened, microscopically roughened slender or flattened lobes; sternite 9 with a long and slender dorsal lobe of A9s, dilated at the base, with an acute or split tip. In addition, they can be recognized by gonocoxite with an apex usually with a black spine or point (except for *T.* (*V.*) *exechostyla* and *T.* (*V.*) *tuta*), with the ventromesal portion in the shape of blackened arm or spine

Females are illustrated for *T.* (*V.*) *quasimarmoratipennis* [43], *T.* (*V.*) *hasyia*, *T.* (*V.*) *subtincta* [56], and *T.* (*V.*) *parvapiculata*. They are characterized by a cercus serrated on the dorsal and ventral margins and short, plate-shaped hypovalva, toothed at the posterior margin.

***Tipula* (*Vestiplex*) *parvapiculata*** Alexander, 1934

Figure 65, Figure 66, Figure 67 and Figure 68; Figure 9B

*Tipula* (*Vestiplex*) *parvapiculata* Alexander, [32]: 30;

*Tipula* (*Vestiplex*) *parvapiculata*: Alexander and Alexander, [34]: 64.


**Diagnosis**


*Tipula* (*V.*) *parvapiculata* can be recognized by the gonocoxite having a small acute black spine, and an elongated bifid at the apex appendage of sternite 9. The female has a slender cercus with a serrated ventral and distally serrated dorsal margin. The hypovalva is short and plate-shaped, with a toothed posterior margin.

**Type of material examined. TAIWAN** • Holotype ♂; Formosa, Taiheizan, Taihoku-shu; 24 October 1932; Keisho Sato leg.; antenna, leg, wing, and genitalia mounted on a slide; USNM; fragment of thorax and slide pinned on the pin are kept at the collection of Kyushu University (Japan) with images available online [53].

**Additional material examined. TAIWAN**—**Yilan Co.** • 2 ♂♂; Cilan Forest Conservation Station; N24.5879, E121.4223; alt. 1637 m; 22 April 2010; CMNH450788; dissected; CMNH448867; NRC • 10 ♂♂; same data as for preceding; CO1 Barcode TIPTW073-10; CMNH543691; CMNH415089; dissected; CMNH453099; CMNH543695; CMNH451365; CMNH415089; CMNH543691; CMNH543692; CMNH543690; CMNH • 1 ♀; Fu-Shan Taiwan Forest Res. Inst. #2Bridge, 5.5 Road Mark; N24.7602, E121.5990; alt. 592 m; 20 April 2010; Chen Young leg.; CMNH543684; CMNH • 3 ♂♂, 1 ♀; MingChih Recreation Park; N24.5771, E121.4068; alt. 1119 m; 23 April 2010; Chen Young leg.; CMNH544091; CMNH544092; CO1 Barcode TIPTW082-10; CMNH543700; CMNH • 2 ♂♂, 1 ♀; same data as for preceding; 4 May 2010; CMNH452180; CMNH442936; CMNH417804; NRC • 25 ♂♂, 1 ♀; same data as for preceding; CMNH428181; CMNH543797; CMNH543701; CMNH543769; CMNH543770; CMNH543799; CMNH543798; CMNH449340; CMNH543771; CMNH543802; CO1 Barcode TIPTW179-10; CMNH543797; CO1 Barcode TIPTW181-10; CMNH543799; CO1 Barcode TIPTW083-10; CMNH543701; CO1 Barcode TIPTW178-10; CMNH543796; CMNH—**Taoyuan Co.** • 1 ♂; Fushing Township, Lalashan Cottage; N24.57094, E.121.4332; alt. 1581 m; 6 April 2012; Han-Tzu Hsu leg.; CMNH • 2 ♀♀; same data as for preceding; 30 March 2010; Chen Young leg.; CO1 Barcode TIPTW738-11; CMNH 544491; CMNH544490; CMNH.


**Redescription**


**Male** (Figure 65A). Body length is 13.1–17.2 mm, wing length is 17.7–20.3 mm, length of antenna is 3.2–4.0 mm. General body coloration is brownish yellow.

Head. Olive gray, vertex and occiput with brown median line. Rostrum yellow, nasus conspicuous. Palpus brown. Antenna 13-segmented, if bent backward reaching base of the wing. Scape, pedicel, and first flagellar segment yellow, segments 2–5 with bicolor appearance, remaining brown. Each flagellomere, except first, with black basal enlargement. Apical flagellomere small, distinctly shorter than preceding flagellomere. Long verticils about the same length as corresponding flagellomeres.
Figure 65Habitus of *Tipula* (*Vestiplex*) *parvapiculata*. (**A**). Male, lateral view. (**B**). Male thorax, dorsal view. (**C**). Female, lateral view. Scale bars: a (**A**,**C**) = 5 mm, b (**B**) = 1 mm.
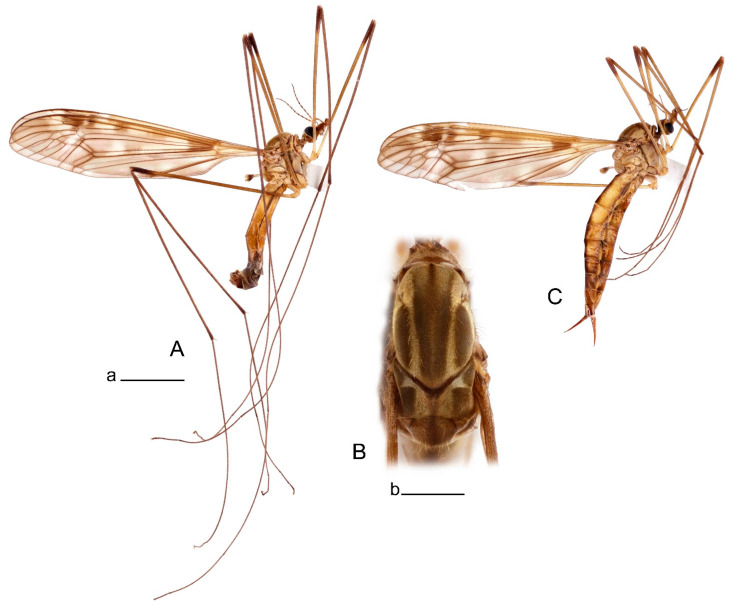



Thorax (Figure 65B). Brownish gray. Pronotum brown, posterior margin yellowish. Prescutum and presutural scutum with four grayish brown stripes, bordered by darker brown. Central stripes fused anteriorly and barely separated by thin gray vitta. Anterior third of central stripes light brown. Interspaces yellowish gray. Postsutural scutum yellowish brown, with brown median line. Each scutal lobe with two brown spots. Scutellum brown, with darker brown median line. Mediotergite yellowish brown, pruinose, with brown line. Pleura brown, pruinose. Legs with coxae and trochanters yellow. Femora yellowish brown with black tip, preceeded by broad yellow subterminal ring, which is absent on hind femora. Rest of segments brownish black. Claw with tooth. Wing clear pale brown, tinged with brown (Figure 65A). Costal and subcostal area yellowish. Brown postarcular darkening in cells br and bm, brown spot at origin of Rs and stigma area. Vein m-cu joining dm at fork of M_3+4_, M_1+2_ about 0.2× as long as length of cell m_1_. Halter yellow, knob basally brown.

Abdomen. Abdominal tergites 1–5 yellow, with broad brown median and narrow lateral lines. Remaining tergites black. Sternites 1–5 yellow with pale median line. Remaining sternites black
Figure 66Male terminalia of *Tipula* (*Vestiplex*) *parvapiculata*. (**A**). Hypopygium, lateral view. (**B**). Hypopygium, dorsal view. (**C**). Tergite 9, dorsal view. (**D**). Gonocoxite. (**E**). Outer gonostylus. (**F**). Left inner gonostylus, lateral view. (**G**). Hypopygium, gonostyli removed, caudal view. (**H**). Adminiculum, ventral view. (**I**). Dorsal lobe of A9s. (**J**). Gonocoxal fragment. Abbreviations: ap—apical point of gonocoxite; ls—lateral sclerite of gonocoxal fragment; ms—medial sclerite of gonocoxal fragment. Scale bars: a (**A**,**B**), b (**C**–**J**) = 0.5 mm.
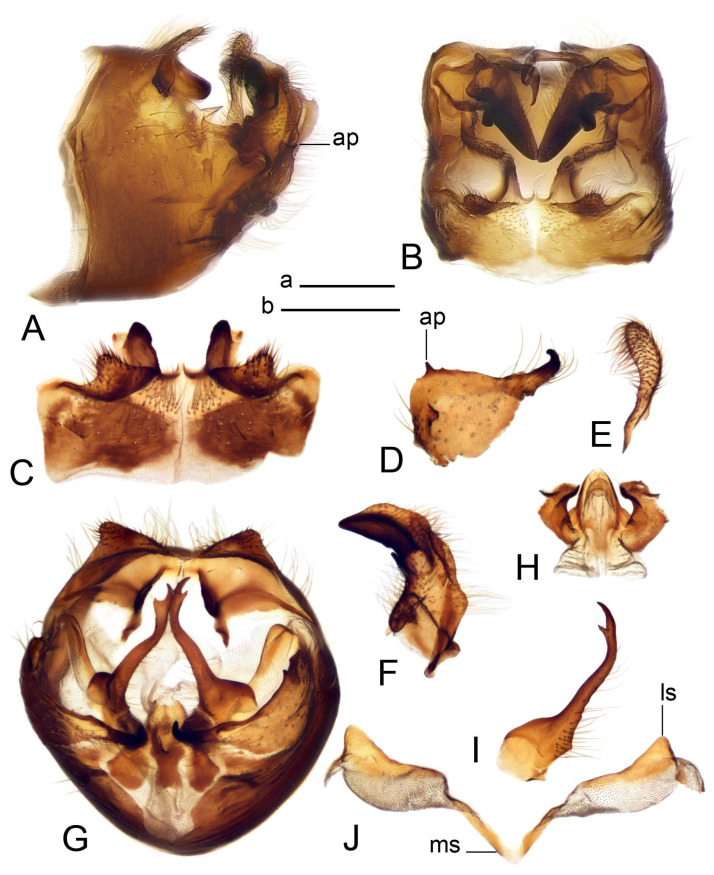



Hypopygium (Figure 66 and Figure 67). Male hypopygium with tergite 9 fused with sternite 9 basally (Figure 66A). Tergite 9 divided along midline by pale membrane, posteriorly emarginate (Figure 66B,C,G). Dorsal portion of tergite 9 with blackened central band and pale area posteriorly to band; posterior margin with a pair of nearly triangular lobes directed caudad and densely covered with setae. Ventral portion with a pair of flattened, oval, blackened microscopically roughened lobes. Gonocoxite separated from sternite 9 by suture, apically with black point, ventromesal portion produced into blackened, slender arm having curved tip (Figure 66D). Outer gonostylus slightly curved, nearly finger-shaped (Figure 66E). Inner gonostylus in the shape of sligthly curved plate that narrows gradually to a stout beak (Figure 66F). Dorsal margin at the middle with blackened angle, a yellow angle at posterior part; lower beak a blackened lobe. Sternite 9 with ventral lobe of A9s darkened, nearly triangular, covered with long setae (Figure 66G); dorsal lobe long and slender, slightly curved, broad at base, apex split into two blackened spines (Figure 66I). Adminiculum relatively small, medially protruded (Figure 66H). Gonocoxal fragment with medial sclerites narrow, fused at base, V-shaped; lateral sclerite a flattened oblong plate (Figure 66J). Sperm pump with central vesicle swollen, bubble-shaped (Figure 67A). Compressor apodeme flattened, with broad and deep median incision (Figure 66B). Posterior immovable apodeme narrowed, anterior immovable apodeme expanded and flattened. Aedeagus about 3× as long as sperm pump, basally brown, becoming yellow towards apex (Figure 67C). Distal part swollen, provided with microscopic thorns (Figure 67D).
Figure 67Male terminalia of *Tipula* (*Vestiplex*) *parvapiculata*. (**A**). Semen pump, lateral view**.** (**B**). Semen pump, dorsal view**.** (**C**). Intromittent organ, lateral view**.** (**D**). Apical part of intromittent organ. Scale bar: a (**A**–**C**) = 0.5 mm, a (**D**) = 0.25 mm.
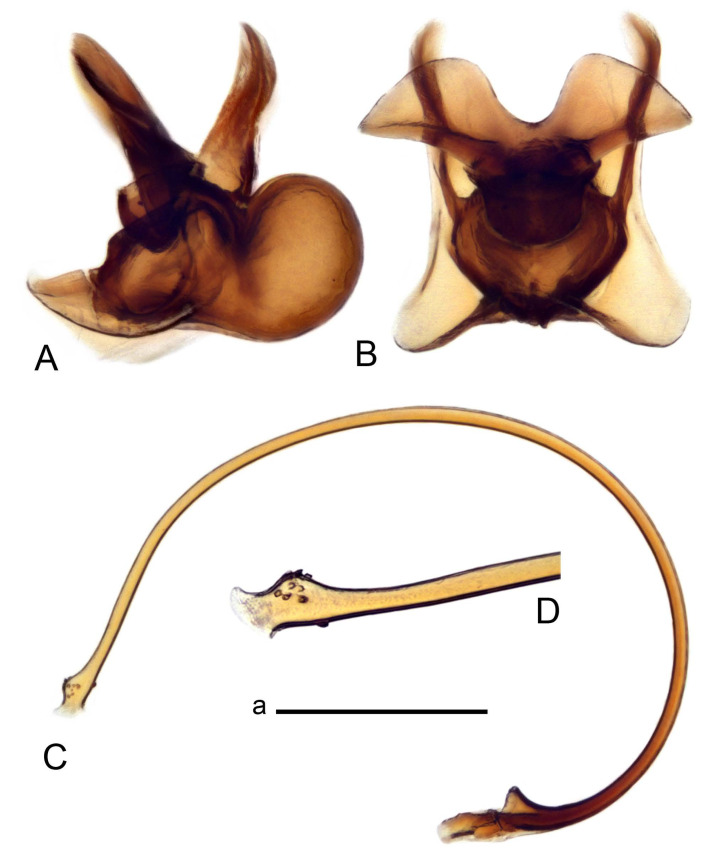

Figure 68Female terminalia of *Tipula* (*Vestiplex*) *parvapiculata*. (**A**). Ovipositor, left lateral view. (**B**). Sternite 8 with hypovalvae, ventral view. (**C**). Posterior part of sternite 8 with hypovalvae pressed by cover glass, ventral view. (**D**). Media incision between hypovalvae, ventral view. (**E**). Sternite 9 and furca. (**F**). Spermatheca. Scale bars: a (**A**), b (**B**,**C**,**E**,**F**) = 0.5 mm; c (**D**) = 0.25 mm.
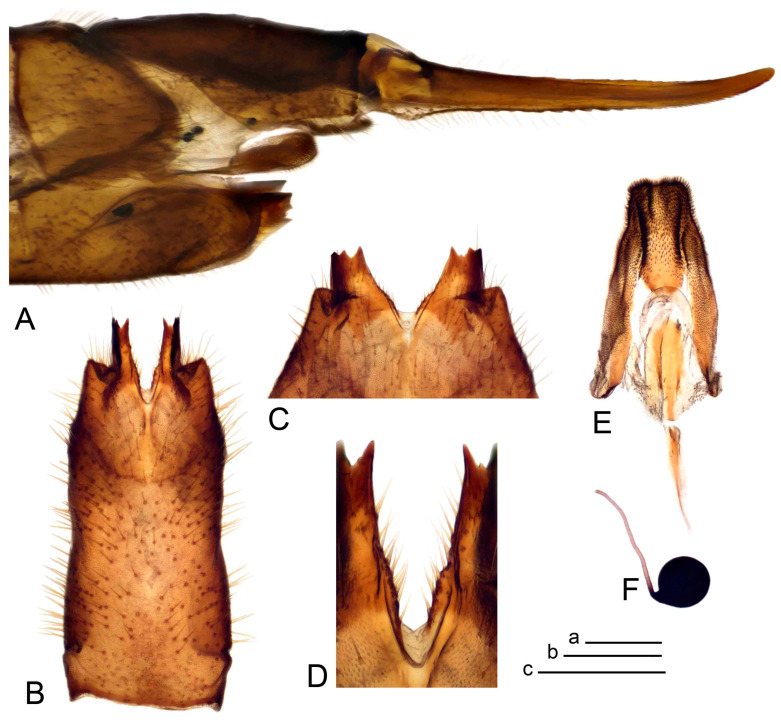



**Female** (Figure 65C). Body length is 19.6–21.5 mm, wing length is 19.1–20.2 mm, length of antenna is 2.0–2.9 mm. Generally similar to male in body coloration.

Ovipositor (Figure 68). Tergite 10 shiny brown. Cercus brown, slender, narrowing towards tip; ventral margin with small serration, dorsal margin serrated on distal half (Figure 68A). Sternite 8 with hypovalva short, plate-shaped (Figure 68B–D). Outer part of hypovalva blackened, posterior margin toothed, with deep median and shallow lateral incisions. Median incision between hypovalvae with indistinctly serrated edge and covered with setae. Lateral angle of sternite 8 sclerotized, nearly triangular with tip narrowed. Sternite 9 with posterior half shovel-shaped, surface covered with short filaments, medially with shallow groove, posterior margin nearly straight, with indistinct lateral incisions; anterior parts straight (Figure 66E). Furca narrow (Figure 66E). Spermatheca spherical (Figure 66F).

**Distribution.** Taiwan (Figure 9B).

**Elevation range in Taiwan.** Adults were collected at altitudes ranging from 600 m to 1600 m.

**Period of activity.** Adults are active from March through to October.

**Remarks.** *Tipula parvapiculata* was described by Alexander [32] based on a single male specimen collected by Keisho Sato in 1932 from Taiheizan (宜蘭太平山). The female asssociation of this species is based on a current DNA gene sequence study.
insects-14-00616-sch009_Scheme 9Scheme 9Distribution maps of *Tipula* (*Vestiplex*) crane flies in Taiwan. (**A**). *T.* (*V.*) *bicornuta*. (**B**). *T.* (*V.*) *parvapiculata*. Yellow circles represent paratypes.
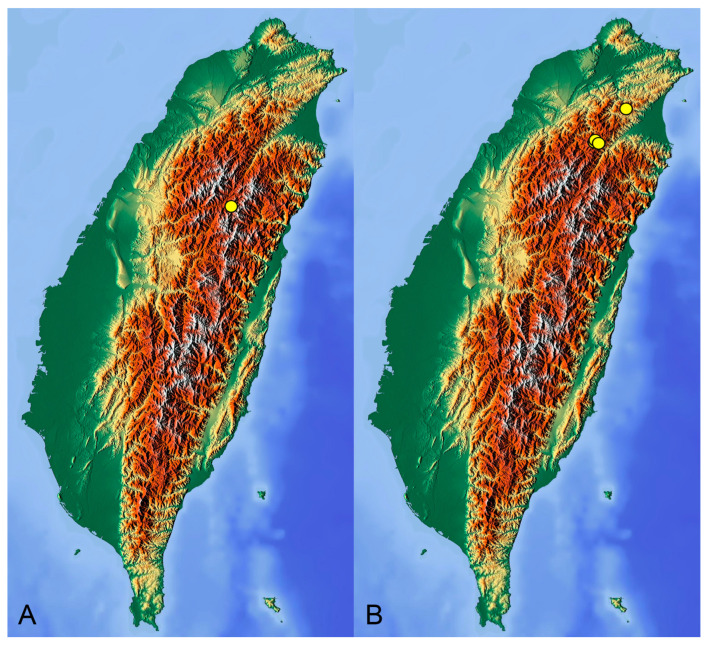



## 4. Discussion

This is the fourth contribution of a faunistic research project dealing with the Tipuloidea of the island of Taiwan [17]. This paper represents an updated taxonomic treatment of Taiwanese *T.* (*Vestiplex*) species and is based on a substantially increased foundation of recently field-collected specimens. The species delimitation in this study was achieved based on both morphological and molecular data. A CO1 barcode library for all Taiwanese *T.* (*Vestiplex*) species has been established. This allowed for testing species delimitations based on morphological features, testing of the association of the male and female stages for a species, and testing of species group clusters.

The CO1 barcode library provided the ability to identify all life stages and associating adults of different sexes represented an important extension of existing identification capabilities. Species delimitation in this study was based on both morphology (male and female terminalia) and mtDNA sequence (CO1) data, which in this study have shown significant congruence with each other (Figure 69). All species level taxa are strongly supported with a bootstrap value of >95, except for *T.* (*V.*) *nestor* with a bootstrap value of 75. The sequence data has also facilitated a positive association of the sexes when both were available for analyses, including the male and female associations of the new species, except for *T.* (*V.*) *diamondi* **sp. nov**. whose female stage is still unknown.

The sequence data have also allowed an analysis of species relationships as represented by species groups. In this study, clusters corresponding to *bicornigera*, *deserrata*, and *coxitalis* species groups are strongly supported with bootstrap values of >90, providing support for the species groupings first based on morphological characters. For the *arisanensis* and *divisotergata* species groups, the sequence clusters have a low value of support of <50, indicating conflict between species group recognition by morphology and molecular characters.

The highest species diversity of *T.* (*Vestiplex*) is observed in the East Palaearctic (81 species) and Oriental (80 species) regions [5]. There are 97 *T.* (*Vestiplex*) species documented in China [5], with the vast majority distributed in mountainous regions of the country (The Qinghai–Tibetan Plateau (QTP), [57]) such as Tibet, Sichuan, Yunnan, Qinghai, and other provinces. The southern part of China, including areas closest to Taiwan, contains only 17 species distributed in Anhui, Guangxi, Zhejiang, Guizhou, Fujian, and Jiangxi provinces [5]. Japan, a series of islands to the north of Taiwan, has only seven species of *T.* (*Vestiplex*). Taiwan has a surprising diversity of *T.* (*Vestiplex*) with 18 species, nearly 10% of the total *T.* (*Vestiplex*) species. Of these, only two species are distributed outside Taiwan, with *T.* (*V.*) *coxitalis* and *T.* (*V.*) *sternotuberculata* also distributed in the mainland of China. The distribution of the other 16 species are restricted to Taiwan, showing an extremely high endemicity rate (89%) at the species level.

In contrast, at the species group level, Taiwan has no endemic species groups. Other members of species groups described in this paper (*arisanensis*, *bicornigera*, *coxitalis*, *deserrata*, *divisotergata*, *immota*, and *subtincta*) are distributed in Oriental and East Palaearctic regions with the main diversity in northern India and China, but also Indochina, with *T.* (*V.*) *verecunda* reaching S. Korea, Japan, and the far east of Russia [49]. This suggests preliminarily that the Taiwanese *T.* (*Vestiplex*) fauna has a remarkable affinity with that of Qinghai–Tibetan Plateau.

## Figures and Tables

**Figure 69 insects-14-00616-f069:**
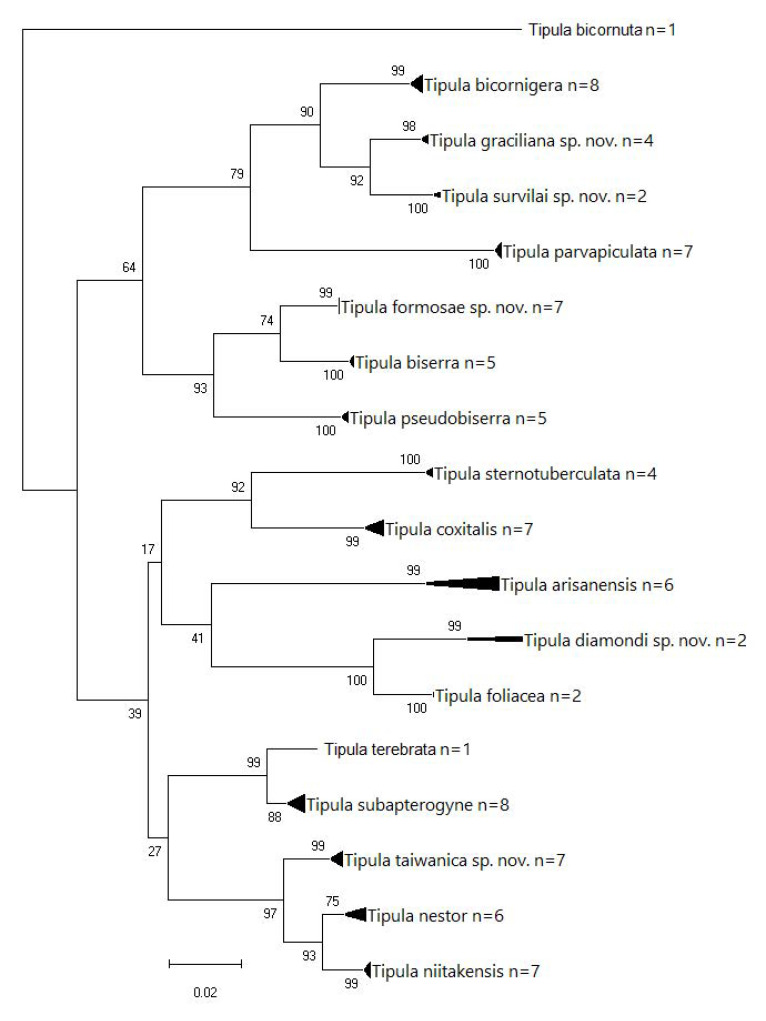
Maximum likelihood tree based on CO1 sequences (mtDNA) of 18 species of *T.* (*Vestiplex*) of Taiwan. *n* = designates number of specimens. Numerical numbers on branches indicate bootstrap values after 1000 replications. Scale bar: nucleotide substitutions per site.

**Table 1 insects-14-00616-t001:** Information for voucher specimens from Taiwan used in CO1 barcode analyses.

Sample	Specimen Number	Species	Sex
TIPTW052-10	CMNH543670	*T.* (*V.*) *arisanensis*	male
TIPTW059-10	CMNH543677	*T.* (*V.*) *arisanensis*	male
TIPTW1013-12	CMNH576481	*T.* (*V.*) *arisanensis*	male
TIPTW160-10	CMNH543778	*T.* (*V.*) *arisanensis*	female
TIPTW531-10	CMNH544149	*T.* (*V.*) *arisanensis*	male
TIPTW976-12	CMNH576444	*T.* (*V.*) *arisanensis*	female
SATIP1096-10	CMNH543574	*T.* (*V.*) *bicornigera*	male
SATIP1107-10	CMNH543585	*T.* (*V.*) *bicornigera*	female
TIPTW1066-12	CMNH576534	*T.*(*V.*) *bicornigera*	male
TIPTW420-10	CMNH544038	*T.*(*V.*) *bicornigera*	male
TIPTW543-10	CMNH544161	*T.*(*V.*) *bicornigera*	female
TIPTW547-10	CMNH544165	*T.*(*V.*) *bicornigera*	female
TIPTW568-10	CMNH544186	*T.*(*V.*) *bicornigera*	female
TIPTW817-11	CMNH544570	*T.* (*V.*) *bicornigera*	male
	2025-000020	*T.* (*V.*) *bicornuta*	male
TIPTW1034-12	CMNH576502	*T.* (*V.*) *biserra*	female
TIPTW1035-12	CMNH576503	*T.* (*V.*) *biserra*	male
TIPTW744-11	CMNH 544497	*T.* (*V.*) *biserra*	male
TIPTW745-11	CMNH 544498	*T.* (*V.*) *biserra*	male
TIPTW755-11	CMNH 544508	*T.* (*V.*) *biserra*	female
SATIP1083-10	CMNH543561	*T.* (*V.*) *coxitalis*	male
SATIP1136-10	CMNH543614	*T.*(*V.*) *coxitalis*	male
SATIP1140-10	CMNH543618	*T.*(*V.*) *coxitalis*	male
TIPTW070-10	CMNH543688	*T.*(*V.*) *coxitalis*	male
TIPTW176-10	CMNH543794	*T.*(*V.*) *coxitalis*	male
TIPTW177-10	CMNH543795	*T.*(*V.*) *coxitalis*	male
TIPTW887-11	CMNH544805	*T.*(*V.*) *coxitalis*	female
TIPTW1028-12	CMNH576496	*T.*(*V.*) *diamondi* **sp. nov.**	male
TIPTW903-11	CMNH544821	*T.*(*V.*) *diamondi* **sp. nov.**	male
TIPTW1113-12	CMNH551699	*T.*(*V.*) *foliacea*	male
TIPTW1114-12	CMNH551700	*T.*(*V.*) *foliacea*	female
SATIP1023-10	CMNH543501	*T.*(*V.*) *formosae* **sp. nov.**	female
SATIP1045-10	CMNH543523	*T.*(*V.*) *formosae* **sp. nov.**	male
SATIP1126-10	CMNH543604	*T.*(*V.*) *formosae* **sp. nov.**	male
SATIP1132-10	CMNH543610	*T.*(*V.*) *formosae* **sp. nov.**	male
TIPTW1076-12	CMNH576544	*T.*(*V.*) *formosae* **sp. nov.**	male
TIPTW1124-12	CMNH551710	*T.*(*V.*) *formosae* **sp. nov.**	female
TIPTW554-10	CMNH544172	*T.*(*V.*) *formosae* **sp. nov.**	female
TIPTW781-11	CMNH544534	*T.*(*V.*) *graciliana* **sp. nov.**	male
TIPTW782-11	CMNH544535	*T.*(*V.*) *graciliana* **sp. nov.**	male
TIPTW802-11	CMNH544555	*T.*(*V.*) *graciliana* **sp. nov.**	male
TIPTW826-11	CMNH544579	*T.*(*V.*) *graciliana* **sp. nov.**	female
TIPTW1129-12	CMNH551715	*T.*(*V.*) *nestor*	male
TIPTW1131-12	CMNH551717	*T.*(*V.*) *nestor*	male
TIPTW1132-12	CMNH551718	*T.*(*V.*) *nestor*	female
TIPTW1133-12	CMNH551719	*T.*(*V.*) *nestor*	female
TIPTW494-10	CMNH544112	*T.*(*V.*) *nestor*	female
TIPTW540-10	CMNH544158	*T.*(*V.*) *nestor*	female
TIPTW1046-12	CMNH576514	*T.*(*V.*) *niitakensis*	female
TIPTW1047-12	CMNH576515	*T.*(*V.*) *niitakensis*	female
TIPTW1048-12	CMNH576516	*T.*(*V.*) *niitakensis*	female
TIPTW1091-12	CMNH576559	*T.*(*V.*) *niitakensis*	male
TIPTW1106-12	CMNH576574	*T.*(*V.*) *niitakensis*	male
TIPTW1107-12	CMNH576575	*T.*(*V.*) *niitakensis*	male
TIPTW1108-12	CMNH576576	*T.*(*V.*) *niitakensis*	male
TIPTW073-10	CMNH543691	*T.*(*V.*) *parvapiculata*	male
TIPTW082-10	CMNH543700	*T.*(*V.*) *parvapiculata*	female
TIPTW083-10	CMNH543701	*T.*(*V.*) *parvapiculata*	male
TIPTW178-10	CMNH543796	*T.*(*V.*) *parvapiculata*	female
TIPTW179-10	CMNH543797	*T.*(*V.*) *parvapiculata*	male
TIPTW181-10	CMNH543799	*T.*(*V.*) *parvapiculata*	male
TIPTW738-11	CMNH 544491	*T.*(*V.*) *parvapiculata*	female
TIPTW1137-12	CMNH551723	*T.* (*V.*) *pseudobiserra* **sp. nov.**	female
TIPTW1139-12	CMNH551725	*T.* (*V.*) *pseudobiserra* **sp. nov.**	male
TIPTW490-10	CMNH544108	*T.* (*V.*) *pseudobiserra* **sp. nov.**	female
TIPTW492-10	CMNH544110	*T.* (*V.*) *pseudobiserra* **sp. nov.**	female
TIPTW493-10	CMNH544111	*T.* (*V.*) *pseudobiserra* **sp. nov.**	male
SATIP1137-10	CMNH543615	*T.*(*V.*) *sternotuberculata*	female
SATIP1138-10	CMNH543616	*T.*(*V.*) *sternotuberculata*	male
TIPTW217-10	CMNH543835	*T.*(*V.*) *sternotuberculata*	male
TIPTW432-10	CMNH544050	*T.*(*V.*) *sternotuberculata*	male
SATIP1035-10	CMNH543513	*T.*(*V.*) *subapterogyne*	male
SATIP1038-10	CMNH543516	*T.*(*V.*) *subapterogyne*	male
SATIP1041-10	CMNH543519	*T.*(*V.*) *subapterogyne*	female
SATIP1042-10	CMNH543520	*T.*(*V.*) *subapterogyne*	female
SATIP1077-10	CMNH543555	*T.*(*V.*) *subapterogyne*	male
SATIP1139-10	CMNH543617	*T.*(*V.*) *subapterogyne*	male
TIPTW921-11	CMNH575727	*T.*(*V.*) *subapterogyne*	female
TIPTW926-11	CMNH575732	*T.*(*V.*) *subapterogyne*	male
SATIP1043-10	CMNH543521	*T.* (*V.*) *survilai* **sp. nov.**	female
TIPTW780-11	CMNH 544533	*T.* (*V.*) *survilai* **sp. nov.**	male
TIPTW1025-12	CMNH576493	*T.* (*V.) taiwanica* **sp. nov.**	female
TIPTW442-10	CMNH544060	*T.* (*V.*) *taiwanica* **sp. nov.**	male
TIPTW545-10	CMNH544163	*T.* (*V.*) *taiwanica* **sp. nov.**	male
TIPTW549-10	CMNH544167	*T.* (*V.*) *taiwanica* **sp. nov.**	male
TIPTW555-10	CMNH544173	*T.* (*V.*) *taiwanica* **sp. nov.**	male
TIPTW558-10	CMNH544176	*T.* (*V.*) *taiwanica* **sp. nov.**	male
TIPTW562-10	CMNH544180	*T.* (*V.*) *taiwanica* **sp. nov.**	female
	CMNH573740	*T.*(*V.*) *terebrata*	male

## Data Availability

All data are provided here except DNA sequences which are available upon request from the corresponding author.

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
