# Peer review of "Taxonomic Revision of Tipula (Vestiplex Bezzi) Crane Flies (Diptera, Tipulidae) in Taiwan with Descriptions of Six New Species†"

_insects, 2023, doi:10.3390/insects14070616_

Round 1
Reviewer 1 Report
In my opinion, a fantastic publication with a detailed review of the Taiwanese species of the Tipula subgenus (Vestiplex Bezzi). At the taxonomic level, the article is well detailed, and although I am not a specialist in this group, I have not detected errors at the morphological level. The photos are of high quality, which praises the publication and can serve as a reference at the identification level. I have indicated some details, and small things that I have not seen entirely clear. Although, at the format level, the text must be completely revised, and adapted to the publication rules of the Insects journal. In addition, I recommend that the text be reviewed very carefully, since I have detected some errors. Finally, congratulations to the researchers for the fabulous publication.

English is fine for me.
Author Response
Dear Reviewer, thank You for Your valuable review. I have addressed to the all the raised issues, still I got a question regarding the following comments. Could you please specify the request?
Line 800: lobe. Gonocoxite separated from sternite 9 by suture, apically produced into black spine
Line 802: Dorsal margin without lobes, outer basal lobe flattened, with apical margin terminating into acute, blackened tooth.
format? Please, revise the format of the entire document

Reviewer 2 Report
Very good and useful work. However, the quality of the work is reduced by many formal errors. As reviewer, I am not obliged to remove formal errors, but I warmly recommend the authors to carefully read the work and remove these errors. As an example, I may pin some of them:
References should be arranged by numbers as usual in „Insects“
Tipula taiwanica (Alexander, 1923) is name already used
Line 17: juonior
Line 34: why underlined names?
L. 54: „diverse significantly“ (differ?)
L. 177: „Wing patterns various both intraspecific“ (varies?)
L. 334 not formatted alternative of the couplet
L. 366: „Tipula (Unplaced)“ reversed italics
Badly formatted text:e.g., lines 340-525 differently distant from left side of page than the following, the same size of letters and distance of rows, compare e.g., lines 948-969 with 564-850.
l. 1021-22 (chinese letters?) 1102, 1909, etc.
L. 2097 DOI without https
Author Response
Review 2.
Very good and useful work. However, the quality of the work is reduced by many formal errors. As reviewer, I am not obliged to remove formal errors, but I warmly recommend the authors to carefully read the work and remove these errors. As an example, I may pin some of them:
References should be arranged by numbers as usual in „Insects“. The manuscript will be formatted according to MDPI, thank You.
Thank You for the evaluation, we are working on all type of errors including those listed here.
Tipula taiwanica (Alexander, 1923) is name already used. – Thank You for for this comment, the name can really cause the issue, still taiwanica is used for other genus prettty similar in name: Tipulodina taiwanica Alexander, 1923
Line 17: juonior – corrected.
Line 34: why underlined names? – not underlined in primary docx file.
- 54: „diverse significantly“ (differ?) – refering to the diversity (polymorphism).
- 177: „Wing patterns various both intraspecific“ (varies?) – corrected.
- 334 not formatted alternative of the couplet – corrected.
- 366: „Tipula (Unplaced)“ reversed italics – corrected.
Badly formatted text:e.g., lines 340-525 differently distant from left side of page than the following, the same size of letters and distance of rows, compare e.g., lines 948-969 with 564-850 – The text is properly formatted in docx file, apparently this consequence of uploading process to MDPI system.
- 1021-22 (chinese letters?) 1102, 1909, etc. – yes, Chinese characters.
- 2097 DOI without https – „https“ added.

Reviewer 3 Report
This is potentially a very valuable contribution to the knowledge of the crane fly subgenus Vestiplex as it occurs in Taiwan. The discussions of the individual species are detailed, the illustrations in general are good, the distribution maps of the species are informative (although the colour green for the localities of the holotypes seems to be less well-chosen), and the CO1 barcodes are valuable additions to more traditional character information.
However, the manuscript is marred by repetitive lines and very poor English in the Abstract, Introduction, Materials and methods section, and Discussion. Remarkably, the species descriptions do not show such flaws, probably because these parts were actually (co-)written by Chen Young. (Some minor typo’s occur in the species descriptions, which can be easily corrected using a spell checker).
The style and grammar of the English of the sections mentioned should be improved before the manuscript can be processed further.
In the Acknowledgements the authors are indicated as the ‘corresponding author’ , ‘P. Starkevich’, ‘junior author’, and ‘Chen Young’. Please avoid possible confusion by simply referring to the authors by their names. From the Acknowledgements it remains unclear whether both authors collected specimens in the field or only one of them actually did so.
As for the illustrations it is confusing why some parts are depicted in a right lateral view, while most illustrations give a left lateral view. This goes especially for the sperm pump and intromittent organ (e.g. Figs. 6J, 8I, 40A, 44C), which could give the impression that in the species concerned the intromittent organ is curled in an opposite direction compared to most species.
It seems that Chen Young did not actually contribute much to the poorly written sections mentioned above, nor corrected them. My advice is at least to do so in order to improve the expression of thought, the quality and support of the assertions made, the overall style and grammar. And maybe have the final result read by a native speaker before resubmitting the manuscript.
Author Response
Review 3
Comments and Suggestions for Authors
This is potentially a very valuable contribution to the knowledge of the crane fly subgenus Vestiplex as it occurs in Taiwan. The discussions of the individual species are detailed, the illustrations in general are good, the distribution maps of the species are informative (although the colour green for the localities of the holotypes seems to be less well-chosen), and the CO1 barcodes are valuable additions to more traditional character information.
Remark: I have added pink red and green-cyan dots please choose more suitable (see first page).
However, the manuscript is marred by repetitive lines and very poor English in the Abstract, Introduction, Materials and methods section, and Discussion. Remarkably, the species descriptions do not show such flaws, probably because these parts were actually (co-)written by Chen Young. (Some minor typo’s occur in the species descriptions, which can be easily corrected using a spell checker).
The style and grammar of the English of the sections mentioned should be improved before the manuscript can be processed further.
Comments on the Quality of English Language
It seems that Chen Young did not actually contribute much to the poorly written sections mentioned above, nor corrected them. My advice is at least to do so in order to improve the expression of thought, the quality and support of the assertions made, the overall style and grammar. And maybe have the final result read by a native speaker before resubmitting the manuscript.
I will send the manuscript to C. Young for additional correction and language improvement, thank You.
In the Acknowledgements the authors are indicated as the ‘corresponding author’ , ‘P. Starkevich’, ‘junior author’, and ‘Chen Young’. Please avoid possible confusion by simply referring to the authors by their names. From the Acknowledgements it remains unclear whether both authors collected specimens in the field or only one of them actually did so. – Corrected.
As for the illustrations it is confusing why some parts are depicted in a right lateral view, while most illustrations give a left lateral view. This goes especially for the sperm pump and intromittent organ (e.g. Figs. 6J, 8I, 40A, 44C), which could give the impression that in the species concerned the intromittent organ is curled in an opposite direction compared to most species. – I have corrected the following illustrations: 6J, 8I, 40A, 44C and 6F, 35E (now it is left ig).

Reviewer 4 Report
This is an excellent piece of work. The perfect and abundant photographs (not superfluous!) with detailed descriptions will serve not only the specialists of the genus Tipula of Taiwan. Very valuable is the association of conspecific males and females by the barcoding; this information (which species) can be also introduced to the Abstract.
1. However, in order to help also the specialists of the world fauna of this genus and family I recommend to introduce also some biogeographical context. Namely: as the Taiwanese species of Tipula are mostly endemic (15 out of 18 species), the question arises what fauna are they most related to, and is this „true” endemism (I mean, the species evolved on Taiwan), or is this result of poor knowledge of the Chinese (and maybe Philippines) fauna. Of course this information cannot be provided now. But your very good characteristics of the groups of species, and the list of these species would help to draw some conclusions on directions of migration/speciation if you add distribution of these species in the world (in minimalistic form, just countries or regions together with species’ names; still better, also accompanied by some conclusions in the Discussion).
2. The young and starting specialists of the genus benefit also if you could provide sketch drawings (or black silhouettes) of features typical for the species groups (exemplified by one species for the group). For instance, the arrangement of the outer and inner gonostylus is not provided, only the structures separated from each other; for me it was also difficult to understand that the dorsal plate is on the sternite (Fig. 3). Therefore, some general groundplan would be much welcome.
3. Venation nomenclature shall be shown on some introductory picture (can be on the first photo of the wing, Fig. 1). In a paper like this, accompanied by the key, it is not enough to state what nomenclature you use.
4. l. 482 and next diagnoses: the differences between closest or most similar relatives should be included in the Diagnosis which becomes then the Differential diagnosis (recommended by the Code of Zool. Nomenclature). Now they are listed in the Remarks at end of descriptions.
5. The place of diagnosis is just under the species name, not after the list of specimens.
6. The etymology of a new species is provided only in one species. I do not consider the lack of etymology a fault, but most authors do. It is necessary when you name a species after some person (survilai?).
Zoobank numbers for new species should be included on acceptance.
Editorial remarks:
Fig. 2. : ig is not listed in Abbreviations; dl is absent here (listed in Abbr.)
Fig. 3. (A). Is: Sternite 9, ventral view, tergite 9 and gonostyli removed. Should be ...(tergite 9 and gonostyli removed). [ALL is removed?]
l. 452: Fig. 4. Hypovalva does not reach base of ovipositor (in the text: ...extending base of...).
I could not check all figures in given time; please check carefully the remaining ones.
The English language is not native to me, but I found several errors, e.g.:
Title: Taxonomic revision of Tipula (Vestiplex Bezzi) should be changed to: Taxonomic revision of Tipula (Vestiplex) Bezzi (the proper form is just below, in the Abstract)
l. 32. Is: ...that share by species... Should be: ...are shared by species...
l. 69. Is: The species of subgenus T. (Vestiplex) has been separated into various species groups... Should be: ..have been..
l. 70: Is: morphological structures of hypopygium... Should be: ... male hypopygium.
l. 94, 96. Is: tipuloidea, should be: Tipuloidea (however, the adjective form: tipuloid can be used in less formal context)
l. 463. Is: The male and female associations of this species is confrimed... Should be: are confirmed
l. 531. Is: (Men, 2015: Fig. 21); should be (Men, 2015: fig. 21) [in the references cited „fig.” begins with a small letter; the capital is restricted to the present paper only]
l. 615. Is: ..tergal arms of segment 9 is much small... Should be: ..tergal arms of segment 9 are much smaller...
I had no time to check all the text, but I feel the profound checking is much needed.
Reviewer 5 Report
This is a very good paper. I think this is a new level of research Tipulidae. The article is well prepared, but I did not find references in the text to the maps that are attached. I think it should be in Distribution for each species. Lines 948-969 are in larger type.
Author Response
Review 5.
This is a very good paper. I think this is a new level of research Tipulidae. The article is well prepared, but I did not find references in the text to the maps that are attached. I think it should be in Distribution for each species. Lines 948-969 are in larger type.
Thank You for such evaluation. The reference to the maps is given below the species name alongside with figures. I have also added the reference to the distribution section for each species as well. Lines 948-969 are the in the same format in docx file, apparently this issue is caused by uploading text to MDPI system.
